
# Self-force framework for transition-to-plunge waveforms

**Lorenzo Küchler[1,2]⋆, Geoffrey Compère[2]†, Leanne Durkan[3]‡ and Adam Pound[1]○**

**1** School of Mathematical Sciences and STAG Research Centre,
University of Southampton, Southampton SO17 1BJ, United Kingdom
**2** Université Libre de Bruxelles and International Solvay Institutes,
C.P. 231, B-1050 Bruxelles, Belgium
**3** Center for Gravitational Physics, Department of Physics,
University of Texas at Austin, Austin, TX, USA, 78712

⋆ l.m.kuchler@soton.ac.uk , † geoffrey.compere@ulb.be ,
‡ leanne.durkan@austin.utexas.edu , ○ a.pound@soton.ac.uk

## Abstract

Compact binaries with asymmetric mass ratios are key expected sources for next-generation gravitational wave detectors. Gravitational self-force theory has been successful in producing post-adiabatic waveforms that describe the quasi-circular inspiral around a non-spinning black hole with sub-radian accuracy, in remarkable agreement with numerical relativity simulations. Current inspiral models, however, break down at the innermost stable circular orbit, missing part of the waveform as the secondary body transitions to a plunge into the black hole. In this work we derive the transition-to-plunge expansion within a multiscale framework and asymptotically match its early-time behaviour with the late inspiral. Our multiscale formulation facilitates rapid generation of waveforms: we build second post-leading transition-to-plunge waveforms, named 2PLT waveforms. Although our numerical results are limited to low perturbative orders, our framework contains the analytic tools for building higher-order waveforms consistent with post-adiabatic inspirals, once all the necessary numerical self-force data becomes available. We validate our framework by comparing against numerical relativity simulations, surrogate models and the effective one-body approach.



# 1   Introduction

Future space-based gravitational wave (GW) detectors such as the Laser Interferometer Space Antenna (LISA) [1] will facilitate new, high-precision tests of general relativity. Due to launch in the mid-2030s, LISA will detect GWs in the mHz frequency band. A key source of such GWs are extreme-mass-ratio inspirals (EMRIs), binary systems in which a supermassive black hole (the primary) of mass $M$ is orbited by a stellar-mass compact object (the secondary) [2]. EMRIs naturally lend themselves to modelling by black hole perturbation theory, where the secondary is treated as a point-like particle of mass $m_p$ with no internal structure, which perturbs the background spacetime governed by the primary. Quantities are then expanded around their background value as an expansion in powers of the small mass ratio, defined by $\varepsilon := m_p/M$, with typical ranges of $10^{-4} - 10^{-7}$ [2].

At the time of writing, the small-mass-ratio expansion, in conjunction with a multiscale (or two-timescale) framework [3,4], has thus far been used to model EMRIs and their emitted waveforms during inspiral for generic orbits in a Kerr background at leading order in $\varepsilon$ [5,6]. The structure of the multiscale approach (in combination with hardware acceleration and other methods) has also enabled waveform generation that is sufficiently rapid for GW data analysis [7]. In the special case of a Schwarzschild background and quasi-circular orbits, these results have been extended through next-to-leading order in $\varepsilon$ [8], which corresponds to second order in gravitational self-force (GSF) theory. The multiscale expansion for quasi-circular orbits [9] takes account of the fact that the orbital phase of the secondary's motion, $\phi_p$, evolves on the fast timescale $\sim M$, whereas the orbital parameters such as the orbital radius $r_p$ and orbital frequency $\Omega$, in addition to the mass and angular momentum of the primary, evolve on the much slower radiation-reaction timescale $\sim M/\varepsilon$. Such an approach (and therefore the waveform model in [8]) is incomplete, however, because the inspiral dynamics break down at the innermost stable circular orbit (ISCO). As the secondary transitions to a plunge into the primary, the orbital parameters evolve more rapidly, on a timescale $\sim M/\varepsilon^{1/5}$ [10,11], whereas the primary mass and angular momentum still evolve on the timescale $\sim M/\varepsilon$. An evolution scheme that takes into account these three disparate timescales is therefore required around the ISCO.

A treatment of the transition to plunge that asymptotically matches with the quasi-circular inspiral for equatorial orbits (in Kerr spacetime) was studied by two of us in [12–14]. That work, however, focused on the orbital motion, not expanding the Einstein field equations, which prevented the construction of transition-to-plunge waveforms. In this paper we present a framework that incorporates the inspiral and the transition-to-plunge regimes both for the secondary's motion and the metric perturbation, which will allow us to build waveform models that extend beyond the ISCO. Our formulation of the transition-to-plunge expansion also differs from past formulations in a way that should naturally facilitate rapid waveform generation.

Accurately modelling the transition to plunge is expected to improve parameter estimation by matched filtering with detected signals. Crucially, this improvement will be dramatically more significant for larger values of $\varepsilon$. Indeed, the duration of the inspiral scales as $\varepsilon^{-1}$, whereas the duration of the transition to plunge scales as $\varepsilon^{-1/5}$. Ignoring the ringdown, taking the ratio of these two timescales tells us that for binaries with mass ratios of 1:10, the transition to plunge takes up $\sim 16\%$ of the entire waveform. For mass ratios of 1:1000, this reduces to $\lesssim 0.4\%$. While GWs are loudest around merger, EMRIs accumulate the majority of their signal-to-noise ratio (SNR) during their long-lasting inspirals, whereas detecting intermediate-mass-ratio coalescences (IMRACs)[1] and comparable-mass binary coalescences relies on the relatively high SNRs around the transition to plunge and merger. Therefore, accurately modelling the transition to plunge becomes more important as $\varepsilon$ increases. An accurate and proper handling

---

[1]We use the terminology of [15,16] instead of intermediate-mass-ratio inspirals (IMRIs).

of the transition-to-plunge (and plunge) regime represents an important step towards full inspiral-merger-ringdown self-force waveforms for such binaries. This will prevent parameter-estimation biases caused by the abrupt termination of inspiral-only waveforms [17] and could also be useful even for EMRI data analysis studies by avoiding ad hoc choices of truncation.

From [8,18], there is evidence to suggest that, at least for a Schwarzschild background and when carried to second perturbative order, the small-mass-ratio expansion accurately describes GWs for mass ratios as large as $\varepsilon \sim 1/10$. Hence, it is reasonable to assume that a small-mass-ratio expansion during the transition to plunge will similarly be applicable for IMRACs as well as EMRIs. The relevance of the transition to plunge for IMRACs and the expected validity of the small-mass-ratio expansion are the main motivations for this work. Our waveform modelling effort therefore also serves as preparatory modelling for third-generation ground-based detectors such as the Einstein Telescope, with expected signals from IMRACs [19]. Further motivation arises from the fact that IMRACs occupy part of the parameter space of mass ratios that is particularly challenging to model. Numerical relativity (NR) has achieved great success in simulating compact binary systems with mass ratios of 1:1 to 1:10. It has also made progress towards the 1:100 regime [20] (and even the 1:1000 regime, for head-on collisions [21]). However, systems with such small mass ratios become prohibitively computationally expensive for NR to simulate. The approach of post-Newtonian (PN) theory, effective for large orbital separations and weak fields, has also had great success. However, systems with small mass ratios spend many orbits in the strong field regime where PN theory loses accuracy. Other approaches to GW modelling, such as phenomenological models, surrogates, and effective one-body (EOB) approaches, require input from first-principles methods. EOB, in particular, synthesizes results from NR, PN and GSF theory to cover a broad parameter space [22]. In addition to providing a first-principles framework (there are no free parameters except for the physical ones), our model should provide qualitatively new information from GSF methods for universal models such as EOB [10, 23–30].

The paper is outlined as follows. Section 2 introduces the equations governing the secondary's motion and presents the Einstein field equations formulated using hyperboloidal slicing and a tensor spherical harmonic decomposition. Section 3 contains the multiscale expansions of the orbital motion, the Einstein field equations and the self-force for the quasi-circular inspiral. We perform these expansions through second post-adiabatic (2PA) order. Despite only needing to model the inspiral to first post-adiabatic (1PA) order [3], we derive the subleading terms to better capture the structure of the asymptotic match with the transition-to-plunge regime. We then compute the near-ISCO behaviour of all quantities (orbital variables, metric perturbation and self-force), which we will match with the corresponding early-time transition-to-plunge solutions. Analogously to the inspiral expansion of section 3, in section 4 we perform the multiscale expansion of the transition-to-plunge dynamics. The transition-to-plunge expansion parameter is $\lambda := \varepsilon^{1/5}$, which implies that each order of $\varepsilon$ corresponds to five orders in $\lambda$. We consider the transition-to-plunge expansion to the seventh post-leading transition-to-plunge (7PLT) order, that is, up to corrections of order $\lambda^7$ with respect to the leading-order term. We finally compute the asymptotic early-time solutions of the orbital quantities, the metric perturbation and the self-force with the aim of matching the near-ISCO inspiral. In section 5 we analytically verify the asymptotic match between the near-ISCO inspiral (to 2PA order) and the early-time transition-to-plunge (to 7PLT order) solutions. This scheme of matched asymptotic expansions enables us to obtain quantities in the transition-to-plunge expansion in terms of already known inspiral quantities, ultimately reducing the number of equations we need to solve. In section 6 we present the waveform generating scheme and the numerical implementation of 2PLT waveforms. We compare our results with NR simulations from the SXS collaboration [31] and surrogate waveform models [32, 33]. In section 7 we also compare our transition-to-plunge model with the one of Apte and Hughes [34] and to

the EOB approach [10, 22]. Finally, we present our conclusions in section 8. The appendices contain relevant analytical expressions, which are also provided as supplementary material in a GitHub repository.

## 2 Coupled Einstein's equations and compact body motion

In this section we present the equations governing the orbital evolution of the secondary and the structure of the perturbatively expanded Einstein field equations in a tensor spherical harmonic basis. The full spacetime metric $\mathbf{g}_{\mu\nu}$, comprising the background $g_{\mu\nu}$ of the primary and the perturbation $h_{\mu\nu} \sim \varepsilon$ due to the small secondary, can be written as

$$\mathbf{g}_{\mu\nu} = g_{\mu\nu} + h_{\mu\nu}. \tag{1}$$

We consider the primary as a Schwarzschild black hole, described by the metric

$$g_{\mu\nu} = \mathrm{diag}(-f, f^{-1}, r^2, r^2 \sin^2\theta)$$

in Boyer-Lindquist coordinates $(t, r, \theta, \phi)$, where $f := 1 - 2M/r$. This background metric is used to raise and lower indices. The tortoise coordinate $x$ is defined from $dx = dr/f$.

We formulate the Einstein field equations using hyperboloidal slicing. The hyperboloidal time $s$ is defined as

$$s := t - \kappa(x), \tag{2}$$

where $\kappa$ is the height function. We consider the slicing such that $s = t$ in a neighbourhood of the worldline ($\kappa(x_p) = 0$) and it becomes null as $x \to \pm\infty$, where $\lim_{x\to\pm\infty} \kappa(x) = \pm x$ (see figure 1 of [9]). We also define

$$H(r) := \left.\frac{d\kappa}{dx}\right|_{x=x(r)}. \tag{3}$$

The primary's mass and spin evolve due to the GW fluxes of energy and angular momentum through its horizon. In order to build a consistent perturbative expansion, we need to take into account this dynamical change. We write the black hole's total mass as $M + \varepsilon\,\delta M$ and total spin as $\varepsilon\,\delta J$, where $M$ is the constant mass of the Schwarzschild background $g_{\mu\nu}$ and $\delta M(s)$ and $\delta J(s)$ are the evolving corrections (normalized by $\varepsilon$), which appear in the metric perturbation $h_{\mu\nu}$.

Within this general setting, we will adopt a multiscale expansion in each of the two regimes we consider: the inspiral and the transition to plunge. Our multiscale expansions follow the approach developed in references [4, 8, 9, 35]; see, for example, appendix A of reference [9] (or the more self-contained section IIA of reference [18]), section IV of reference [35], and section 7 of reference [4]. The key idea in this approach is that the particle's trajectory and the spacetime metric only depend on the time $s$ through their dependence on a set of dynamical mechanical variables that characterize the binary. This allows us to recast the Einstein equations, coupled to the companion's equation of motion, as a problem on the binary's mechanical phase space. Generating waveforms then divides into an offline step (solving the problem on the phase space) followed by an online step (evolving along a physical trajectory in the phase space). We will recall key advantages of this approach over the course of our analysis. In this section, we will describe the coupled field equations and orbital evolution in a form that applies to both the inspiral and the transition to plunge; we then specialize to each of the two regimes in subsequent sections.

## 2.1 Orbital motion and binary phase space

We consider the motion of the secondary on quasi-circular orbits in the equatorial plane of the primary. The worldline $z^\mu(\varepsilon, t)$ can be parametrized as

$$z^\mu(\varepsilon, t) = \left(t, r_p(\varepsilon, J^a(\varepsilon, t)), \frac{\pi}{2}, \phi_p(\varepsilon, t)\right), \tag{4}$$

where, recall, we label spacetime coordinates with a subscript $p$ when evaluated on the worldline, and the hyperboloidal time $s$ reduces to $t$ on the worldline. The quantities $J^a = (\Omega, \delta M, \delta J)$ are the set of mechanical parameters that characterize the slowly evolving binary system: the orbital frequency $\Omega$ is related to the azimuthal phase $\phi_p$ by

$$\frac{d\phi_p}{dt} = \Omega, \tag{5}$$

while $\delta M$ and $\delta J$ are the corrections to the primary's mass and spin described above. Note that throughout this paper, we suppress functional dependence on the background mass $M$.

In the inspiral regime, $(J^a, \phi_p)$ represent good coordinates on the binary phase space. The multiscale expansion in the inspiral will consist of writing all quantities of interest as functions of $(\varepsilon, J^a, \phi_p)$ and then performing expansions in powers of $\varepsilon$ at fixed $(J^a, \phi_p)$. This approach fails during the transition to plunge, which occurs in a narrow frequency interval of width $\sim \varepsilon^{2/5}$ around the ISCO frequency. In the transition-to-plunge regime, we will therefore adopt a new frequency coordinate:

$$\Delta\Omega := \frac{\Omega - \Omega_*}{\varepsilon^{2/5}}, \tag{6}$$

where $\Omega_* := 1/(6\sqrt{6}M)$ denotes the geodesic ISCO frequency. By construction, $\Delta\Omega \sim \varepsilon^0$ in the transition-to-plunge regime. Our multiscale expansion in this regime will then consist of expansions in (non-integer) powers of $\varepsilon$ at fixed $(\Delta J^a, \phi_p)$, where $\Delta J^a := (\Delta\Omega, \delta M, \delta J)$.

Any function of $(\varepsilon, J^a, \phi_p)$ can be re-expressed equivalently as a function of $(\varepsilon, \Delta J^a, \phi_p)$. In this section we use the notation $(\Delta)J^a$ to denote either $J^a$ for the inspiral or $\Delta J^a$ for the transition to plunge. We will also use the notation $\delta M^+ := \delta M$ and $\delta M^- := \delta J$, which is motivated by the fact that $\delta M$ is the correction to the leading even-parity multipole moment, while $\delta J$ is the correction to the leading odd-parity multipole moment. We define $(\Delta)J^a$ as functions of hyperboloidal time $s$: on a given slice of constant $s$, $(\Delta)\Omega$ is equal to its value at the point where the slice intersects the worldline, and $\delta M^\pm$ are equal to their values where the slice intersects the horizon. In both the inspiral and the transition to plunge, the state of the system can be computed at a given value of $(\Delta)J^a$, and the system can then be evolved to new values using an evolution equation of the form

$$\frac{d(\Delta)J^a}{ds} = F^{(\Delta)J^a}(\varepsilon, (\Delta)J^b). \tag{7}$$

The forcing terms $F^{(\Delta)\Omega}$ will be obtained in terms of the self-force using the equation of motion (10) given below, while $F^{\delta M}$ and $F^{\delta J}$ are determined from the horizon fluxes of energy and angular momentum. We remark that the solutions to the ordinary differential equations (5) and (7) explicitly depend on $\varepsilon$, which justifies the $\varepsilon$ dependence of $\phi_p$ introduced in eq. (4).

Since we use $t$ as our time parameter along the particle's worldline, we will write the particle's equation of motion directly in terms of it. Defining the redshift $U := dt/d\tau$, where $\tau$ is the proper time as measured in the background spacetime, we can write the four-velocity $u^\mu := dz^\mu/d\tau$ as

$$u^\mu(\varepsilon, (\Delta)J^a) = U(\varepsilon, (\Delta)J^a)\left(1, F^{(\Delta)J^b}(\varepsilon, (\Delta)J^c)\frac{\partial r_p(\varepsilon, (\Delta)J^d)}{\partial(\Delta)J^b}, 0, \Omega\right), \tag{8}$$

with summation over the repeated $b$ index. The normalization of the four-velocity for massive particles, $g_{\mu\nu}u^{\mu}u^{\nu} = -1$, leads to an equation for the redshift,

$$U^{-2} = -g_{\mu\nu}\frac{dz^{\mu}}{dt}\frac{dz^{\nu}}{dt}. \tag{9}$$

The trajectory is governed by the equation of motion

$$\frac{d^2z^{\mu}}{dt^2} + U^{-1}\frac{dU}{dt}\frac{dz^{\mu}}{dt} + \Gamma^{\mu}_{\nu\sigma}\frac{dz^{\nu}}{dt}\frac{dz^{\sigma}}{dt} = U^{-2}f^{\mu}, \tag{10}$$

where $\Gamma^{\mu}_{\nu\sigma}$ are the background Schwarzschild Christoffel symbols, and $f^{\mu}$ is the gravitational self-force per unit mass $m_p$. The self-force has only two independent components because $f^{\theta} = 0$ on equatorial orbits and because the normalization $g_{\mu\nu}u^{\mu}u^{\nu} = -1$ implies $u^{\mu}f_{\mu} = 0$. Explicitly, the self-force per unit mass $m_p$ (the self-acceleration) acting on the secondary is given by [36, 37]

$$f^{\mu} = -\frac{1}{2}P^{\mu\nu}(\delta^{\rho}_{\nu} - h^{\mathcal{R}\rho}_{\nu})(2h^{\mathcal{R}}_{\beta\rho;\alpha} - h^{\mathcal{R}}_{\alpha\beta;\rho})u^{\alpha}u^{\beta} + O(\varepsilon^3), \qquad P^{\mu\nu} := g^{\mu\nu} + u^{\mu}u^{\nu}. \tag{11}$$

We have split the metric perturbation $h_{\mu\nu}$ as $h_{\mu\nu} = h^{\mathcal{P}}_{\mu\nu} + h^{\mathcal{R}}_{\mu\nu}$, where $h^{\mathcal{P}}_{\mu\nu}$ is an analytically known puncture and $h^{\mathcal{R}}_{\mu\nu}$ is the residual field as defined in [38]. A semicolon indicates a covariant derivative with respect to the background metric $g_{\mu\nu}$.

Our phase-space formalism here differs from the formulation of the inspiral in the main text of [9] and the formulation of the transition to plunge in [14]. Those references, rather than using three variables $(\Delta)J^a$ to characterize the slowly evolving state of the system, used a single "slow time" variable ($\varepsilon t$ during the inspiral and $\varepsilon^{1/5}(t - t_*)$ during the transition to plunge, where $t_*$ is the time at which the particle reaches the ISCO). The two formulations are formally equivalent, in the sense that the equations in the slow-time formalism can be obtained from those in the phase-space formalism by expanding $(\Delta)J^a$ for small $\varepsilon$ at fixed slow time. We use the phase-space approach due to its better accuracy (see the comparison between the 1PAT1 and 1PAT2 models in [8]) and because it will enable our approach to waveform generation. The phase-space formulation we use here was first presented in appendix A of [9] for the inspiral regime. Reference [4] detailed it for generic inspirals in Kerr spacetime. Here we apply it to the transition to plunge for the first time.

## 2.2 Einstein's field equations

We now introduce the formalism that we use to tackle Einstein's field equations, extending the phase-space approach from [9] to include the transition-to-plunge expansion. The metric perturbation due to the small secondary can be written as

$$h_{\mu\nu}(\varepsilon, s, x^i) = \sum_{n\geq 1}\varepsilon^n h^n_{\mu\nu}((\Delta)J^a(s), \phi_p(s), x^i), \qquad x^i = (r, \theta, \phi), \tag{12}$$

where $s$ is the hyperboloidal time defined in eq. (2). The number $n$ is a natural number in the case of the inspiral expansion, and an integer multiple of $1/5$ in the transition-to-plunge expansion. The reason for these specific non-integer powers will become clear in later sections. In either regime, the integer part $\lfloor n \rfloor$ denotes the level of non-linearity of the perturbation: terms with $\lfloor n \rfloor = 1$ are linear (meaning $h^n_{\mu\nu}$ for $1 \leq n < 2$ are generated by sources that are at most linear in lower-order $h^n_{\mu\nu}$'s); terms with $\lfloor n \rfloor = 2$ are quadratic (meaning $h^n_{\mu\nu}$ for $2 \leq n < 3$ are generated by sources that are at most quadratic in lower-order $h^n_{\mu\nu}$'s); and so on. The level of non-linearity $\lfloor n \rfloor$ in the transition-to-plunge expansion is incremented by 1 every 5 orders in

the expansion. For the inspiral, we will denote $h_{\mu\nu}^n$ with parentheses as $h_{\mu\nu}^{(n)}(J^a, \phi_p, x^i)$, while for the transition to plunge we will denote $h_{\alpha\beta}^n$ with square brackets as $h_{\alpha\beta}^{[5n]}(\Delta J^a, \phi_p, x^i)$. Our convention allows us to label tensors at each order in perturbation theory with an integer superscript. In this section we introduce both cases simultaneously. All the time dependence of the metric perturbation is encoded in $(\Delta)J^a$ and $\phi_p$.

It will be convenient to introduce the $n^{th}$-order trace-reversed metric perturbation

$$\bar{h}_{\mu\nu}^n := h_{\mu\nu}^n - \frac{1}{2} g_{\mu\nu} g^{\alpha\beta} h_{\alpha\beta}^n$$

along with the sum $\bar{h}_{\mu\nu} := \sum_n \varepsilon^n \bar{h}_{\mu\nu}^n$. We note that expansions analogous to eq. (12) also hold for the puncture and residual fields, $h_{\mu\nu}^{\mathcal{P}}$ and $h_{\mu\nu}^{\mathcal{R}}$, such that $h_{\mu\nu} = h_{\mu\nu}^{\mathcal{P}} + h_{\mu\nu}^{\mathcal{R}}$. In the puncture scheme, the secondary is replaced with a singular puncture in the spacetime geometry. The puncture diverges on the worldline and approximates the physical behaviour of the metric near the secondary. At linear order, the puncture scheme is equivalent to considering the secondary as a point particle of mass $m_p$ moving on the worldline $z^\mu$.

It will also be convenient to isolate the metric perturbations' dependence on $\delta M^\pm$. The $n^{th}$-order metric perturbation is a polynomial of order $\lfloor n \rfloor$ in $\delta M$ and $\delta J$, that is, we can decompose it as

$$\lfloor n \rfloor = 1: \qquad h_{\mu\nu}^n((\Delta)J^a, \phi_p, x^i) = h_{\mu\nu}^{n,a}((\Delta)\Omega, \phi_p, x^i)\delta M^a, \tag{13a}$$

$$\lfloor n \rfloor = 2: \qquad h_{\mu\nu}^n((\Delta)J^a, \phi_p, x^i) = h_{\mu\nu}^{n,ab}((\Delta)\Omega, \phi_p, x^i)\delta M^a \delta M^b, \tag{13b}$$

where $\delta M^a := (1, \delta M, \delta J)$ and the repeated indices are summed over. The components that are purely along $\delta M^\pm$ (i.e., $h_{\mu\nu}^{n,\delta M^\pm}$, $h_{\mu\nu}^{n,\delta M^\pm \delta M^\pm}$, and $h_{\mu\nu}^{n,\delta M^\pm \delta M^\mp}$) represent perturbations towards a slowly-evolving Kerr metric with mass $M + \varepsilon\, \delta M$ and spin $\varepsilon\, \delta J$. This means that these components do not depend on the orbital phase $\phi_p$, and after the harmonic decomposition we perform below, they only receive $\ell = 0, 1$, $m = 0$ contributions at the linear level and $\ell = 0, 1, 2$, $m = 0$ at the quadratic level [9].

We now turn to the field equations and their harmonic decomposition. We will perform the multiscale expansion separately for the inspiral and transition-to-plunge regimes in sections 3.2 and 4.2, respectively. We first substitute the metric (1) into the vacuum Einstein equations (which apply at all points off the secondary's worldline) and work in Lorenz gauge, $\nabla^\nu \bar{h}_{\mu\nu} = 0$. The expansion of the field equations in (potentially non-integer) powers of $\varepsilon$, in terms of the coefficients $h_{\mu\nu}^n$, will depend on the regime. Hence, in this section, we focus on the generic structure of the field equations, expressed in terms of powers of the total metric perturbation $h_{\mu\nu}$. Up to terms cubic in $h_{\mu\nu}$ (i.e., neglecting terms of order $\varepsilon^4$) and using $G_{\mu\nu}[g] = 0$, we obtain

$$E_{\mu\nu}[\bar{h}] = 2\delta^2 G_{\mu\nu}[\bar{h}, \bar{h}] + 2\delta^3 G_{\mu\nu}[\bar{h}, \bar{h}, \bar{h}] + O(\varepsilon^4) \tag{14}$$

away from the worldline. Here $E_{\mu\nu}[\bar{h}] := \nabla^\alpha \nabla_\alpha \bar{h}_{\mu\nu} + 2R_\mu{}^\alpha{}_\nu{}^\beta \bar{h}_{\alpha\beta}$ is $-2\delta G_{\mu\nu}$ in the Lorenz gauge, where $\delta G_{\mu\nu}$ is the linearized Einstein tensor. Following the notation of [9], tensors inside square brackets, i.e. tensors that are being operated on, have their indices suppressed. $\delta^2 G_{\mu\nu}$ and $\delta^3 G_{\mu\nu}$ are the quadratic and cubic couplings of linear perturbations, that is, the pieces in the expansion of $G_{\mu\nu}[g+h]$ that are quadratic and cubic in $\bar{h}_{\mu\nu}$. An explicit expression for $\delta^2 G_{\mu\nu}$ can be found in [9]. Equation (14) can be extended to the worldline using a puncture scheme, in which the puncture contribution to $\bar{h}_{\mu\nu}$ is moved to the right-hand side of the field equations and treated as a source [39]:

$$E_{\mu\nu}[\bar{h}^{\mathcal{R}}] = 2\delta^2 G_{\mu\nu}[\bar{h}, \bar{h}] + 2\delta^3 G_{\mu\nu}[\bar{h}, \bar{h}, \bar{h}] - E_{\mu\nu}[\bar{h}^{\mathcal{P}}] + O(\varepsilon^4). \tag{15}$$

At low orders, working in the puncture formulation is equivalent to using a point-particle source,

$$T_{\mu\nu} = m_p \int u_\mu u_\nu \frac{\delta^4(x^\mu - z^\mu(\tau))}{\sqrt{-\det g}} d\tau + O(\varepsilon^2). \tag{16}$$

The form of this source allows us to more easily justify our multiscale ansatz in eq. (28) below. The second-order terms in $T_{\mu\nu}$ are made up of terms proportional to $h_{\mu\nu}^{\mathcal{R}}$ multiplying delta functions supported on the worldline [40]. Although the third-order terms have not been derived, the reasoning in [40] implies that they will be structurally similar. In terms of this $T_{\mu\nu}$, we can rewrite eq. (15) as

$$E_{\mu\nu}[\bar{h}] = -16\pi T_{\mu\nu} + 2\delta^2 G_{\mu\nu}[\bar{h}, \bar{h}] + 2\delta^3 G_{\mu\nu}[\bar{h}, \bar{h}, \bar{h}] + O(\varepsilon^4). \tag{17}$$

We refer to [40] for discussion of the strict interpretation of (and equivalence between) eqs. (15) and (17). We will not explicitly require the $O(\varepsilon^2)$ and higher terms in the stress-energy tensor, and in later sections we will freely move between the puncture formulation and the point-particle stress-energy formulation.

We next decompose the fields into tensor spherical harmonic modes, using the Barack-Lousto-Sago basis of harmonics [41]. We start by decomposing the trace-reversed metric perturbations $\bar{h}_{\mu\nu}^n$ as

$$\bar{h}_{\mu\nu}^n = \sum_{i\ell m} \frac{a_{i\ell}}{r} \bar{h}_{i\ell m}^n ((\Delta)J^a, \phi_p, r) Y_{\mu\nu}^{i\ell m}(r, \theta, \phi), \tag{18}$$

and analogously $\bar{h}_{i\ell m} := \sum_n \varepsilon^n \bar{h}_{i\ell m}^n$, where $i = 1, \dots, 10$, $\ell \geq 0$ and $m = -\ell, \dots, \ell$. A useful property of this basis is that the corresponding expansion of $h_{\mu\nu}$ is identical to eq. (18) but with the $i = 3, 6$ terms exchanged, i.e., $\bar{h}_{3\ell m}^n = h_{6\ell m}^n$ and $\bar{h}_{6\ell m}^n = h_{3\ell m}^n$. The tensor harmonics $Y_{\mu\nu}^{i\ell m}$ and the normalization factors $a_{i\ell}$ are defined in appendix A.1. The harmonic modes $\bar{h}_{i\ell m}^n$ (and similarly the harmonic modes $\Phi_{i\ell m}$ of any symmetric tensor $\Phi_{\mu\nu}$) are computed as

$$\bar{h}_{i\ell m}^n = \frac{r}{a_{i\ell}\kappa_i} \oint dS\, \eta^{\mu\alpha}\eta^{\nu\beta} \bar{h}_{\mu\nu}^n Y_{\alpha\beta}^{i\ell m*}, \tag{19}$$

with $\oint dS = \int_0^{2\pi} d\phi \int_0^\pi d\theta \sin\theta$, $\kappa_i = f^2$ if $i = 3$ and $1$ otherwise, and $\eta^{\mu\nu}$ defined in eq. (A.4) below.

To motivate our ansatz for the tensor-harmonic modes of the metric perturbation, we first decompose the source terms in the Einstein equation (17). Changing the integration variable in eq. (16) to $t$, we can evaluate the integral and obtain

$$T_{\mu\nu} = m_p \frac{u_\mu u_\nu}{U r_p^2} \delta(r - r_p)\delta(\theta - \pi/2)\delta(\phi - \phi_p) + O(\varepsilon^2). \tag{20}$$

Given that $m_p = M\varepsilon$, the harmonic modes of the point-particle stress-energy tensor then read

$$\begin{aligned} T_{i\ell m} &= -\frac{r f(r)}{4 a_{i\ell}\kappa_i} \oint dS\, \eta^{\mu\alpha}\eta^{\nu\beta} T_{\mu\nu} Y_{\alpha\beta}^{i\ell m*}(r, \theta, \phi) + O(\varepsilon^2) \\ &= -\frac{\varepsilon f(r_p) M}{4 a_{i\ell}\kappa_i} \eta^{\mu\alpha}\eta^{\nu\beta} \frac{u_\mu u_\nu}{U r_p} Y_{\alpha\beta}^{i\ell m*}\left(r, \frac{\pi}{2}, 0\right) e^{-im\phi_p}\delta(r - r_p) + O(\varepsilon) \\ &:= \varepsilon\, t_{i\ell m} e^{-im\phi_p}\delta(r - r_p) + O(\varepsilon^2). \end{aligned} \tag{21}$$

Here we have used the analog of eq. (19) with an additional factor $-f(r)/4$ to simplify later expressions. The mode amplitudes $t_{i\ell m}$ are evaluated on the worldline (4). The harmonic

modes of the quadratic Einstein tensor, $\delta^2 G_{i\ell m}$, can be computed from

$$\delta^2 G_{i\ell m}[\bar{h}, \bar{h}] = -\frac{r f(r)}{4 a_{i\ell} \kappa_i} \oint dS\, \eta^{\mu\alpha} \eta^{\nu\beta} \delta^2 G_{\mu\nu}[\bar{h}, \bar{h}] Y_{\alpha\beta}^{i\ell m*}. \tag{22}$$

The metric perturbations appearing in the integral can themselves be decomposed as prescribed by eq. (18). Schematically, we can rewrite the harmonic modes of the quadratic Einstein tensor in terms of the modes of the metric perturbations as

$$\delta^2 G_{i\ell m}[\bar{h}, \bar{h}] = \sum_{i_1 \ell_1 m_1} \sum_{i_2 \ell_2 m_2} \delta^2 G_{i\ell m}^{i_1 \ell_1 m_1, i_2 \ell_2 m_2} \bar{h}_{i_1 \ell_1 m_1} \bar{h}_{i_2 \ell_2 m_2}, \tag{23}$$

where $\delta^2 G_{i\ell m}^{i_1 \ell_1 m_1, i_2 \ell_2 m_2}$ is a bilinear differential operator acting on $\bar{h}_{i_1 \ell_1 m_1}$ and $\bar{h}_{i_2 \ell_2 m_2}$ separately [42]. It is important to notice that $m = m_1 + m_2$ since the integration over $\phi$ in eq. (22) gives

$$\int d\phi\, e^{-im\phi} e^{im_1 \phi} e^{im_2 \phi} \propto \delta_{m, m_1 + m_2}. \tag{24}$$

An equivalent reasoning holds for the cubic Einstein tensor.

After the harmonic decomposition, the field equations (17) are given by a set of coupled partial differential equations for the harmonic modes $\bar{h}_{i\ell m}$ ($i = 1, \ldots, 10$),

$$E_{ij\ell m} \bar{h}_{j\ell m} = -16\pi T_{i\ell m} + 2\delta^2 G_{i\ell m}[\bar{h}, \bar{h}] + 2\delta^3 G_{i\ell m}[\bar{h}, \bar{h}, \bar{h}] + O(\varepsilon^4), \tag{25}$$

with summation over the repeated index $j$ only. The decomposed linear Einstein operator $E_{ij\ell m}$ is given by

$$E_{ij\ell m} = \frac{\delta_{ij}}{4}\left[(\partial_t)_r^2 - (\partial_x)_t^2 + 4V_\ell(r)\right] + \mathcal{M}_r^{ij}(r) + \mathcal{M}_t^{ij}(r)(\partial_t)_r. \tag{26}$$

The potential is $V_\ell(r) := \frac{f}{4}\left[\frac{2M}{r^3} + \frac{\ell(\ell+1)}{r^2}\right]$. The operator matrix $\mathcal{M}^{ij} := \mathcal{M}_r^{ij} + \mathcal{M}_t^{ij}(\partial_t)_r$, with $i, j = 1, \ldots, 10$, couples between modes $\bar{h}_{j\ell m}$ with different $j$ but the same $\ell$ and $m$. The explicit components $\mathcal{M}^{ij}$ are given in appendix A.2. Since $\mathcal{M}^{ij} = 0$ for $i = 1, \ldots, 7$ with $j = 8, 9, 10$ and for $i = 8, 9, 10$ with $j = 1, \ldots, 7$, the field equations at each order in the multiscale expansion will split into seven coupled equations for the even modes ($\bar{h}_{i\ell m}$, $i = 1, \ldots, 7$) and three coupled equations for the odd modes ($\bar{h}_{i\ell m}$, $i = 8, 9, 10$).

When acting on a function of $(\Delta)J^a(s)$, $\phi_p(s)$ and $r$, derivatives with respect to $t$ and $r$ become operators on phase space. The $t$ derivative at fixed radial coordinate $r$, $(\partial_t)_r$, becomes

$$(\partial_t)_r = \Omega \frac{\partial}{\partial \phi_p} + F^{(\Delta)J^a} \frac{\partial}{\partial (\Delta)J^a}, \tag{27a}$$

where we have used eq. (7) for $d(\Delta)J^a/ds$. Likewise, for the radial derivative at fixed $t$, $(\partial_r)_t$,

$$f(\partial_r)_t = (\partial_x)_t = (\partial_x)_{(\Delta)J^a, \phi_p} - H\left(\Omega \frac{\partial}{\partial \phi_p} + F^{(\Delta)J^a} \frac{\partial}{\partial (\Delta)J^a}\right), \tag{27b}$$

where $H$ is defined in eq. (3). Consequently, the linear and nonlinear operators $E_{ij\ell m}$ and $\delta^n G_{i\ell m}$ become operators on phase space. We use this to promote the Einstein equation (decomposed in tensor harmonics) to a partial differential equation in $((\Delta)J^a, \phi_p, r)$ rather than $(s, r)$, treating $((\Delta)J^a, \phi_p)$ as independent coordinates. The solution on phase space becomes a solution on spacetime when evaluated on a physical trajectory $((\Delta)J^a(s), \phi_p(s))$ that satisfies eqs. (5) and (7).

Since the source (21) has a $2\pi/m$ periodicity in $\phi_p$, we adopt the following ansatz for the metric perturbations:

$$\bar{h}^n_{i\ell m}((\Delta)J^a,\phi_p,r) = R^n_{i\ell m}((\Delta)J^a,r)e^{-im\phi_p}.\tag{28}$$

In summary, the trace-reversed metric perturbation is therefore decomposed as

$$\bar{h}_{\mu\nu}(\varepsilon,s,x^i) = \sum_{n\geq 1}\varepsilon^n \sum_{i\ell m}\frac{a_{i\ell}}{r}R^n_{i\ell m}((\Delta)J^a(\varepsilon,s),r)e^{-im\phi_p(\varepsilon,s)}Y^{i\ell m}_{\mu\nu}(r,\theta,\phi),\tag{29}$$

with analogous expansions for the puncture and residual fields. The metric perturbation is likewise expanded as $h_{\mu\nu}=\sum_n \varepsilon^n \sum_{i\ell m}\frac{a_{i\ell}}{r}R^n_{i\ell m}((\Delta)J^a,r)e^{-im\phi_p}Y^{i\ell m}_{\mu\nu}(r,\theta,\phi)$, where $\underline{3}=6$, $\underline{6}=3$ and $\underline{i}=i$ otherwise. Note that $a_{i\ell}=a_{\underline{i}\ell}$. Following the discussion above eq. (24), we can write

$$\delta^2 G_{i\ell m}\big[\bar{h}_{i_1\ell_1 m_1},\bar{h}_{i_2\ell_2 m_2}\big] = \delta^2 G_{i\ell m}\big[R_{i_1\ell_1 m_1},R_{i_2\ell_2 m_2}\big]e^{-im\phi_p},\tag{30}$$

and similarly for $\delta^3 G_{i\ell m}$. Hence, at all orders, the sources and solutions only depend on $\phi_p$ through the exponential $e^{-im\phi_p}$, which we can then factor out of the equations.

Finally, the harmonic decomposition of the field equations (14) reads

$$\widehat{E}_{ij\ell m}R_{j\ell m} = 2\widehat{\delta^2 G}_{i\ell m}[R,R] + 2\widehat{\delta^3 G}_{i\ell m}[R,R,R] + O(\varepsilon^4),\tag{31}$$

where $R_{i\ell m}:=\sum_n \varepsilon^n R^n_{i\ell m}$. The operators $\widehat{E}_{ij\ell m}$ and $\widehat{\delta^n G}_{i\ell m}$ are given by $E_{ij\ell m}$ and $\delta^n G_{i\ell m}$ with the prescription (27) for $t$ and $r$ derivatives and the further replacement of $\phi_p$ derivatives with $\partial_{\phi_p}\to -im$. Equation (31) is complete once we include the Lorenz gauge condition $Z_\mu := \nabla^\nu \bar{h}_{\mu\nu}=0$. Substituting eq. (29) and taking the derivatives as prescribed by eq. (27), we obtain the harmonic decomposition of the Lorenz gauge condition,

$$\left[Z_{raj}(r)+Z_{taj}(r)\left(-im\Omega+\frac{d(\Delta)J^b}{dt}\frac{\partial}{\partial(\Delta)J^b}\right)\right]R_{j\ell m}=0,\tag{32}$$

with $a=1,2,3,4$ and where $Z_{raj}$ are operators that contain $(\partial_x)_{(\Delta)J^a}$ and $Z_{taj}(r)$ is a radial vector, which are given explicitly in appendix A.3.

# 3 Quasi-circular inspiral

As mentioned in section 1, two disparate timescales characterize the quasi-circular inspiral: the phase $\phi_p$ evolves on the orbital timescale $\sim M$, while the mechanical parameters $J^a=(\Omega,\delta M,\delta J)=(\Omega,\delta M^\pm)$ evolve on the radiation-reaction timescale $\sim M/\varepsilon$. In order to reflect this behaviour, we perform an *inspiral expansion* of all orbital quantities, in integer powers of the mass ratio $\varepsilon$ at fixed mechanical parameters $J^a$. The multiscale nature of this expansion will become evident in section 3.2. Explicitly, we expand the orbital radius and redshift as

$$r_p(\varepsilon,J^a) = r_{(0)}(\Omega) + \sum_{n=1}^{\infty}\varepsilon^n r_{(n)}(J^a),\tag{33}$$

$$U(\varepsilon,J^a) = U_{(0)}(\Omega) + \sum_{n=1}^{\infty}\varepsilon^n U_{(n)}(J^a).\tag{34}$$

Terms labelled with a subscript $(n)$ in parentheses appear at order $\varepsilon^n$ with $n=0,1,2,\ldots$. The leading-order term in the inspiral expansion is known as the adiabatic or the zeroth post-adiabatic (0PA) order. As we will show below, the adiabatic order only depends on $\Omega$ and not

on $\delta M$ and $\delta J$. The $n^{th}$ subleading term is called the $n^{th}$ post-adiabatic or $n$PA order and depends on the full set of mechanical parameters. Since we are expanding all functions at fixed $J^a$, we also expand the rates of change $dJ^a/dt$ as

$$\frac{1}{\varepsilon}\frac{dJ^a}{dt}(\varepsilon, J^a) = F_{(0)}^{J^a}(\Omega) + \sum_{n=1}^{\infty} \varepsilon^n F_{(n)}^{J^a}(J^b). \tag{35}$$

The factor of $\varepsilon^{-1}$ appearing on the left-hand side reflects the fact that the evolution of the mechanical parameters takes place over the radiation-reaction timescale. The slow time $\tilde{t} = \varepsilon t$ could be introduced to absorb this factor, but we opt to keep a lighter notation.

The inspiral expansion of the self-force reads

$$f^\mu(\varepsilon, J^a) = \varepsilon f_{(1)}^\mu(\Omega) + \sum_{n=2}^{\infty} \varepsilon^n f_{(n)}^\mu(J^a), \quad \mu = t, \phi, \tag{36a}$$

$$f^r(\varepsilon, J^a) = \varepsilon f_{(1)}^r(J^a) + \sum_{n=2}^{\infty} \varepsilon^n f_{(n)}^r(J^a). \tag{36b}$$

Consistently with equatorial motion, we have $f^\theta = 0$. In the quasi-circular case, the split of the self-force into dissipative and conservative pieces is straightforward: the dissipative self-force is antisymmetric under time reversal $(t, \phi) \rightarrow (-t, -\phi)$ and is therefore given by $f_{\text{diss}}^\mu = (f^t, 0, 0, f^\phi)$. The conservative piece is then $f_{\text{cons}}^\mu = (0, f^r, 0, 0)$, which is symmetric under time reversal. In section 3.3 we perform the inspiral expansion of eq. (11) and obtain explicit expressions for the self-force in terms of the metric perturbations. This allows us to show that, as anticipated in eq. (36a), the dissipative first-order self-force only depends on $\Omega$.

Adiabatic inspiral waveforms have been computed since the seminal work by Poisson and collaborators in 1993 [43–45], while 1PA inspiral waveforms were only obtained in 2021 [8]. Higher-order $n$PA waveforms will not be required for detection or parameter estimation for LISA EMRI sources [46]. Though they would be useful for mass ratios closer to unity [18], higher-order $n$PA waveforms are unlikely to be obtained in the near future, and key theoretical ingredients, such as the third-order puncture and third-order self-force, have not yet been derived. However, since the matching procedure between two asymptotically expanded series mixes the perturbative orders, we derive the behaviour of the 2PA approximation for a better understanding of the asymptotic match between the inspiral and the transition to plunge. For this purpose, it is sufficient to obtain the structure of the third-order self-force without the need for its explicit expression.

## 3.1 Orbital motion at 0PA, 1PA and 2PA order

We perform the inspiral expansion of the worldline (4) and the four-velocity (8), and substitute them into the normalization condition (9) and the equation of motion (10). At each order $\varepsilon^n$, $n \geq 0$, we obtain algebraic equations for $U_{(n)}$ and $r_{(n)}$ from the normalization condition and the radial component of the equation of motion, respectively. We obtain the forcing terms $F_{(n)}^\Omega$ from the time component of the equation of motion at order $\varepsilon^{n+1}$. The forcing terms $F_{(n)}^{\delta M}$ and $F_{(n)}^{\delta J}$ can be determined from the GW fluxes of energy and angular momentum through the horizon of the primary. For the purpose of this paper we are only interested in their structure, which we derive in section 3.2.

The 0PA and 1PA quantities were given in [9] and are repeated here for completeness. At adiabatic order we obtain

$$r_{(0)} = \frac{M}{(M\Omega)^{2/3}}, \qquad U_{(0)} = \frac{1}{\sqrt{1 - 3(M\Omega)^{2/3}}}, \qquad F_{(0)}^\Omega = -\frac{3\Omega f_{(0)}}{(M\Omega)^{2/3} U_{(0)}^4 D} f_{(1)}^t, \tag{37}$$

where we have defined

$$D := 1 - 6(M\Omega)^{2/3}, \qquad f_{(0)} := 1 - \frac{2M}{r_{(0)}} = 1 - 2(M\Omega)^{2/3}. \tag{38}$$

The adiabatic motion is driven by the dissipative first-order self-force only. Since $f_{(1)}^t$ does not depend on $\delta M$ and $\delta J$ (see eq. (62) below), the adiabatic motion is only determined by the orbital frequency $\Omega$. The 1PA quantities read

$$r_{(1)} = -\frac{f_{(1)}^r}{3\Omega^2 U_{(0)}^2 f_{(0)}}, \qquad U_{(1)} = 0, \tag{39a}$$

$$
\begin{aligned}
F_{(1)}^\Omega = -\frac{3\Omega f_{(0)}}{(M\Omega)^{2/3} U_{(0)}^4 D} f_{(2)}^t - \frac{4(1 - 6(M\Omega)^{2/3} + 12(M\Omega)^{4/3})}{(M\Omega) U_{(0)}^6 f_{(0)} D^2} f_{(1)}^t f_{(1)}^r \\
- \frac{2}{(M\Omega)^{1/3} U_{(0)}^4 f_{(0)} D} F_{(0)}^{J^a} \partial_{J^a} f_{(1)}^r.
\end{aligned} \tag{39b}
$$

Corrections to the orbital radius depend on the first-order radial self-force. At 1PA order, the slow evolution of the orbital frequency is driven by the full first-order and the dissipative second-order self-force. Given the structure of the self-force presented in section 3.3 below, both $r_{(1)}$ and $F_{(1)}^\Omega$ are linear in $\delta M^a = (1, \delta M, \delta J)$ and can be decomposed as $r_{(1)} = r_{(1)}^a \delta M^a$ and $F_{(1)}^\Omega = F_{(1)}^{\Omega a} \delta M^a$. Finally, the 2PA quantities are given by

$$
\begin{aligned}
r_{(2)} = &-\frac{1}{3\Omega^2 U_{(0)}^2 f_{(0)}} f_{(2)}^r + \frac{2\left(4 - 45(M\Omega)^{2/3} + 114(M\Omega)^{4/3} - 72(M\Omega)^2\right)}{3\Omega^3 (M\Omega) U_{(0)}^6 D^3} \left(f_{(1)}^t\right)^2 \\
&+ \frac{\left(1 - 4(M\Omega)^{2/3}\right)}{9\Omega^3 (M\Omega)^{1/3} U_{(0)}^4 f_{(0)}^3} \left(f_{(1)}^r\right)^2 - \frac{2 f_{(0)}}{\Omega^2 (M\Omega) U_{(0)}^8 D^2} f_{(1)}^t \partial_\Omega f_{(1)}^t,
\end{aligned} \tag{40a}
$$

$$U_{(2)} = \frac{2 f_{(0)}}{\Omega^2 (M\Omega)^{2/3} U_{(0)}^5 D^2} \left(f_{(1)}^t\right)^2 + \frac{1}{6\Omega^2 U_{(0)} f_{(0)}^2} \left(f_{(1)}^r\right)^2, \tag{40b}$$

$$
\begin{aligned}
F_{(2)}^\Omega = &-\frac{3\Omega f_{(0)}}{(M\Omega)^{2/3} U_{(0)}^4 D} f_{(3)}^t \\
&+ \frac{4\left(1 - 6(M\Omega)^{2/3} + 12(M\Omega)^{4/3}\right)}{3\Omega (M\Omega)^{1/3} U_{(0)}^2 f_{(0)}^2 D} \left(F_{(0)}^\Omega f_{(2)}^r + F_{(1)}^\Omega f_{(1)}^r\right) \\
&+ \frac{8}{\Omega (M\Omega)^2 U_{(0)}^{10} D^6} \left(22 - 481(M\Omega)^{2/3} + 3909(M\Omega)^{4/3} - 14610(M\Omega)^2\right. \\
&\left. + 26784(M\Omega)^{8/3} - 22680(M\Omega)^{10/3} + 6480(M\Omega)^4\right) \left(f_{(1)}^t\right)^3 \\
&- \frac{12 f_{(0)} \left(16 - 175(M\Omega)^{2/3} - 252(M\Omega)^2 + 420(M\Omega)^{4/3}\right)}{(M\Omega)^2 U_{(0)}^{12} D^5} \left(f_{(1)}^t\right)^2 \partial_\Omega f_{(1)}^t \\
&+ \frac{36\Omega f_{(0)}^2}{(M\Omega)^2 U_{(0)}^{14} D^4} \left[f_{(1)}^t \left(\partial_\Omega f_{(1)}^t\right)^2 + \left(f_{(1)}^t\right)^2 \partial_\Omega^2 f_{(1)}^t\right] \\
&+ \frac{\left(3 - 26(M\Omega)^{2/3} - 120(M\Omega)^2 + 84(M\Omega)^{4/3}\right)}{\Omega (M\Omega)^{4/3} U_{(0)}^8 f_{(0)}^3 D^2} f_{(1)}^t \left(f_{(1)}^r\right)^2
\end{aligned} \tag{40c}
$$

$$- \frac{2}{\left((M\Omega)^{1/3} - 8(M\Omega) + 12(M\Omega)^{5/3}\right)U_{(0)}^4} \left(F_{(0)}^{J^a}\partial_{J_a}f_{(2)}^r + F_{(1)}^{J^a}\partial_{J_a}f_{(1)}^r\right)$$

$$+ \frac{\left(1 - 4(M\Omega)^{2/3}\right)^2}{\Omega(M\Omega)^{2/3}U_{(0)}^4 f_{(0)}^3 D} f_{(1)}^r F_{(0)}^{J^a}\partial_{J_a}f_{(1)}^r.$$

The third-order dissipative self-force begins to appear at 2PA order, alongside the full first- and second-order self-forces. The 2PA quantities $r_{(2)}$, $U_{(2)}$ and $F_{(2)}^{\Omega}$ are quadratic in $\delta M^a$.

The quantity $D$ defined in eq. (38) vanishes at the ISCO, and inverse powers of it in the above expressions indicate how rapidly a term in the inspiral expansion diverges as the inspiral approaches the ISCO.

## 3.2 Einstein's field equations at 0PA, 1PA and 2PA order

We now consider the field equations in the inspiral regime. The expansion (29) of the trace-reversed metric perturbation is

$$\bar{h}_{\mu\nu}(\varepsilon, s, x^i) = \sum_{n=1}^{\infty}\varepsilon^n \sum_{i\ell m} \frac{a_{i\ell}}{r} R_{i\ell m}^{(n)}(J^a(\varepsilon,s), r)e^{-im\phi_p(\varepsilon,s)}Y_{\mu\nu}^{i\ell m}(r, \theta, \phi), \tag{41}$$

where $n$ takes integer values. By factoring out the rapidly oscillating phases $e^{-im\phi_p}$, we factor out the orbital "fast-time" dynamics from the field equations. The Einstein equation (31) will consequently reduce to a sequence of radial ordinary differential equations for the slowly evolving mode amplitudes $R_{i\ell m}^{(n)}$ (see eq. (52) below).

Recalling eq. (35), we start by performing the inspiral expansion of eq. (27). We obtain

$$\widehat{(\partial_t)}_r = -im\Omega + \varepsilon \sum_{n=0}^{\infty}\varepsilon^n F_{(n)}^{J^a}\partial_{J^a}, \tag{42a}$$

$$\widehat{(\partial_x)}_t = (\partial_x)_{J^a} + imH\Omega - \varepsilon H \sum_{n=0}^{\infty}\varepsilon^n F_{(n)}^{J^a}\partial_{J^a}, \tag{42b}$$

where we have again used a hat to denote operators on functions of $J^a$ and $r$, for which $\partial_{\phi_p} \to -im$. The linearized Einstein operator (26) is then expanded as

$$\widehat{E}_{ij\ell m}(\varepsilon, J^a, r) = E_{ij\ell m}^{(0)}(\Omega, r) + \varepsilon E_{ij\ell m}^{(1)}(\Omega, r) + \varepsilon^2 E_{ij\ell m}^{(2)}(J^a, r) + O(\varepsilon^3), \tag{43}$$

where

$$E_{ij\ell m}^{(0)} = -\frac{\delta_{ij}}{4}\left[\partial_x^2 + im\Omega(\partial_x H + 2H\partial_x) + m^2\Omega^2\left(1 - H^2\right) - 4V_\ell\right] \\ + \mathcal{M}_r^{ij} - im\Omega\mathcal{M}_t^{ij}, \tag{44a}$$

$$E_{ij\ell m}^{(1)} = \frac{\delta_{ij}}{4}\left[\left(\partial_x H + 2imH^2\Omega - 2im\Omega\right)F_{(0)}^{J^a}\partial_{J^a} + 2HF_{(0)}^{J^a}\partial_{J^a}\partial_x \\ -im\left(1 - H^2\right)F_{(0)}^{\Omega}\right] + \mathcal{M}_t^{ij}F_{(0)}^{J^a}\partial_{J^a} \\ := E_{ij\ell m}^{(1)A} + F_{(0)}^{\Omega}E_{ij\ell m}^{(1)B}, \tag{44b}$$

$$E_{ij\ell m}^{(2)} = \frac{\delta_{ij}}{4}\left[\left(\partial_x H + 2imH^2\Omega - 2im\Omega\right)F_{(1)}^{J^a}\partial_{J^a} + 2HF_{(1)}^{J^a}\partial_{J^a}\partial_x - im\left(1 - H^2\right)F_{(1)}^{\Omega} \\ +(1 - H^2)F_{(0)}^{\Omega}\partial_{\Omega}F_{(0)}^{J^a}\partial_{J^a} + (1 - H^2)F_{(0)}^{J^a}F_{(0)}^{J^b}\partial_{J^a}\partial_{J^b}\right] + \mathcal{M}_t^{ij}F_{(1)}^{J^a}\partial_{J^a} \\ := E_{ij\ell m}^{(2)A} + F_{(0)}^{\Omega}E_{ij\ell m}^{(2)B} + F_{(1)}^{\Omega}E_{ij\ell m}^{(2)C} + \left(F_{(0)}^{\Omega}\right)^2 E_{ij\ell m}^{(2)D} + F_{(0)}^{\Omega}\partial_{\Omega}F_{(0)}^{\Omega}E_{ij\ell m}^{(2)E}. \tag{44c}$$

In the inspiral expansion we use capital Latin letters $A, B, \ldots$ to denote the components of an expression according to the decomposition in polynomials of $F^\Omega_{(n)}$, $n \geq 0$, and their $J^a$ derivatives, which are the only terms that diverge at the ISCO. Note that the forcing terms $F^{\delta M}_{(n)}$ and $F^{\delta J}_{(n)}$, $n \geq 0$, have themselves been decomposed in an analogous way (see eq. (55) below).

We similarly expand the source terms in the field equation (25). We can formally expand the harmonic mode amplitudes $t_{i\ell m}$ of the first-order point-particle stress-energy tensor (21) as a functional of the orbital radius $r_p$ and its rate of change $\dot{r}_p := dr_p/dt$,

$$\varepsilon\, t_{i\ell m}\left[r_p(\varepsilon, J^a), \dot{r}_p(\varepsilon, J^a)\right] = \varepsilon\, t^{(1)}_{i\ell m}(\Omega) + \varepsilon^2 t^{(2)}_{i\ell m}(J^a) + \varepsilon^3 t^{(3)}_{i\ell m}(J^a) + O(\varepsilon^3), \tag{45}$$

where

$$t^{(1)}_{i\ell m} = t_{i\ell m}\big|_{(0)}\,, \tag{46a}$$

$$t^{(2)}_{i\ell m} = r_{(1)} \frac{\partial t_{i\ell m}}{\partial r_p}\bigg|_{(0)} + F^\Omega_{(0)} \partial_\Omega r_{(0)} \frac{\partial t_{i\ell m}}{\partial \dot{r}_p}\bigg|_{(0)} := t^{(2)A}_{i\ell m} + F^\Omega_{(0)} t^{(2)B}_{i\ell m}\,, \tag{46b}$$

$$
\begin{aligned}
t^{(3)}_{i\ell m} &= r_{(2)} \frac{\partial t_{i\ell m}}{\partial r_p}\bigg|_{(0)} + \frac{r^2_{(1)}}{2} \frac{\partial^2 t_{i\ell m}}{\partial r^2_p}\bigg|_{(0)} + F^\Omega_{(1)} \partial_\Omega r_{(0)} \frac{\partial t_{i\ell m}}{\partial \dot{r}_p}\bigg|_{(0)} + F^{J^a}_{(0)} \partial_{J^a} r_{(1)} \frac{\partial t_{i\ell m}}{\partial \dot{r}_p}\bigg|_{(0)} \\
&\quad + \frac{1}{2}\left(F^\Omega_{(0)} \partial_\Omega r_{(0)}\right)^2 \frac{\partial^2 t_{i\ell m}}{\partial \dot{r}^2_p}\bigg|_{(0)} + r_{(1)} F^\Omega_{(0)} \partial_\Omega r_{(0)} \frac{\partial^2 t_{i\ell m}}{\partial r_p \partial \dot{r}_p}\bigg|_{(0)} \\
&:= t^{(3)A}_{i\ell m} + F^\Omega_{(0)} t^{(3)B}_{i\ell m} + F^\Omega_{(1)} t^{(3)C}_{i\ell m} + \left(F^\Omega_{(0)}\right)^2 t^{(3)D}_{i\ell m} + F^\Omega_{(0)} \partial_\Omega F^\Omega_{(0)} t^{(3)E}_{i\ell m}\,.
\end{aligned}
\tag{46c}
$$

We use the notation $t_{i\ell m}\big|_{(0)} := t_{i\ell m}(r_{(0)}, 0)$. We can compute these terms explicitly by substituting eqs. (8), (33), (34) and (35) into the definition (21). In the notation of [9, 41, 47], the leading-order modes $t^{(1)}_{i\ell m}$ are given by

$$t^{(1)}_{i\ell m} = -\frac{1}{4}\mathcal{E}_{(0)} \alpha^{(1)}_{i\ell m} \begin{cases} Y^*_{\ell m}(\tfrac{\pi}{2}, 0) & i = 1, \ldots, 7, \\ \partial_\theta Y^*_{\ell m}(\tfrac{\pi}{2}, 0) & i = 8, 9, 10. \end{cases} \tag{47}$$

Here $\mathcal{E}_{(0)} = M f_{(0)} U_{(0)}$ and the coefficients $\alpha^{(1)}_{i\ell m}$ are given by (dropping the $\ell$ and $m$ indices)

$$\alpha^{(1)}_1 = \frac{f^2_{(0)}}{r_{(0)}}, \quad \alpha^{(1)}_{2,5,9} = 0, \quad \alpha^{(1)}_3 = \frac{f_{(0)}}{r_{(0)}}, \quad \alpha^{(1)}_4 = 2im f_{(0)} \Omega, \quad \alpha^{(1)}_6 = r_{(0)} \Omega^2,$$

$$\alpha^{(1)}_7 = \left[\ell(\ell+1) - 2m^2\right] r_{(0)} \Omega^2, \quad \alpha^{(1)}_8 = 2f_{(0)} \Omega, \quad \alpha^{(1)}_{10} = 2im\Omega^2 r_{(0)}. \tag{48}$$

We will use punctures rather than stress-energy terms when writing down the field equations for $n > 1$, meaning $t^{(2)}_{i\ell m}$ and $t^{(3)}_{i\ell m}$ will not be explicitly needed. However, for completeness, the 1PA modes $t^{(2)}_{i\ell m}$ are given in appendix B.1. We next perform the inspiral expansion of harmonic modes of the quadratic and the cubic Einstein tensor using eq. (41). Up to order $\varepsilon^3$, we are interested in the structure of the following terms:

$$
\begin{aligned}
\widehat{\delta^2 G}_{i\ell m}[R^{(1)}, R^{(1)}] &= \delta^2 G^{(0)}_{i\ell m}[R^{(1)}, R^{(1)}] + \varepsilon\Big(\delta^2 G^{(1)A}_{i\ell m}[R^{(1)}, R^{(1)}] \\
&\quad + F^\Omega_{(0)} \delta^2 G^{(1)B}_{i\ell m}[R^{(1)}, R^{(1)}]\Big) + O(\varepsilon^2),
\end{aligned}
\tag{49}
$$

$$\widehat{\delta^2 G}_{i\ell m}[R^{(1)}, R^{(2)}] = \delta^2 G^{(0)}_{i\ell m}[R^{(1)}, R^{(2)}] + O(\varepsilon), \tag{50}$$

$$\widehat{\delta^3 G}_{i\ell m}[R^{(1)}, R^{(1)}, R^{(1)}] = \delta^3 G^{(0)}_{i\ell m}[R^{(1)}, R^{(1)}, R^{(1)}] + O(\varepsilon). \tag{51}$$

The $O(\varepsilon)$ terms originate from the $O(\varepsilon)$ terms that appear when taking $t$ and $r$ derivatives of the metric perturbations appearing in the quadratic and cubic Einstein tensors as prescribed by eq. (42).

Finally, we substitute these expansions along with $R_{i\ell m} = \sum_n \varepsilon^n R^{(n)}_{i\ell m}$ into the field equations (31). The result reads

$$E^{(0)}_{ij\ell m} R^{(1)}_{j\ell m} = -16\pi\, t^{(1)}_{i\ell m} \delta(r - r_{(0)}), \tag{52a}$$

$$E^{(0)}_{ij\ell m} R^{(2)\mathcal{R}}_{j\ell m} = 2\delta^2 G^{(0)}_{i\ell m}[R^{(1)}, R^{(1)}] - E^{(0)}_{ij\ell m} R^{(2)\mathcal{P}}_{j\ell m} - E^{(1)}_{ij\ell m} R^{(1)}_{j\ell m}, \tag{52b}$$

$$E^{(0)}_{ij\ell m} R^{(3)\mathcal{R}}_{j\ell m} = 2\delta^3 G^{(0)}_{i\ell m}[R^{(1)}, R^{(1)}, R^{(1)}] + 4\delta^2 G^{(0)}_{i\ell m}[R^{(1)}, R^{(2)}]$$

$$+ 2\delta^2 G^{(1)A}_{i\ell m}[R^{(1)}, R^{(1)}] + 2F^{\Omega}_{(0)}\delta^2 G^{(1)B}_{i\ell m}[R^{(1)}, R^{(1)}] \tag{52c}$$

$$- E^{(0)}_{ij\ell m} R^{(3)\mathcal{P}}_{j\ell m} - E^{(1)}_{ij\ell m} R^{(2)}_{j\ell m} - E^{(2)}_{ij\ell m} R^{(1)}_{j\ell m}.$$

As mentioned above, we have written the field equations for $n > 1$ in terms of the residual field, using punctures $R^{(2)\mathcal{P}}_{i\ell m}$ and $R^{(3)\mathcal{P}}_{i\ell m}$ rather than $t^{(2)}_{i\ell m}$ and $t^{(3)}_{i\ell m}$. Also as alluded to previously, the field equations have been reduced to ordinary differential equations in $r$. This is a consequence of the fact that derivatives with respect to $J^a$ are accompanied by forcing terms $F^{J^a}_{(n)}$, which are suppressed by powers of $\varepsilon$ by virtue of eq. (42). Such derivatives therefore become sources rather than appearing on the left-hand side of the field equations.

Each of the equations (52) represents 10 coupled radial ordinary differential equations for each value of $\ell$ and $m$. Several properties that reduce the level of coupling are summarized in section V.E of [9]. We do not report them here since they are not of major interest to our analysis. Equation (52a) does not contain terms that are singular at the ISCO frequency and is therefore solved by a function that is smooth in $\Omega$, $R^{(1)}_{i\ell m}(J^a, r)$. At second order, substituting eq. (44b) into eq. (52b), we can separate the terms proportional to $F^{\Omega}_{(0)}$ from those that are not. Accordingly, we write the second-order puncture and residual fields as

$$R^{(2)}_{i\ell m}(J^a, r) = R^{(2)A}_{i\ell m}(J^a, r) + F^{\Omega}_{(0)} R^{(2)B}_{i\ell m}(J^a, r), \tag{53}$$

where $R^{(2)A}_{i\ell m}$ and $R^{(2)B}_{i\ell m}$ are smooth functions of the orbital frequency. This allows us to split the field equations (52b) as follows:

$$E^{(0)}_{ij\ell m} R^{(2)\mathcal{R}A}_{j\ell m} = 2\delta^2 G^{(0)}_{i\ell m}[R^{(1)}, R^{(1)}] - E^{(0)}_{ij\ell m} R^{(2)\mathcal{P}A}_{j\ell m} - E^{(1)A}_{ij\ell m} R^{(1)}_{j\ell m}, \tag{54a}$$

$$E^{(0)}_{ij\ell m} R^{(2)\mathcal{R}B}_{j\ell m} = -E^{(0)}_{ij\ell m} R^{(2)\mathcal{P}B}_{j\ell m} - E^{(1)B}_{ij\ell m} R^{(1)}_{j\ell m}. \tag{54b}$$

In addition to the puncture and the first-order field $R^{(1)\mathcal{R}}_{i\ell m}$, the second-order mode amplitudes $R^{(2)\mathcal{R}B}_{i\ell m}$ are also sourced by $\partial_{\Omega} R^{(1)\mathcal{R}}_{i\ell m}$. Note that $R^{(2)\mathcal{R}B}_{i\ell m}$ is not sourced by the quadratic term $\delta^2 G^{(0)}_{i\ell m}$ even though it is a second-order perturbation. Therefore, recalling the decomposition in eq. (13), we can deduce from eq. (54) that the fields $R^{(2)\mathcal{R}A}_{i\ell m}$ and $R^{(2)\mathcal{R}B}_{i\ell m}$ are quadratic and linear in $\delta M^a = (1, \delta M, \delta J)$, respectively.

The forcing terms $F^{\delta M}_{(n)}$ and $F^{\delta J}_{(n)}$, $n \geq 0$, which drive the evolution of the corrections to the background mass and spin, are computed from the GW fluxes of energy and angular momentum into the primary. Since the fluxes are proportional to the square of the time derivative of the metric perturbation, the structure of the forcing terms directly follows from the ones of

$R_{i\ell m}^{(1)}$ and $R_{i\ell m}^{(2)}$ that we have just derived. While $F_{(0)}^{\delta M}$ and $F_{(0)}^{\delta J}$ are functions of $\Omega$ only and are smooth at the ISCO, $F_{(1)}^{\delta M}$ and $F_{(1)}^{\delta J}$ display the following structure:

$$F_{(1)}^{\delta M^\pm}(J^a) = F_{(1)A}^{\delta M^\pm}(J^a) + F_{(0)}^\Omega F_{(1)B}^{\delta M^\pm}(J^a). \tag{55}$$

From the right-hand side of eq. (52c) and using eqs. (44b), (44c) and (53), we can finally infer the structure of the third-order metric perturbation mode amplitudes:

$$
\begin{aligned}
R_{i\ell m}^{(3)}(J^a, r) = {}& R_{i\ell m}^{(3)A}(J^a, r) + F_{(0)}^\Omega R_{i\ell m}^{(3)B}(J^a, r) + F_{(1)}^\Omega R_{i\ell m}^{(3)C}(J^a, r) \\
& + \left(F_{(0)}^\Omega\right)^2 R_{i\ell m}^{(3)D}(J^a, r) + F_{(0)}^\Omega \partial_\Omega F_{(0)}^\Omega R_{i\ell m}^{(3)E}(J^a, r).
\end{aligned}
\tag{56}
$$

The cubic term $\delta^3 G_{i\ell m}^{(0)}$ in eq. (52c) only sources $R_{i\ell m}^{(3)A}$, while the quadratic terms $\delta^2 G_{i\ell m}^{(n)}$ only source $R_{i\ell m}^{(3)A}$ and $R_{i\ell m}^{(3)B}$.

### 3.3 Self-force

We now obtain explicit expressions for the self-force in terms of the metric perturbations. After using eqs. (33), (34) and (35) in eqs. (4) and (8), we arrive at inspiral expansions of the worldline and the four-velocity to 1PA order:

$$z^\mu = z_{(0)}^\mu + \varepsilon z_{(1)}^\mu + O(\varepsilon^2) = \left(t, r_{(0)}, \frac{\pi}{2}, \phi_p\right) + \varepsilon\left(0, r_{(1)}, 0, 0\right) + O(\varepsilon^2), \tag{57}$$

$$u^\mu = u_{(0)}^\mu + \varepsilon u_{(1)}^\mu + O(\varepsilon^2) = \left(U_{(0)}, 0, 0, U_{(0)}\Omega\right) + \varepsilon\left(0, U_{(0)}F_{(0)}^\Omega \partial_\Omega r_{(0)}, 0, 0\right) + O(\varepsilon^2), \tag{58}$$

where we have already used the fact that $U_{(1)} = 0$. Substituting these expansions together with the inspiral expansion of the residual piece of the metric perturbation,

$$h_{\mu\nu}^{\mathcal{R}} = \sum_{n=1}^\infty \varepsilon^n \sum_{i\ell m} \frac{a_{i\ell}}{r} R_{i\ell m}^{(n)\mathcal{R}}(J^a, r) e^{-im\phi_p} Y_{\mu\nu}^{i\ell m}(r, \theta, \phi), \tag{59}$$

into eq. (11), we find that the first- and second-order self-forces have the following form:

$$f_{(1)}^\mu = f_{(1)}^\mu(\Omega), \quad \mu = t, \phi, \qquad f_{(1)}^r = f_{(1)}^r(J^a), \tag{60a}$$

$$f_{(2)}^\mu = f_{(2)A}^\mu(J^a) + F_{(0)}^\Omega f_{(2)B}^\mu(J^a). \tag{60b}$$

The terms appearing in these expressions are explicitly given by

$$f_{(1)}^\mu = \frac{1}{2} g^{\mu\nu} h_{u_{(0)}u_{(0)},\nu}^{(1)\mathcal{R}}, \tag{61a}$$

$$
\begin{aligned}
f_{(2)A}^\mu = {}& \frac{1}{2} g^{\mu\nu} h_{u_{(0)}u_{(0)},\nu}^{(2)\mathcal{R}A} + \frac{1}{2} r_{(1)}\left[\partial_r g^{\mu\nu} h_{u_{(0)}u_{(0)},\nu}^{(1)\mathcal{R}} + g^{\mu\nu} h_{u_{(0)}u_{(0)},\nu r}^{(1)\mathcal{R}}\right] \\
& - P_{(0)}^{\mu\nu}\left[3 U_{(0)}^2 \Omega^2 f_{(0)} r_{(1)} \delta_r^\sigma h_{\sigma\nu}^{(1)\mathcal{R}} + \frac{1}{2} h_\nu^{(1)\mathcal{R}\rho} h_{u_{(0)}u_{(0)},\rho}^{(1)\mathcal{R}}\right] \\
& - F_{(0)}^{\delta M^\pm}\left[P_{(0)}^{\mu\nu} U_{(0)} u_{(0)}^\beta h_{\beta\nu,\delta M^\pm}^{(1)\mathcal{R}} - \frac{1}{2} u_{(0)}^\mu U_{(0)} h_{u_{(0)}u_{(0)},\delta M^\pm}^{(1)\mathcal{R}} - \frac{\delta_t^\mu}{2} g^{tt} h_{u_{(0)}u_{(0)},\delta M^\pm}^{(1)\mathcal{R}}\right],
\end{aligned}
\tag{61b}
$$

$$
\begin{aligned}
f_{(2)B}^\mu = {}& \frac{1}{2} g^{\mu\nu} h_{u_{(0)}u_{(0)},\nu}^{(2)\mathcal{R}B} - P_{(0)}^{\mu\nu} U_{(0)} u_{(0)}^\beta h_{\beta\nu,\Omega}^{(1)\mathcal{R}} + \frac{1}{2} u_{(0)}^\mu U_{(0)} h_{u_{(0)}u_{(0)},\Omega}^{(1)\mathcal{R}} + \frac{\delta_t^\mu}{2} g^{tt} h_{u_{(0)}u_{(0)},\Omega}^{(1)\mathcal{R}} \\
& + U_{(0)} \partial_\Omega r_{(0)}\left[g^{\mu\nu} h_{\alpha r,\nu}^{(1)\mathcal{R}} u_{(0)}^\alpha + 2 P_{(0)}^{\mu\nu} \Gamma_{\alpha r}^{\sigma(0)} h_{\sigma\nu}^{(1)\mathcal{R}} u_{(0)}^\alpha \right. \\
& \left. - g^{\mu\nu} h_{\beta\nu,r}^{(1)\mathcal{R}} u_{(0)}^\beta - \frac{1}{2} u_{(0)}^\mu h_{u_{(0)}u_{(0)},r}^{(1)\mathcal{R}}\right].
\end{aligned}
\tag{61c}
$$

We have made use of the notation introduced in [9], where $h^{(n)\mathcal{R}}_{u_{(0)}u_{(0)},\nu} := h^{(n)\mathcal{R}}_{\alpha\beta,\nu} u^\alpha_{(0)} u^\beta_{(0)}$. We warn the reader that the $\nu$ derivative does not act on the four-velocities. Any $t$ derivative appearing in these final expanded expressions should be interpreted as $\partial_t \mapsto \Omega\partial_{\phi_p}$, as the full expansion of the time derivative from eq. (42) has already been applied. All fields are evaluated on the adiabatic worldline $z^\mu_{(0)}$. As a consequence, the self-force depends only on the mode amplitudes $R^{(n)\mathcal{R}}_{i\ell m}$ and not on the rapidly oscillating phases. Indeed, since on the worldline $\phi = \phi_p$, we have that $Y^{i\ell m}_{\mu\nu} e^{-im\phi_p} \propto e^{im(\phi-\phi_p)} = 1$. The $t$ component of the first-order self-force,

$$f^t_{(1)} = \frac{g^{tt}}{2} h^{(1)\mathcal{R}}_{u_{(0)}u_{(0)},\phi_p}\Omega = \frac{i\Omega}{2f_{(0)}r_{(0)}} \sum_{i\ell m} m\, a_{i\ell} R^{(1)\mathcal{R}}_{i\ell m}(J^a, r_{(0)}) Y^{i\ell m}_{\alpha\beta}(r_{(0)}, \pi/2, 0) u^\alpha_{(0)} u^\beta_{(0)}, \quad (62)$$

only receives contributions from the $m \neq 0$ modes. It is therefore fully determined in terms of the orbital frequency $\Omega$ and does not depend on the parameters of the slowly-evolving background $\delta M$ and $\delta J$, which only enter into the $\ell = 0$ and $\ell = 1$, $m = 0$ contributions [9]. Recalling eq. (13), we note that since only the $m \neq 0$ modes of $h^{(2)\mathcal{R}}_{\mu\nu}$ appear in the dissipative second-order self-force, it is linear in $\delta M^a = (1, \delta M, \delta J)$. The expressions for the radial self-force are such that $f^r_{(1)}$ is linear in $\delta M^a$, while $f^r_{(2)}$ is quadratic (given the discussion below eq. (54), the $f^r_{(2)B}$ component is however linear in $\delta M^a$).

In obtaining the results above it is crucial to note the following: since $u^\mu_{(0)}$ has components only along the $t$ and $\phi$ directions and since $\phi$ and $\phi_p$ derivatives of the metric perturbations only differ by an overall minus sign, we have the property

$$h^{(n)\mathcal{R}}_{\alpha\beta,\nu} u^\nu_{(0)} = \left(h^{(n)\mathcal{R}}_{\alpha\beta,\phi_p} + h^{(n)\mathcal{R}}_{\alpha\beta,\phi}\right)\Omega U_{(0)} + O(\varepsilon) = O(\varepsilon)$$

for all $n \geq 1$. Further details on this derivation (without the terms proportional to $F^{\delta M}_{(0)}$ and $F^{\delta J}_{(0)}$ in eq. (61b)) can be found in [48]. In order to obtain eqs. (61b) and (61c), one only needs to replace the $F^\Omega \partial_\Omega$ terms occurring in [48] with $F^{J^a}\partial_{J^a}$.

While the first-order self-force (61a) agrees with the one of [9] when re-written in the slow-time formulation, our expression at second order corrects the analogous one obtained in [9], where several terms were missed.

The structure of the first- and second-order self-forces (60) follows the one of the respective metric perturbations obtained in the previous subsection. At third order, given the structure of the metric perturbations (56), the self-force admits the following structure:

$$f^\mu_{(3)} = f^\mu_{(3)A}(J^a) + F^\Omega_{(0)} f^\mu_{(3)B}(J^a) + F^\Omega_{(1)} f^\mu_{(3)C}(J^a) + \left(F^\Omega_{(0)}\right)^2 f^\mu_{(3)D}(J^a) + F^\Omega_{(0)}\partial_\Omega F^\Omega_{(0)} f^\mu_{(3)E}(J^a). \quad (63)$$

### 3.4 Near-ISCO solution: orbital motion

Recall that the function $D$, which appears as a pole in eqs. (37), (39) and (40), vanishes at the ISCO frequency $\Omega_* = 1/(6\sqrt{6}M)$. The ISCO marks the breakdown of the inspiral expansion as the motion enters into the transition-to-plunge regime. In order to asymptotically match with the transition-to-plunge expansion in section 5 below, we are interested in the near-ISCO limit of the inspiral motion. After substituting the self-force and the forcing terms $F^{\delta M}_{(1)}$ and $F^{\delta J}_{(1)}$ using eqs. (60), (63) and (55), we take the $\Omega \to \Omega_*$ limit of eqs. (37), (39) and (40). For easier comparison with the transition-to-plunge motion, we replace the difference $\Omega - \Omega_*$ using the near-ISCO scaling of the orbital frequency, $\Omega = \Omega_* + \lambda^2\Delta\Omega$ where $\lambda := \varepsilon^{1/5}$ (recall eq. (6)). Near the ISCO we then have $D = -4\sqrt{6}M\lambda^2\Delta\Omega + O(\lambda^4\Delta\Omega^2)$. At adiabatic order, we find that $r_{(0)}$ and $F^\Omega_{(0)}$ have the following near-ISCO solutions:

$$r_{(0)}(\Omega \to \Omega_*) = 6M + \sum_{n=1}^{\infty} r^{(2n,n)}_{(0)}\lambda^{2n}\Delta\Omega^n, \quad (64a)$$

$$\varepsilon\, F^{\Omega}_{(0)}(\Omega \to \Omega_*) = F^{(3,-1)}_{(0)} \frac{\lambda^3}{\Delta\Omega} + \sum_{n=0}^{\infty} F^{(5+2n,n)}_{(0)} \lambda^{5+2n} \Delta\Omega^n \,. \tag{64b}$$

We have highlighted the terms that diverge at the ISCO, which contain negative powers of $\Delta\Omega$, by taking them out of the summation. The expansion coefficients are constants constructed from $\Omega_*$. We use the following notation: each coefficient $r^{(i,j)}_{(n)}$ and $F^{(i,j)}_{(n)}$ with $n \in \mathbb{N}$ and $i, j \in \mathbb{Z}$ is labelled according to the powers of, respectively, $\lambda$ and $\Delta\Omega$ with which it appears in the expansion. Note that each couple $(i, j)$ originates from a single term $(n)$, but we keep the notation $(n)$ explicit for book-keeping purposes. Starting from 1PA order the expansion coefficients in general depend on $\delta M$ and $\delta J$. Some of these coefficients are defined explicitly in appendix B.2. Expanding the 1PA equations (39), we obtain

$$\varepsilon\, r_{(1)}(\Omega \to \Omega_*, \delta M^{\pm}) = \sum_{n=0}^{\infty} r^{(5+2n,n)}_{(1)} \lambda^{5+2n} \Delta\Omega^n \,, \tag{65a}$$

$$\varepsilon^2 F^{\Omega}_{(1)}(\Omega \to \Omega_*, \delta M^{\pm}) = F^{(6,-2)}_{(1)} \frac{\lambda^6}{\Delta\Omega^2} + F^{(8,-1)}_{(1)} \frac{\lambda^8}{\Delta\Omega} + \sum_{n=0}^{\infty} F^{(10+2n,n)}_{(1)} \lambda^{10+2n} \Delta\Omega^n \,. \tag{65b}$$

The near-ISCO expansion of the 2PA equations (40) gives

$$\varepsilon^2 r_{(2)}(\Omega \to \Omega_*, \delta M^{\pm}) = r^{(4,-3)}_{(2)} \frac{\lambda^4}{\Delta\Omega^3} + r^{(6,-2)}_{(2)} \frac{\lambda^6}{\Delta\Omega^2} + r^{(8,-1)}_{(2)} \frac{\lambda^8}{\Delta\Omega} + \sum_{n=0}^{\infty} r^{(10+2n,n)}_{(2)} \lambda^{10+2n} \Delta\Omega^n \,, \tag{66a}$$

$$\varepsilon^3 F^{\Omega}_{(2)}(\Omega \to \Omega_*, \delta M^{\pm}) = F^{(3,-6)}_{(2)} \frac{\lambda^3}{\Delta\Omega^6} + F^{(5,-5)}_{(2)} \frac{\lambda^5}{\Delta\Omega^5} + \cdots + \sum_{n=0}^{\infty} F^{(15+2n,n)}_{(2)} \lambda^{15+2n} \Delta\Omega^n \,. \tag{66b}$$

We now determine the range of validity of the adiabatic and post-adiabatic approximations based on the near-ISCO solutions computed in this subsection. The 0PA inspiral is valid outside the ISCO as long as 1PA corrections remain subleading. The adiabatic breakdown frequency $\Omega^{0PA}_{bd}$ is therefore reached when the leading-order terms in the $\varepsilon\, F^{\Omega}_{(0)}$ and $\varepsilon^2 F^{\Omega}_{(1)}$ near-ISCO solutions, respectively $F^{(3,-1)}_{(0)} \lambda^3 \Delta\Omega^{-1}$ and $F^{(6,-2)}_{(1)} \lambda^6 \Delta\Omega^{-2}$, become comparable. This condition leads to (we recall that $\lambda = \varepsilon^{1/5}$)

$$\Omega^{0PA}_{bd} = \Omega_* - \varepsilon \left| \frac{F^{(6,-2)}_{(1)}}{F^{(3,-1)}_{(0)}} \right| \,. \tag{67}$$

Similarly, the 1PA motion breaks down when the term $F^{(3,-6)}_{(2)} \lambda^3 \Delta\Omega^{-6}$ in the 2PA near-ISCO inspiral (66b) becomes of the same magnitude as the leading-order term in the 1PA solution (65b). The 1PA breakdown frequency is therefore given by

$$\Omega^{1PA}_{bd} = \Omega_* - \varepsilon^{1/4} \left| \frac{F^{(3,-6)}_{(2)}}{F^{(6,-2)}_{(1)}} \right|^{1/4} \,, \tag{68}$$

which was already obtained in eq. (29) of [18]. We give the numerical values of the coefficients appearing in the expressions above in appendix D.2. As a consequence of the alternating structure between even and odd post-adiabatic orders in the near-ISCO limit of the inspiral motion [14, 18], the ratio of the degree of divergence at the ISCO of the $(2n + 1)$PA forcing term over the $2n$PA forcing term is identical for all $n$. The qualitative behaviour in terms of the

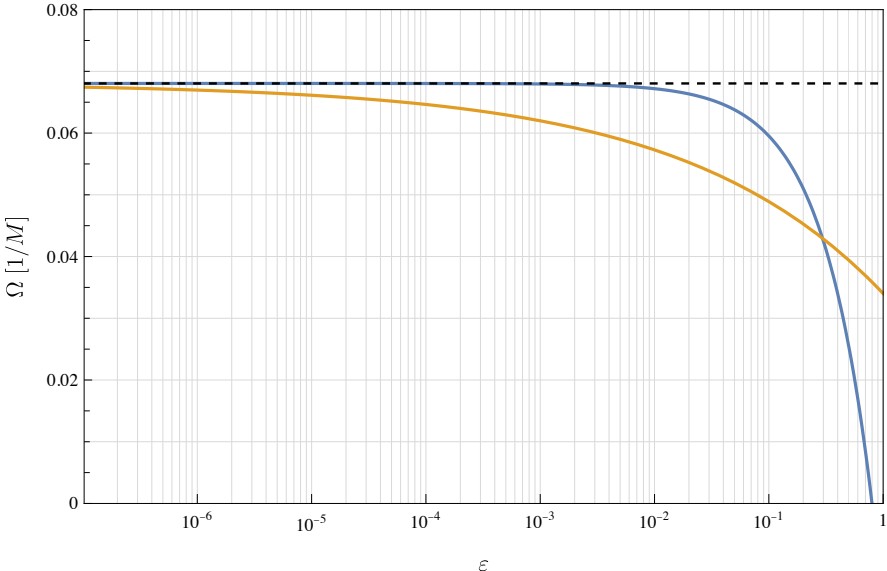

Figure 1: Breakdown frequencies in terms of the mass ratio for the 0PA motion (eq. (67)) in blue and the 1PA motion (eq. (68)) in orange. The horizontal dashed line marks the ISCO frequency. The 0PA and 1PA approximations are valid in the regions below the blue and orange curves, respectively.

mass ratio for such $2n$ will therefore be similar to the one in eq. (67). Equivalently, the ratio of the degree of divergence at the ISCO of the $(2n + 2)$PA forcing term over the $(2n + 1)$PA forcing term is identical for all $n$ and is again qualitatively similar to the one in eq. (68). We plot the 0PA and 1PA breakdown frequencies as a function of the mass ratio in figure 1. The breakdown frequencies derived using the near-ISCO solutions provide a rough estimate of the domain of validity of the inspiral expansion truncated at some order in the mass ratio. This estimate is most accurate for small mass ratios: for nearly comparable-mass systems, the separation between subsequent post-adiabatic orders is less marked. In this scenario, the breakdown might already occur at frequencies where the near-ISCO solutions are not yet valid, and the method we have used here to derive the breakdown frequencies can therefore not be applied. The breakdown frequencies represent the upper limit on the validity of a given $n$PA approximation. This does not necessarily coincide with the range of frequencies where the inspiral accurately describes the motion, and the transition-to-plunge expansion becomes more accurate before the breakdown frequency is reached. We discuss this in more detail in section 5.

### 3.5 Near-ISCO solution: metric perturbation and self-force

The first-order quantities $R^{(1)}_{i\ell m}$ and $f^{\mu}_{(1)}$ are smooth functions of the orbital frequency and can therefore be Taylor-expanded around the ISCO as

$$R^{(1)}_{i\ell m}(\Omega \to \Omega_*, \delta M^{\pm}, r) = R^{(1)}_{i\ell m}\Big|_* + \lambda^2 \Delta\Omega\, \partial_\Omega R^{(1)}_{i\ell m}\Big|_* + \frac{\lambda^4 \Delta\Omega^2}{2}\, \partial_\Omega^2 R^{(1)}_{i\ell m}\Big|_* + O(\lambda^6 \Delta\Omega^3), \quad (69)$$

$$f^{\mu}_{(1)}(\Omega \to \Omega_*, \delta M^{\pm}) = f^{\mu}_{(1)}\Big|_* + \lambda^2 \Delta\Omega\, \partial_\Omega f^{\mu}_{(1)}\Big|_* + \frac{\lambda^4 \Delta\Omega^2}{2}\, \partial_\Omega^2 f^{\mu}_{(1)}\Big|_* + O(\lambda^6 \Delta\Omega^3), \quad (70)$$

where we have introduced the short notation $|_*$ to indicate functions evaluated at $\Omega = \Omega_*$. The expansion coefficients are constants in $\Omega$, but still depend on $\delta M$, $\delta J$ and, in the case of

the metric perturbations, on the field point $r$. At second order, the metric perturbation mode amplitudes and the self-force have the structure given by eqs. (53) and (60b), respectively. Taylor-expanding the smooth functions of $\Omega$ in these expressions and using the near-ISCO solution (64b), we obtain

$$
\begin{aligned}
R_{i\ell m}^{(2)}(\Omega \to \Omega_*, \delta M^\pm, r) = {}& \frac{F_{(0)}^{(3,-1)}}{\lambda^2 \Delta\Omega} R_{i\ell m}^{(2)B}\Big|_* + R_{i\ell m}^{(2)A}\Big|_* + F_{(0)}^{(3,-1)} \partial_\Omega R_{i\ell m}^{(2)B}\Big|_* \\
& + F_{(0)}^{(5,0)} R_{i\ell m}^{(2)B}\Big|_* + \lambda^2\Delta\Omega\left(\partial_\Omega R_{i\ell m}^{(2)A}\Big|_* + F_{(0)}^{(7,1)} R_{i\ell m}^{(2)B}\Big|_*\right. \\
& \left. + F_{(0)}^{(5,0)} \partial_\Omega R_{i\ell m}^{(2)B}\Big|_* + \frac{1}{2}F_{(0)}^{(3,-1)} \partial_\Omega^2 R_{i\ell m}^{(2)B}\Big|_*\right) + O(\lambda^4\Delta\Omega^2),
\end{aligned}
\tag{71}
$$

$$
\begin{aligned}
f_{(2)}^\mu(\Omega \to \Omega_*, \delta M^\pm) = {}& \frac{F_{(0)}^{(3,-1)}}{\lambda^2 \Delta\Omega} f_{(2)B}^\mu\Big|_* + f_{(2)A}^\mu\Big|_* + F_{(0)}^{(3,-1)} \partial_\Omega f_{(2)B}^\mu\Big|_* \\
& + F_{(0)}^{(5,0)} f_{(2)B}^\mu\Big|_* + \lambda^2\Delta\Omega\left(\partial_\Omega f_{(2)A}^\mu\Big|_* + F_{(0)}^{(7,1)} f_{(2)B}^\mu\Big|_*\right. \\
& \left. + F_{(0)}^{(5,0)} \partial_\Omega f_{(2)B}^\mu\Big|_* + \frac{1}{2}F_{(0)}^{(3,-1)} \partial_\Omega^2 f_{(2)B}^\mu\Big|_*\right) + O(\lambda^4\Delta\Omega^2).
\end{aligned}
\tag{72}
$$

Finally, substituting eqs. (64b) and (65b) into eqs. (56) and (63) and Taylor-expanding the smooth functions of $\Omega$, we compute the near-ISCO behaviour of the third-order metric perturbation mode amplitudes and self-force:

$$
\begin{aligned}
R_{i\ell m}^{(3)}(\Omega \to \Omega_*, \delta M^\pm, r) = {}& -\frac{\left(F_{(0)}^{(3,-1)}\right)^2}{\lambda^6 \Delta\Omega^3} R_{i\ell m}^{(3)E}\Big|_* + \frac{1}{\lambda^4 \Delta\Omega^2}\left[F_{(1)}^{(6,-2)} R_{i\ell m}^{(3)C}\Big|_*\right. \\
& + \left(F_{(0)}^{(3,-1)}\right)^2 R_{i\ell m}^{(3)D}\Big|_* - F_{(0)}^{(3,-1)}F_{(0)}^{(5,0)} R_{i\ell m}^{(3)E}\Big|_* \\
& \left. - \left(F_{(0)}^{(3,-1)}\right)^2 \partial_\Omega R_{i\ell m}^{(3)E}\Big|_*\right] + O(\lambda^{-2}\Delta\Omega^{-1}),
\end{aligned}
\tag{73}
$$

$$
\begin{aligned}
f_{(3)}^\mu(\Omega \to \Omega_*, \delta M^\pm) = {}& -\frac{\left(F_{(0)}^{(3,-1)}\right)^2}{\lambda^6 \Delta\Omega^3} f_{(3)E}^\mu\Big|_* + \frac{1}{\lambda^4 \Delta\Omega^2}\left[F_{(1)}^{(6,-2)} f_{(3)C}^\mu\Big|_*\right. \\
& + \left(F_{(0)}^{(3,-1)}\right)^2 f_{(3)D}^\mu\Big|_* - F_{(0)}^{(3,-1)}F_{(0)}^{(5,0)} f_{(3)E}^\mu\Big|_* \\
& \left. - \left(F_{(0)}^{(3,-1)}\right)^2 \partial_\Omega f_{(3)E}^\mu\Big|_*\right] + O(\lambda^{-2}\Delta\Omega^{-1}).
\end{aligned}
\tag{74}
$$

## 4 Transition to plunge

There are three timescales during the transition to plunge: as during the inspiral, the azimuthal phase $\phi_p$ changes on the orbital period $\sim M$ and the primary black hole's evolution takes place on a timescale $\sim M/\varepsilon$. However, the orbital parameters now evolve on the ISCO-crossing timescale $\sim M/\varepsilon^{1/5}$. Fortunately, the ISCO-crossing timescale and background evolution timescale are commensurate, which allows us to expand all quantities simultaneously in an $\varepsilon^{1/5}$ expansion.

In the Ori-Thorne analysis [11], the orbital frequency is held fixed at its geodesic ISCO value during the full transition-to-plunge regime, leading to inconsistencies in the normalization of the four-velocity [49]. It was shown in [14] that in the transition-to-plunge expansion

that matches with the quasi-circular inspiral, the orbital frequency scales with the small mass ratio to the power 2/5. As anticipated in section 2.1, the mechanical parameters we consider for the transition to plunge are $\Delta J^a = (\Delta\Omega, \delta M, \delta J) = (\Delta\Omega, \delta M^\pm)$, where $\Delta\Omega$ is defined from the near-ISCO scaling of the orbital frequency,

$$\Omega(t) = \Omega_* + \lambda^2 \Delta\Omega(t), \qquad \Omega_* := \frac{1}{6\sqrt{6}M}, \qquad \lambda := \varepsilon^{1/5}. \tag{75}$$

As during the inspiral, $\delta M$ and $\delta J$ are the variations of mass and spin of the primary divided by $\varepsilon$, which we collectively denote as $\delta M^\pm$.

We now introduce the *transition-to-plunge expansion* of the orbital quantities, that is, an expansion in integer powers of $\lambda$ at fixed mechanical parameters $\Delta J^a$. For the orbital radius and the redshift we have

$$r_p(\lambda, \Delta J^a) = r_* + \lambda^2 \sum_{n=0}^{2} \lambda^n r_{[n]}(\Delta\Omega) + \lambda^2 \sum_{n=3}^{\infty} \lambda^n r_{[n]}(\Delta J^a), \qquad r_* := 6M, \tag{76}$$

$$U(\lambda, \Delta J^a) = U_* + \lambda^2 \sum_{n=0}^{6} \lambda^n U_{[n]}(\Delta\Omega) + \lambda^2 \sum_{n=7}^{\infty} \lambda^n U_{[n]}(\Delta J^a), \qquad U_* := \sqrt{2}, \tag{77}$$

where $r_*$ and $U_*$ are the geodesic ISCO values. We indicate the leading transition-to-plunge order or the zeroth post-leading transition-to-plunge (0PLT) order with a subscript $[0]$ in square brackets. Here this term appears at the leading order $\lambda^2$, where the power 2 can be interpreted as a critical exponent under the scaling $\lambda \mapsto 0$ in the transition-to-plunge expansion [11, 14, 49–51]. We refer to the $n^{th}$ subleading term in the transition-to-plunge expansion as $n^{th}$ post-leading transition-to-plunge or $n$PLT order and label it with a subscript $[n]$. As we will see in the next section, for each orbital variable an associated number of low PLT orders only depend on $\Delta\Omega$, while the higher-order terms are in general functions of $\Delta J^a$. During the transition to plunge, the evolution of the mechanical parameters $\Delta J^a$ is given by

$$\frac{1}{\lambda}\frac{d\Delta\Omega}{dt} = \frac{1}{\lambda^3}\frac{d\Omega}{dt} = \sum_{n=0}^{2} \lambda^n F_{[n]}^{\Delta\Omega}(\Delta\Omega) + \sum_{n=3}^{\infty} \lambda^n F_{[n]}^{\Delta\Omega}(\Delta J^a), \tag{78}$$

$$\frac{1}{\lambda^5}\frac{d\delta M^\pm}{dt} = \sum_{n=0}^{2} \lambda^n F_{[n]}^{\delta M^\pm}(\Delta\Omega) + \sum_{n=3}^{\infty} \lambda^n F_{[n]}^{\delta M^\pm}(\Delta J^a). \tag{79}$$

The ISCO-crossing "slow-time" variable $\lambda t$ could be introduced to absorb the inverse power of $\lambda$ on the left-hand side of eq. (78), while the background evolution time $\lambda^5 t$ could be introduced to absorb the $\lambda^{-5}$ factor in eq. (79). As for the inspiral, we find it most natural not to introduce any of these auxiliary times and simply consider the evolution in Boyer-Lindquist time $t$, keeping the factors of $\lambda$ explicit. The transition-to-plunge expansion of the self-force reads

$$f^\mu(\lambda, \Delta J^a) = \sum_{n=0}^{\infty} \lambda^{5+n} f_{[5+n]}^\mu(\Delta J^a). \tag{80}$$

We refer to the term appearing at order $\lambda^{5+n}$ as the $n$PLT term because the leading-order term in the self-force is proportional to $\lambda^5$ (this is also the case for the metric perturbation, see section 4.2). This is consistent with our convention to denote the first non-vanishing term in the transition-to-plunge expansion as the 0PLT term (independently of the power of $\lambda$ multiplying this leading-order term). As we will see from the explicit expressions we derive in section 4.3, the self-force at 1PLT order is identically zero, $f_{[6]}^\mu = 0$. The dissipative self-force up to 2PLT order is independent of $\delta M$ and $\delta J$: $f_{[5]}^t$ and $f_{[5]}^\phi$ are constants, while $f_{[7]}^t$

and $f^{\phi}_{[7]}$ are constants multiplied by $\Delta\Omega$. Up to 5PLT order, the dissipative self-force is linear in $\delta M^a = (1, \delta M, \delta J)$, with terms up to 10PLT order being quadratic. The conservative self-force is linear in $\delta M^a$ up to 4PLT order and quadratic up to 9PLT order.

## 4.1 Orbital motion from 0PLT to 7PLT order

Using eqs. (75), (76), (77), (78) and (79), we perform the transition-to-plunge expansion of the worldline (4) and the four-velocity (8). We substitute these expansions together with the one of the self-force (80) into the normalization condition (9) and the equation of motion (10). At order $\lambda^{2+n}$, $n \geq 0$, we obtain algebraic equations for $r_{[n]}$ and $U_{[n]}$ from the radial component of the equation of motion and the normalization of the four-velocity, respectively. The forcing terms $F^{\Delta\Omega}_{[n]}$ satisfy ordinary differential equations that are obtained from the $t$ component of the equation of motion at order $\lambda^{5+n}$. The forcing terms $F^{\delta M}_{[n]}$ and $F^{\delta J}_{[n]}$ are determined from flux-balance laws at the horizon of the primary. As we will demonstrate in section 5, it is necessary to solve the transition-to-plunge motion up to 7PLT order to ensure a continuous ($\mathcal{C}^0$) composite solution for the rate of change $d\Omega/dt$ that involves the 1PA inspiral (this implies a $\mathcal{C}^1$ composite solution for $\Omega$ and a $\mathcal{C}^2$ composite solution for $\phi_p$ after one and two time integrations, respectively).

At leading order we obtain

$$U_{[0]} = 4\sqrt{3}M\Delta\Omega, \tag{81a}$$

$$r_{[0]} = -24\sqrt{6}M^2\Delta\Omega. \tag{81b}$$

The 1PLT corrections to the orbital radius and redshift vanish: $r_{[1]} = U_{[1]} = 0$. At 2PLT order we obtain

$$U_{[2]} = 24\sqrt{2}M^2\Delta\Omega^2, \tag{82a}$$

$$r_{[2]} = 144M^3\left(5\Delta\Omega^2 - 18\sqrt{6}MF^{\Delta\Omega}_{[0]}\frac{dF^{\Delta\Omega}_{[0]}}{d\Delta\Omega}\right). \tag{82b}$$

The expressions up to 7PLT order are presented in appendix C.1.

The leading-order forcing term $F^{\Delta\Omega}_{[0]}(\Delta\Omega)$ satisfies the following ordinary differential equation:

$$\left(F^{\Delta\Omega}_{[0]}\right)^2\frac{d^2F^{\Delta\Omega}_{[0]}}{d\Delta\Omega^2} + F^{\Delta\Omega}_{[0]}\left(\frac{dF^{\Delta\Omega}_{[0]}}{d\Delta\Omega}\right)^2 - \frac{1}{9\sqrt{6}M}\Delta\Omega F^{\Delta\Omega}_{[0]} = -\frac{f^t_{[5]}}{432\sqrt{6}M^3}. \tag{83}$$

This equation is in disguise the Painlevé transcendental equation of the first kind, identified in [50] following [10, 11]. This can be seen as follows. We define the time $s_{[0]} = \int d\Delta\Omega(F^{\Delta\Omega}_{[0]})^{-1}$ with the integration constant chosen such that $s_{[0]} = 0$ at the ISCO crossing. This definition implies

$$\frac{d\Delta\Omega}{ds_{[0]}} = F^{\Delta\Omega}_{[0]}, \quad \frac{d^2\Delta\Omega}{ds^2_{[0]}} = \frac{dF^{\Delta\Omega}_{[0]}}{d\Delta\Omega}F^{\Delta\Omega}_{[0]}, \quad \frac{d^3\Delta\Omega}{ds^3_{[0]}} = \left(F^{\Delta\Omega}_{[0]}\right)^2\frac{d^2F^{\Delta\Omega}_{[0]}}{d\Delta\Omega^2} + F^{\Delta\Omega}_{[0]}\left(\frac{dF^{\Delta\Omega}_{[0]}}{d\Delta\Omega}\right)^2. \tag{84}$$

After integrating eq. (83) once and using $s_{[0]} = 0$ at the ISCO, it becomes indeed a Painlevé transcendental equation of the first kind,

$$\frac{d^2\Delta\Omega}{ds^2_{[0]}} - \frac{1}{18\sqrt{6}M}\Delta\Omega^2 = -\frac{f^t_{[5]}}{432\sqrt{6}M^3}s_{[0]}. \tag{85}$$

We select the unique monotonic solution of this differential equation as done in [11, 14].

The subleading forcing terms, $F_{[n]}^{\Delta\Omega}(\Delta\Omega)$ for $1 \leq n \leq 2$ and $F_{[n]}^{\Delta\Omega}(\Delta J^a)$ for $n \geq 3$, obey the sourced ordinary differential equations

$$
\left(F_{[0]}^{\Delta\Omega}\right)^2 \frac{d^2 F_{[n]}^{\Delta\Omega}}{d\Delta\Omega^2} + 2F_{[0]}^{\Delta\Omega} F_{[n]}^{\Delta\Omega} \frac{d^2 F_{[0]}^{\Delta\Omega}}{d\Delta\Omega^2} + F_{[n]}^{\Delta\Omega} \left(\frac{dF_{[0]}^{\Delta\Omega}}{d\Delta\Omega}\right)^2
$$
$$
+ 2F_{[0]}^{\Delta\Omega} \frac{dF_{[0]}^{\Delta\Omega}}{d\Delta\Omega} \frac{dF_{[n]}^{\Delta\Omega}}{d\Delta\Omega} - \frac{1}{9\sqrt{6}M} \Delta\Omega F_{[n]}^{\Delta\Omega} = S_{[n]}^{\Delta\Omega}. \quad (86)
$$

The source terms $S_{[n]}^{\Delta\Omega}$ are listed in appendix C.1 from $n = 2$ to $n = 5$. For $n = 1$, the source is zero, $S_{[1]}^{\Delta\Omega} = 0$. The homogeneous differential operator is the linearization of the non-linear Painlevé operator on the left-hand side of eq. (83). The linearized Painlevé solutions around the monotonic solution are all oscillatory, see [14]. We therefore set all these homogeneous solutions to zero. In particular, we set $F_{[1]}^{\Delta\Omega} = 0$. The leading-order transition-to-plunge motion is driven by the $t$ component of the first-order self-force evaluated at the ISCO, $f_{[5]}^t$. At 2PLT order, the equations start to depend also on $f_{[7]}^t(\Delta\Omega)$. The 3PLT order additionally requires the knowledge of $f_{[8]}^t(\Delta J^a)$, $f_{[5]}^r(\Delta J^a)$ and $f_{[7]}^r(\Delta J^a)$. Hence, starting from 3PLT order, the forcing terms depend on $\delta M$ and $\delta J$. In general, $n$PLT corrections with $n \geq 4$ require $f_{[5+n]}^t(\Delta J^a)$ and $f_{[4+n]}^r(\Delta J^a)$, in addition to self-force terms already appearing at lower orders. Therefore, starting from 6PLT order, self-force data quadratic in $\delta M^a = (1, \delta M, \delta J)$ is required in order to solve for the motion. This leads us to write the following decompositions in terms of $\delta M^a$:

$$
S_{[n]}^{\Delta\Omega} = S_{[n]a}^{\Delta\Omega} \delta M^a, \qquad F_{[n]}^{\Delta\Omega} = F_{[n]a}^{\Delta\Omega} \delta M^a, \qquad n = 3, 4, 5,
$$
$$
S_{[n]}^{\Delta\Omega} = S_{[n]ab}^{\Delta\Omega} \delta M^a \delta M^b, \qquad F_{[n]}^{\Delta\Omega} = F_{[n]ab}^{\Delta\Omega} \delta M^a \delta M^b, \qquad n = 6, 7. \quad (87)
$$

Obtaining the transition-to-plunge motion to 7PLT order amounts to solving the non-linear Painlevé equation for $F_{[0]}^{\Delta\Omega}$ and in total 22 sourced linearized Painlevé equations for $F_{[n]}^{\Delta\Omega}$, $n = 2, 3, 4, 5, 6, 7$ (one for $F_{[2]}^{\Delta\Omega}$; 9 for all $F_{[n]a}^{\Delta\Omega}$ with $n = 3, 4, 5$ and $a = 1, 2, 3$; and 18 for all $F_{[n]ab}^{\Delta\Omega}$ with $n = 6, 7$ and $ab = (ab)$ symmetrized with $a, b = 1, 2, 3$).

## 4.2 Einstein's field equations from 0PLT to 7PLT order

We now turn to the expansion of the field equations (31) during the transition-to-plunge regime. The transition-to-plunge expansion (29) of the trace-reversed metric perturbation takes the form

$$
\bar{h}_{\mu\nu}(\lambda, s, x^i) = \sum_{n=0}^{\infty} \lambda^{5+n} \sum_{i\ell m} \frac{a_{i\ell}}{r} R_{i\ell m}^{[5+n]}(\Delta J^a(\lambda, s), r) e^{-im\phi_p(\lambda, s)} Y_{\mu\nu}^{i\ell m}(r, \theta, \phi). \quad (88)
$$

We call the term appearing at order $\lambda^{5+n}$ the $n$PLT term. The residual and puncture parts of $R_{i\ell m}^{[n]}$ are denoted respectively as $R_{i\ell m}^{[n]\mathcal{R}}$ and $R_{i\ell m}^{[n]\mathcal{P}}$.

In analogy with eq. (42), the transition-to-plunge expansion of eq. (27) gives

$$
\widehat{(\partial_t)}_r = -im\Omega_* - im\lambda^2 \Delta\Omega + \lambda \sum_{n=0}^{\infty} \lambda^n F_{[n]}^{\Delta\Omega} \partial_{\Delta\Omega} + \lambda^5 \sum_{n=0}^{\infty} \lambda^n F_{[n]}^{\delta M^{\pm}} \partial_{\delta M^{\pm}}, \quad (89a)
$$

$$
\widehat{(\partial_x)}_t = (\partial_x)_{\Delta J^a} + H\left[im\Omega_* + im\lambda^2 \Delta\Omega - \lambda \sum_{n=0}^{\infty} \lambda^n F_{[n]}^{\Delta\Omega} \partial_{\Delta\Omega} - \lambda^5 \sum_{n=0}^{\infty} \lambda^n F_{[n]}^{\delta M^{\pm}} \partial_{\delta M^{\pm}}\right]. \quad (89b)
$$

Here we have used eqs. (78) and (79) and recalled the replacement $\partial_{\phi_p} \to -im$ in hatted operators acting on functions of $(\Delta J^a, r)$. The linearized Einstein operator (26) is then expanded as

$$\widehat{E}_{ij\ell m}(\lambda, \Delta J^a, r) = \sum_{n=0}^{3} \lambda^n E_{ij\ell m}^{[n]}(\Delta\Omega, r) + \sum_{n=4}^{6} \lambda^n E_{ij\ell m}^{[n]a}(\Delta\Omega, r)\delta M^a$$
$$+ \lambda^7 E_{ij\ell m}^{[7]ab}(\Delta\Omega, r)\delta M^a \delta M^b + \dots, \tag{90}$$

where

$$E_{ij\ell m}^{[0]} = -\frac{\delta_{ij}}{4}\left[\partial_x^2 + im\Omega_*(\partial_x H + 2H\partial_x) + m^2\Omega_*^2(1 - H^2) - 4V_\ell\right] \tag{91a}$$
$$+ \mathcal{M}_r^{ij} - im\Omega_* \mathcal{M}_t^{ij},$$

$$E_{ij\ell m}^{[1]} = \frac{\delta_{ij}}{4}\left[\left(\partial_x H - 2im\Omega_*(1 - H^2)\right)F_{[0]}^{\Delta\Omega}\partial_{\Delta\Omega} + 2HF_{[0]}^{\Delta\Omega}\partial_{\Delta\Omega}\partial_x\right] + \mathcal{M}_t^{ij}F_{[0]}^{\Delta\Omega}\partial_{\Delta\Omega}, \tag{91b}$$

$$E_{ij\ell m}^{[2]} = \frac{\delta_{ij}}{4}\Bigg[-2m^2\Omega_*\Delta\Omega(1 - H^2)$$
$$+ (1 - H^2)\left(F_{[0]}^{\Delta\Omega}\left(\partial_{\Delta\Omega}F_{[0]}^{\Delta\Omega}\right)\partial_{\Delta\Omega} + \left(F_{[0]}^{\Delta\Omega}\right)^2\partial_{\Delta\Omega}^2\right) \tag{91c}$$
$$- im\Delta\Omega(\partial_x H + 2H\partial_x)\Bigg] - im\Delta\Omega\mathcal{M}_t^{ij}.$$

Note that the leading term, $E_{ij\ell m}^{[0]}$, is identical to the leading term $E_{ij\ell m}^{(0)}$ in the inspiral evaluated at the ISCO frequency. The linearized Einstein operators $E_{ij\ell m}^{[n]a}$, $n = 3, 4, 5, 6$ and $E_{ij\ell m}^{[7]ab}$ are provided in appendix C.2. Just as in the inspiral, this expansion of the linearized Einstein operator will reduce the partial differential equations in $(\Delta J^a, r)$ to ordinary differential equations in $r$ because derivatives with respect to $\Delta J^a$ are accompanied by powers of $\lambda$.

Unlike in the inspiral, where nonlinear sources appear at the first subleading order, in the transition to plunge there are several intermediate orders (2PLT through 4PLT) in which no nonlinearities appear. At these orders, the sources are constructed entirely from subleading terms in the expansions of (i) $\widehat{E}_{ij\ell m}$ and (ii) the point-particle stress-energy mode amplitudes $t_{i\ell m}$. We obtain the transition-to-plunge expansion of $t_{i\ell m}$ by substituting eqs. (8), (76), (77), (78) and (79) together with the results of section 4.1 and appendix C.1 into the definition (21). In order to compactify the notation, we absorb the radial $\delta$-function appearing in eq. (21) into the definition of the mode amplitudes. We obtain

$$\varepsilon\, t_{i\ell m}(\lambda, \Delta J^a)\delta(r - r_p) = \lambda^5 \sum_{n=0}^{4} \lambda^n t_{i\ell m}^{[5+n]}(\Delta\Omega) + \lambda^5 \sum_{n=5}^{\infty} \lambda^n t_{i\ell m}^{[5+n]}(\Delta J^a). \tag{92}$$

The leading-order modes are given by

$$t_{i\ell m}^{[5]} = -\frac{1}{4}\delta(r - 6M)\mathcal{E}_*\alpha_{i\ell m}^{[5]}\begin{cases} Y_{\ell m}^*(\frac{\pi}{2}, 0) & i = 1, \dots, 7, \\ \partial_\theta Y_{\ell m}^*(\frac{\pi}{2}, 0) & i = 8, 9, 10, \end{cases} \tag{93}$$

where $\mathcal{E}_* = Mf_*U_* = 2\sqrt{2}M/3$ (here $f_* := 1 - 2M/r_* = 2/3$) and the coefficients $\alpha_{i\ell m}^{[5]}$ are

given by (dropping $\ell$ and $m$ indices)

$$\alpha_1^{[5]} = \frac{f_*^2}{r_*} = \frac{2}{27M}, \quad \alpha_{2,5,9}^{[5]} = 0, \quad \alpha_3^{[5]} = \frac{f_*}{r_*} = \frac{1}{9M}, \quad \alpha_4^{[5]} = 2imf_*\Omega_* = \frac{i\sqrt{2}m}{9\sqrt{3}M},$$

$$\alpha_6^{[5]} = r_*\Omega_*^2 = \frac{1}{36M}, \alpha_7^{[5]} = [\ell(\ell+1)-2m^2]r_*\Omega_*^2 = \frac{\ell(\ell+1)-2m^2}{36M}, \tag{94}$$

$$\alpha_8^{[5]} = 2f_*\Omega_* = \frac{\sqrt{2}}{9\sqrt{3}M}, \quad \alpha_{10}^{[5]} = 2im\Omega_*^2 r_* = \frac{im}{18M}.$$

At 1PLT order we obtain $t_{i\ell m}^{[6]} = 0$. The 2PLT modes are given by

$$t_{i\ell m}^{[7]} = \Delta\Omega\, t_{i\ell m}^{[7]A} = -\frac{\Delta\Omega}{4}\mathcal{E}_*\alpha_{i\ell m}^{[7]A} \begin{cases} Y_{\ell m}^*(\frac{\pi}{2},0) & i = 1,\ldots,7, \\ \partial_\theta Y_{\ell m}^*(\frac{\pi}{2},0) & i = 8,9,10, \end{cases} \tag{95}$$

where (dropping again the $\ell$ and $m$ indices and using the short notation $\delta := \delta(r-6M)$ and $\delta' := \delta'(r-6M)$)

$$\alpha_1^{[7]A} = \frac{16}{3}\sqrt{\frac{2}{3}}M\delta', \quad \alpha_{2,5,9}^{[7]A} = 0, \quad \alpha_3^{[7]A} = \frac{2}{3}\sqrt{\frac{2}{3}}\left[\delta + 12M\delta'\right],$$

$$\alpha_6^{[7]A} = \frac{2}{3}\sqrt{\frac{2}{3}}\left[\delta + 3M\delta'\right], \quad \alpha_7^{[7]A} = \frac{2}{3}\sqrt{\frac{2}{3}}\left(\ell(\ell+1)-2m^2\right)\left[\delta + 3M\delta'\right], \tag{96}$$

$$\alpha_8^{[7]A} = \frac{8}{9}\left[\delta + 6M\delta'\right], \quad \alpha_{10}^{[7]A} = \frac{4}{3}\sqrt{\frac{2}{3}}im\left[\delta + 3M\delta'\right].$$

The modes at 3PLT and 4PLT order are given in appendix C.3. Higher subleading terms can also be straightforwardly calculated.

Up to 2PLT order, the expanded field equations (31) then read

$$E_{ij\ell m}^{[0]} R_{j\ell m}^{[5]} = -16\pi t_{i\ell m}^{[5]}, \tag{97}$$

$$E_{ij\ell m}^{[0]} R_{j\ell m}^{[6]} = -E_{ij\ell m}^{[1]} R_{j\ell m}^{[5]}, \tag{98}$$

$$E_{ij\ell m}^{[0]} R_{j\ell m}^{[7]} = -16\pi t_{i\ell m}^{[7]} - E_{ij\ell m}^{[1]} R_{j\ell m}^{[6]} - E_{ij\ell m}^{[2]} R_{j\ell m}^{[5]}. \tag{99}$$

The first of these equations is equivalent to the field equation for $R_{i\ell m}^{(1)}$ in the inspiral (52a) evaluated at the ISCO frequency. Therefore,

$$R_{i\ell m}^{[5]}(\Delta J^a, r) = R_{i\ell m}^{[5]A}(\delta M^\pm, r) = R_{i\ell m}^{[5]Aa}(r)\delta M^a = R_{i\ell m}^{(1)}(\Omega_*, \delta M^\pm, r).$$

As a consequence, the right-hand side of eq. (98) vanishes, $E_{ij\ell m}^{[1]} R_{j\ell m}^{[5]} = 0$, leading to $R_{i\ell m}^{[6]}(\Delta J^a, r) = 0$. Using $t_{i\ell m}^{[7]} = \Delta\Omega\, t_{i\ell m}^{[7]A}$, eq. (99) reduces to

$$\begin{aligned} E_{ij\ell m}^{[0]} R_{j\ell m}^{[7]} &= -16\pi\Delta\Omega\, t_{i\ell m}^{[7]A} \\ &\quad + \Delta\Omega\left[\frac{\delta_{ij}}{4}\left[2m^2\Omega_*(1-H^2)+im\left(\partial_x H + 2H\partial_x\right)\right]+im\mathcal{M}_t^{ij}\right]R_{j\ell m}^{[5]}. \end{aligned} \tag{100}$$

We can therefore write $R_{i\ell m}^{[7]}(\Delta J^a, r) = \Delta\Omega R_{i\ell m}^{[7]A}(\delta M^\pm, r) = \Delta\Omega R_{i\ell m}^{[7]Aa}(\delta M^\pm, r)\delta M^a$. The $\Delta\Omega$ dependence then factors out and the mode amplitudes $R_{i\ell m}^{[7]A}$ solve a system of 10 coupled ordinary differential equations in the radius $r$ for each value of $\ell$ and $m$.

The field equations (31) up to 4PLT order (equivalently, up to $O(\lambda^9)$) take the form

$$E_{ij\ell m}^{[0]}R_{j\ell m}^{[5+n]} = -16\pi t_{i\ell m}^{[5+n]} - \sum_{k=1}^{n} E_{ij\ell m}^{[k]}R_{j\ell m}^{[5+n-k]}. \tag{101}$$

Given the structure of the sources, the mode amplitudes $R_{i\ell m}^{[n]}$ for $n = 8, 9$ reduce (as for $n = 7$) to sums of terms factored into $\Delta\Omega$-dependent and $\Delta\Omega$-independent pieces while still being linear in $\delta M^a$. Up to 4PLT order we summarize such decompositions as

$$R_{i\ell m}^{[5]}(\Delta J^a, r) = R_{i\ell m}^{[5]A}(\delta M^\pm, r) = R_{i\ell m}^{[5]Aa}(r)\delta M^a, \tag{102a}$$

$$R_{i\ell m}^{[6]}(\Delta J^a, r) = 0, \tag{102b}$$

$$R_{i\ell m}^{[7]}(\Delta J^a, r) = \Delta\Omega R_{i\ell m}^{[7]A}(\delta M^\pm, r) = \Delta\Omega R_{i\ell m}^{[7]Aa}(r)\delta M^a, \tag{102c}$$

$$R_{i\ell m}^{[8]}(\Delta J^a, r) = F_{[0]}^{\Delta\Omega}R_{i\ell m}^{[8]A}(\delta M^\pm, r) = F_{[0]}^{\Delta\Omega}R_{i\ell m}^{[8]Aa}(r)\delta M^a, \tag{102d}$$

$$R_{i\ell m}^{[9]}(\Delta J^a, r) = \Delta\Omega^2 R_{i\ell m}^{[9]A}(\delta M^\pm, r) + F_{[0]}^{\Delta\Omega}\partial_{\Delta\Omega}F_{[0]}^{\Delta\Omega}R_{i\ell m}^{[9]B}(\delta M^\pm, r)$$
$$= \left(\Delta\Omega^2 R_{i\ell m}^{[9]Aa}(r) + F_{[0]}^{\Delta\Omega}\partial_{\Delta\Omega}F_{[0]}^{\Delta\Omega}R_{i\ell m}^{[9]Ba}(r)\right)\delta M^a. \tag{102e}$$

As we have done for the inspiral, we deduce the structure of the $F_{[n]}^{\delta M^\pm}$ forcing terms from the horizon fluxes, which are quadratic in $\partial_t h_{\mu\nu}$. We obtain

$$F_{[0]}^{\delta M^\pm}(\Delta J^a) = F_{[0]ab}^{\delta M^\pm}\delta M^a \delta M^b, \tag{103a}$$

$$F_{[1]}^{\delta M^\pm}(\Delta J^a) = 0, \tag{103b}$$

$$F_{[2]}^{\delta M^\pm}(\Delta J^a) = \Delta\Omega F_{[2]Aab}^{\delta M^\pm}\delta M^a \delta M^b, \tag{103c}$$

where $F_{[0]ab}^{\delta M^\pm}$ and $F_{[2]Aab}^{\delta M^\pm}$ are given purely by numerical values.

Nonlinear sources appear in the field equations starting from order $\lambda^{10}$ (5PLT order), entering through the harmonic modes of the quadratic Einstein tensor defined in eq. (23). Substituting eq. (88), we find that the structure of its transition-to-plunge expansion reads

$$\widehat{\delta^2 G}_{i\ell m} = \delta^2 G_{i\ell m}^{[0]}[R^{[5]}, R^{[5]}] + \lambda^2\left(\delta^2 G_{i\ell m}^{[2]}[R^{[5]}, R^{[5]}] + 2\delta^2 G_{i\ell m}^{[0]}[R^{[5]}, R^{[7]}]\right) + O(\lambda^3). \tag{104}$$

This expansion originates from taking $t$ and $r$ derivatives of the metric perturbation in eq. (88) as prescribed by eq. (89). The term $\delta^2 G_{i\ell m}^{[1]}[R^{[5]}, R^{[5]}]$, which would appear at order $\lambda$, vanishes since $R_{i\ell m}^{[5]}$ does not depend on $\Delta\Omega$.

Finally, up to 7PLT order, the expanded field equations (31) for the residual fields read

$$E_{ij\ell m}^{[0]}R_{j\ell m}^{[10]\mathcal{R}} = 2\delta^2 G_{i\ell m}^{[0]}[R^{[5]}, R^{[5]}] - E_{ij\ell m}^{[0]}R_{j\ell m}^{[10]\mathcal{P}} - \sum_{k=1}^{5} E_{ij\ell m}^{[k]}R_{j\ell m}^{[10-k]}, \tag{105a}$$

$$E_{ij\ell m}^{[0]}R_{j\ell m}^{[11]\mathcal{R}} = -E_{ij\ell m}^{[0]}R_{j\ell m}^{[11]\mathcal{P}} - \sum_{k=1}^{6} E_{ij\ell m}^{[k]}R_{j\ell m}^{[11-k]}, \tag{105b}$$

$$E_{ij\ell m}^{[0]}R_{j\ell m}^{[12]\mathcal{R}} = 2\delta^2 G_{i\ell m}^{[2]}[R^{[5]}, R^{[5]}] + 4\delta^2 G_{i\ell m}^{[0]}[R^{[5]}, R^{[7]}] - E_{ij\ell m}^{[0]}R_{j\ell m}^{[12]\mathcal{P}} - \sum_{k=1}^{7} E_{ij\ell m}^{[k]}R_{j\ell m}^{[12-k]}, \tag{105c}$$

Each of these equations comprises of 10 coupled radial ordinary differential equations for each value of $\ell$ and $m$.

As we have seen above, at each $n$PLT order ($n \geq 0$) the mode amplitudes $R_{i\ell m}^{[5+n]}$ can be written as a sum of terms factored into $\Delta\Omega$-dependent and $\Delta\Omega$-independent pieces. Writing the field equations explicitly up to 7PLT order, we deduce the following structure:

$$
\begin{aligned}
R_{i\ell m}^{[10]}(r, \Delta J^a) = R_{i\ell m}^{[10]A} &+ \left(F_{[0]}^{\Delta\Omega}\right)^2 \left(\partial_{\Delta\Omega}^2 F_{[0]}^{\Delta\Omega}\right) R_{i\ell m}^{[10]B} + F_{[0]}^{\Delta\Omega} \left(\partial_{\Delta\Omega} F_{[0]}^{\Delta\Omega}\right)^2 R_{i\ell m}^{[10]C} \\
&+ \Delta\Omega\, F_{[0]}^{\Delta\Omega} R_{i\ell m}^{[10]D} + F_{[2]}^{\Delta\Omega} R_{i\ell m}^{[10]E},
\end{aligned}
\tag{106a}
$$

$$
\begin{aligned}
R_{i\ell m}^{[11]}(r, \Delta J^a) = F_{[3]}^{\Delta\Omega} R_{i\ell m}^{[11]A} &+ \left(\partial_{\Delta\Omega} F_{[0]}^{\Delta\Omega}\right) F_{[2]}^{\Delta\Omega} R_{i\ell m}^{[11]B} + F_{[0]}^{\Delta\Omega} \left(\partial_{\Delta\Omega} F_{[2]}^{\Delta\Omega}\right) R_{i\ell m}^{[11]C} \\
&+ \left(F_{[0]}^{\Delta\Omega}\right)^2 R_{i\ell m}^{[11]D} + \Delta\Omega\, F_{[0]}^{\Delta\Omega} \left(\partial_{\Delta\Omega} F_{[0]}^{\Delta\Omega}\right) R_{i\ell m}^{[11]E} + \Delta\Omega^3 R_{i\ell m}^{[11]F},
\end{aligned}
\tag{106b}
$$

$$
\begin{aligned}
R_{i\ell m}^{[12]}(r, \Delta J^a) = F_{[4]}^{\Delta\Omega} R_{i\ell m}^{[12]A} &+ \left(\partial_{\Delta\Omega} F_{[0]}^{\Delta\Omega}\right) F_{[3]}^{\Delta\Omega} R_{i\ell m}^{[12]B} + F_{[0]}^{\Delta\Omega} \left(\partial_{\Delta\Omega} F_{[3]}^{\Delta\Omega}\right) R_{i\ell m}^{[12]C} \\
&+ \left(F_{[0]}^{\Delta\Omega}\right)^2 \left(\partial_{\Delta\Omega} F_{[0]}^{\Delta\Omega}\right) R_{i\ell m}^{[12]D} + \left(\partial_{\Delta\Omega} F_{[0]}^{\Delta\Omega}\right)^2 F_{[2]}^{\Delta\Omega} R_{i\ell m}^{[12]E} \\
&+ F_{[0]}^{\Delta\Omega} \left(\partial_{\Delta\Omega}^2 F_{[0]}^{\Delta\Omega}\right) F_{[2]}^{\Delta\Omega} R_{i\ell m}^{[12]F} + F_{[0]}^{\Delta\Omega} \left(\partial_{\Delta\Omega} F_{[0]}^{\Delta\Omega}\right) \left(\partial_{\Delta\Omega} F_{[2]}^{\Delta\Omega}\right) R_{i\ell m}^{[12]G} \\
&+ \left(F_{[0]}^{\Delta\Omega}\right)^2 \left(\partial_{\Delta\Omega}^2 F_{[2]}^{\Delta\Omega}\right) R_{i\ell m}^{[12]H} + \Delta\Omega\, R_{i\ell m}^{[12]I} + \Delta\Omega^2 F_{[0]}^{\Delta\Omega} R_{i\ell m}^{[12]J} \\
&+ \Delta\Omega\, F_{[2]}^{\Delta\Omega} R_{i\ell m}^{[12]K} + \Delta\Omega\, F_{[0]}^{\Delta\Omega} \left(\partial_{\Delta\Omega} F_{[0]}^{\Delta\Omega}\right)^2 R_{i\ell m}^{[12]L} \\
&+ \Delta\Omega \left(F_{[0]}^{\Delta\Omega}\right)^2 \left(\partial_{\Delta\Omega}^2 F_{[0]}^{\Delta\Omega}\right) R_{i\ell m}^{[12]M}.
\end{aligned}
\tag{106c}
$$

All the terms on the right-hand side labelled with capital Latin letters do not depend on $\Delta\Omega$ and are only functions of $\delta M$, $\delta J$ and the radial field point $r$. Note that at 5PLT order the dependency on $F_{[0]}^{\delta M^\pm}$ is included in the term $R_{i\ell m}^{[10]A}(r)$, while at 7PLT order the dependency on $F_{[2]}^{\delta M^\pm}$ is included in the term $\Delta\Omega R_{i\ell m}^{[12]I}(r)$, as a consequence of eq. (103). Recalling eq. (13), we have $R_{i\ell m}^{[n]} = R_{i\ell m}^{[n]a}(\Delta\Omega, r)\delta M^a$ for $5 \leq n \leq 9$, while $R_{i\ell m}^{[n]} = R_{i\ell m}^{[n]ab}(\Delta\Omega, r)\delta M^a \delta M^b$ for $10 \leq n \leq 14$. Note that the equations of motion (86) can be used to simplify some of these expressions by substituting the $\partial_{\Delta\Omega}^2 F_{[0]}^{\Delta\Omega}$ and $\partial_{\Delta\Omega}^2 F_{[2]}^{\Delta\Omega}$ terms. However, the current form turns out to be more convenient when asymptotically matching with the inspiral fields in section 5.3 below.

## 4.3 Self-force

We now perform the transition-to-plunge expansion of the self-force (11). Using eqs. (75), (76), (77), (78) and (79), we obtain the transition-to-plunge expansions of the worldline (4) and the four-velocity (8) as

$$
z^\mu = z_* + \lambda^2 z_{[0]}^\mu + \lambda^3 z_{[1]}^\mu + O(\lambda^4) = \left(t, r_*, \frac{\pi}{2}, \phi_p\right) + \lambda^2 \left(0, r_{[0]}, 0, 0\right) + O(\lambda^4),
\tag{107}
$$

$$
\begin{aligned}
u^\mu = u_* + \lambda^2 u_{[0]}^\mu + \lambda^3 u_{[1]}^\mu + O(\lambda^4) = &\, (U_*, 0, 0, U_* \Omega_*) \\
&+ \lambda^2 \left(U_{[0]}, 0, 0, U_{[0]}\Omega_* + U_* \Delta\Omega\right) \\
&+ \lambda^3 \left(0, U_* F_{[0]}^{\Delta\Omega} \partial_{\Delta\Omega} r_{[0]}, 0, 0\right) + O(\lambda^4),
\end{aligned}
\tag{108}
$$

where we have already used the fact that $r_{[1]} = U_{[1]} = 0$. The residual piece of the metric perturbation is expanded as

$$
h_{\mu\nu}^{\mathcal{R}} = \sum_{n=5}^\infty \lambda^n \sum_{i\ell m} \frac{a_{i\ell}}{r} R_{i\ell m}^{[n]\mathcal{R}}(\Delta J^a, r) e^{-im\phi_p} Y_{\mu\nu}^{i\ell m}(r, \theta, \phi).
\tag{109}
$$

Substituting the expansions above into eq. (11), we obtain up to 3PLT order

$$f^{\mu}_{[5]}(\Delta J^a) = \frac{1}{2} g^{\mu\nu} h^{[5]\mathcal{R}}_{u_* u_*,\nu}, \tag{110a}$$

$$f^{\mu}_{[6]}(\Delta J^a) = 0, \tag{110b}$$

$$f^{\mu}_{[7]}(\Delta J^a) = \Delta\Omega f^{\mu}_{[7]A}(\delta M^{\pm}), \tag{110c}$$

$$f^{\mu}_{[8]}(\Delta J^a) = F^{\Delta\Omega}_{[0]} f^{\mu}_{[8]A}(\delta M^{\pm}), \tag{110d}$$

where, using the structure of the metric perturbations obtained in the previous subsection,

$$
\begin{aligned}
f^{\mu}_{[7]A}(\delta M^{\pm}) &:= \frac{1}{2} g^{\mu\nu} h^{[7]\mathcal{R}A}_{u_* u_*,\nu} - 12\sqrt{6} M^2 \left( \partial_r g^{\mu\nu} h^{[5]\mathcal{R}}_{u_* u_*,\nu} + g^{\mu\nu} h^{[5]\mathcal{R}}_{u_* u_*,\nu r} \right) \\
&\quad + g^{\mu\nu} u^{\alpha}_* \left( 4\sqrt{3} M h^{[5]\mathcal{R}}_{\alpha t,\nu} + 4\sqrt{3} M \Omega_* h^{[5]\mathcal{R}}_{\alpha\phi,\nu} + U_* h^{[5]\mathcal{R}}_{\alpha\phi,\nu} \right) + \frac{\delta^{\mu}_t}{2} g^{tt} h^{[5]\mathcal{R}}_{u_* u_*,\phi_p},
\end{aligned}
\tag{111a}
$$

$$
\begin{aligned}
f^{\mu}_{[8]A}(\delta M^{\pm}) &:= \frac{1}{2} g^{\mu\nu} h^{[8]\mathcal{R}A}_{u_* u_*,\nu} - P^{\mu\nu} U_* u^{\beta}_* h^{[7]\mathcal{R}A}_{\beta\nu} + \frac{1}{2} u^{\mu}_* U_* h^{[7]\mathcal{R}A}_{u_* u_*} + \frac{\delta^{\mu}_t}{2} g^{tt} h^{[7]\mathcal{R}A}_{u_* u_*} \\
&\quad + U_* \partial_{\Delta\Omega} r_{[0]} \left( g^{\mu\nu} h^{[5]\mathcal{R}}_{\alpha r,\nu} u^{\alpha}_* + 2 P^{\mu\nu} \Gamma^{\sigma}_{\alpha r} h^{[5]\mathcal{R}}_{\sigma\nu} u^{\alpha}_* \right. \\
&\quad \left. - g^{\mu\nu} h^{[5]\mathcal{R}}_{\beta\nu,r} u^{\beta}_* - \frac{1}{2} u^{\mu}_* h^{[5]\mathcal{R}}_{u_* u_*,r} \right).
\end{aligned}
\tag{111b}
$$

Any $t$ derivative in these final expanded expressions needs to be computed with the rule $\partial_t \mapsto \Omega_* \partial_{\phi_p}$, as the full expansion of the time derivative has already been applied. All fields are evaluated at the ISCO. Like for the inspiral, the forces only depend on the mode amplitudes $R^{[n]\mathcal{R}}_{i\ell m}$ and not on the oscillatory phase. The computational details are analogous to the inspiral. Recalling eq. (13) and following the same reasoning as below eq. (62), we deduce that the dissipative self-forces at 0PLT and 2PLT order do not depend on $\delta M$ and $\delta J$, while the conservative pieces are linear in $\delta M^a = (1, \delta M, \delta J)$. The 3PLT self-force is linear in $\delta M^a$ as well.

By comparing eqs. (110) and (111) with the corresponding results for the inspiral (61), we can anticipate some of the results that we will obtain from the asymptotic match of section 5.3. It is easy to see that $f^{\mu}_{[5]}$ is given by the first-order self-force in the inspiral (61a) evaluated at the ISCO, after identifying $h^{[5]}_{\mu\nu} = h^{(1)}_{\mu\nu}|_*$. The 2PLT term $f^{\mu}_{[7]}$ matches the linear term in the Taylor expansion of $f^{\mu}_{(1)}$ around the ISCO frequency, $f^{\mu}_{[7]} = \Delta\Omega \partial_{\Omega} f^{\mu}_{(1)}|_*$. This is true if we consider the matching condition $h^{[7]A}_{\mu\nu} = \partial_{\Omega} h^{(1)}_{\mu\nu}|_*$. Finally, by comparing $f^{\mu}_{[8]A}$ with $f^{\mu}_{(2)B}$ and anticipating that $h^{[8]A}_{\mu\nu} = h^{(2)B}_{\mu\nu}|_*$, we recognize that $f^{\mu}_{[8]A} = f^{\mu}_{(2)B}|_*$. The matching conditions for the metric perturbations that we have assumed to hold in order to derive these results are obtained in section 5.3 below.

At each perturbative order, the structure of the self-forces (110) follows one of the metric perturbations obtained in section 4.2. We therefore write the structure of the self-force up to 7PLT order as

$$f^{\mu}_{[9]}(\Delta J^a) = \Delta\Omega^2 f^{\mu}_{[9]A} + F^{\Delta\Omega}_{[0]} \left( \partial_{\Delta\Omega} F^{\Delta\Omega}_{[0]} \right) f^{\mu}_{[9]B}, \tag{112a}$$

$$
\begin{aligned}
f^{\mu}_{[10]}(\Delta J^a) &= f^{\mu}_{[10]A} + \left( F^{\Delta\Omega}_{[0]} \right)^2 \left( \partial^2_{\Delta\Omega} F^{\Delta\Omega}_{[0]} \right) f^{\mu}_{[10]B} + F^{\Delta\Omega}_{[0]} \left( \partial_{\Delta\Omega} F^{\Delta\Omega}_{[0]} \right)^2 f^{\mu}_{[10]C} \\
&\quad + \Delta\Omega F^{\Delta\Omega}_{[0]} f^{\mu}_{[10]D} + F^{\Delta\Omega}_{[2]} f^{\mu}_{[10]E},
\end{aligned}
\tag{112b}
$$

$$f^{\mu}_{[11]}(\Delta J^a) = F^{\Delta\Omega}_{[3]}f^{\mu}_{[11]A} + \left(\partial_{\Delta\Omega}F^{\Delta\Omega}_{[0]}\right)F^{\Delta\Omega}_{[2]}f^{\mu}_{[11]B} + F^{\Delta\Omega}_{[0]}\left(\partial_{\Delta\Omega}F^{\Delta\Omega}_{[2]}\right)f^{\mu}_{[11]C}$$
$$+ \left(F^{\Delta\Omega}_{[0]}\right)^2 f^{\mu}_{[11]D} + \Delta\Omega F^{\Delta\Omega}_{[0]}\left(\partial_{\Delta\Omega}F^{\Delta\Omega}_{[0]}\right)f^{\mu}_{[11]E} + \Delta\Omega^3 f^{\mu}_{[11]F}, \tag{112c}$$

$$f^{\mu}_{[12]}(\Delta J^a) = F^{\Delta\Omega}_{[4]}f^{\mu}_{[12]A} + \left(\partial_{\Delta\Omega}F^{\Delta\Omega}_{[0]}\right)F^{\Delta\Omega}_{[3]}f^{\mu}_{[12]B} + F^{\Delta\Omega}_{[0]}\left(\partial_{\Delta\Omega}F^{\Delta\Omega}_{[3]}\right)f^{\mu}_{[12]C}$$
$$+ \left(F^{\Delta\Omega}_{[0]}\right)^2\left(\partial_{\Delta\Omega}F^{\Delta\Omega}_{[0]}\right)f^{\mu}_{[12]D} + \left(\partial_{\Delta\Omega}F^{\Delta\Omega}_{[0]}\right)^2 F^{\Delta\Omega}_{[2]}f^{\mu}_{[12]E}$$
$$+ F^{\Delta\Omega}_{[0]}\left(\partial^2_{\Delta\Omega}F^{\Delta\Omega}_{[0]}\right)F^{\Delta\Omega}_{[2]}f^{\mu}_{[12]F} + F^{\Delta\Omega}_{[0]}\left(\partial_{\Delta\Omega}F^{\Delta\Omega}_{[0]}\right)\left(\partial_{\Delta\Omega}F^{\Delta\Omega}_{[2]}\right)f^{\mu}_{[12]G}$$
$$+ \left(F^{\Delta\Omega}_{[0]}\right)^2\left(\partial^2_{\Delta\Omega}F^{\Delta\Omega}_{[2]}\right)f^{\mu}_{[12]H} + \Delta\Omega f^{\mu}_{[12]I} + \Delta\Omega^2 F^{\Delta\Omega}_{[0]}f^{\mu}_{[12]J}$$
$$+ \Delta\Omega F^{\Delta\Omega}_{[2]}f^{\mu}_{[12]K} + \Delta\Omega F^{\Delta\Omega}_{[0]}\left(\partial_{\Delta\Omega}F^{\Delta\Omega}_{[0]}\right)^2 f^{\mu}_{[12]L}$$
$$+ \Delta\Omega\left(F^{\Delta\Omega}_{[0]}\right)^2\left(\partial^2_{\Delta\Omega}F^{\Delta\Omega}_{[0]}\right)f^{\mu}_{[12]M}. \tag{112d}$$

All the terms on the right-hand side labelled with capital Latin letters do not depend on $\Delta\Omega$ and are only functions of $\delta M$ and $\delta J$. Concerning the decomposition in $\delta M^a = (1, \delta M, \delta J)$, the dissipative self-force is linear in $\delta M^a$ up to 5PLT order, and quadratic up to 10PLT order. Similarly, the conservative self-force is linear in $\delta M^a$ up to 4PLT order, and quadratic up to 9PLT order. This difference is due to the fact that the dissipative self-force $f^{\mu}_{[n]\text{diss}}$ depends on $h^{[n]}_{\mu\nu}$ only through a $\phi_p$ derivative, which does not contain any $\delta M^{\pm}$ dependence (see the discussion below eq. (62)), and the (non-)linearity structure is given by the metric perturbations of order $n-1$ and lower.

## 4.4 Early-time solution: orbital motion

At early times, the transition-to-plunge motion is expected to asymptotically match with the inspiral's near-ISCO solution. The early-time limit is reached as $\Delta\Omega \to -\infty$. We substitute the structure of the self-force (110) and (112) and the $F^{\delta M}_{[n]}$ and $F^{\delta J}_{[n]}$ forcing terms (103) into eqs. (83) and (86) with the sources listed in appendix C.1. We find that the early-time solutions for the $F^{\Delta\Omega}_{[n]}$ forcing terms are consistent with the following series expansions:

$$\lambda^{3+n}F^{\Delta\Omega}_{[n]}(\Delta\Omega \to -\infty, \delta M^{\pm}) = \lambda^{3+n}\sum_{i=0}^{\infty} F^{(3+n,c^-_{[n]}-5i)}_{[n]}\Delta\Omega^{c^-_{[n]}-5i} \qquad \forall\, n \geq 0, \tag{113}$$

with $c^-_{[n]} := \frac{n-2}{2}$ for $n \geq 0$ even, and $c^-_{[n]} := \frac{n-7}{2}$ for $n \geq 3$ odd, recalling that $F^{\Delta\Omega}_{[1]} = 0$, and hence there is no $n = 1$ term. We have verified eq. (113) up to $n = 7$ and assume this structure holds to any $n$PLT order with $n > 7$. Explicitly,

$$\lambda^3 F^{\Delta\Omega}_{[0]}(\Delta\Omega \to -\infty) = \lambda^3\left[\frac{F^{(3,-1)}_{[0]}}{\Delta\Omega} + \frac{F^{(3,-6)}_{[0]}}{\Delta\Omega^6} + O(\Delta\Omega^{-11})\right], \tag{114}$$

$$\lambda^4 F^{\Delta\Omega}_{[1]}(\Delta\Omega \to -\infty, \delta M^{\pm}) = \lambda^4\left[\frac{0}{\Delta\Omega^3} + \frac{0}{\Delta\Omega^8} + O(\Delta\Omega^{-13})\right], \tag{115}$$

$$\lambda^5 F^{\Delta\Omega}_{[2]}(\Delta\Omega \to -\infty) = \lambda^5\left[F^{(5,0)}_{[2]} + \frac{F^{(5,-5)}_{[2]}}{\Delta\Omega^5} + O(\Delta\Omega^{-10})\right], \tag{116}$$

$$\lambda^6 F^{\Delta\Omega}_{[3]}(\Delta\Omega \to -\infty, \delta M^{\pm}) = \lambda^6\left[\frac{F^{(6,-2)}_{[3]}}{\Delta\Omega^2} + \frac{F^{(6,-7)}_{[3]}}{\Delta\Omega^7} + O(\Delta\Omega^{-12})\right], \tag{117}$$

where we have included the vanishing term $F_{[1]}^{\Delta\Omega}$ to clearly illustrate the alternating structure; cf. table I in [18]. This alternating pattern between even and odd orders in the early-time transition-to-plunge solutions was also found in [14]. In a similar manner, taking the $\Delta\Omega \to -\infty$ limit of eqs. (81b), (82b) and (C.2), we obtain the early-time behaviour of the $n$PLT corrections to the orbital radius,

$$\lambda^{2+n} r_{[n]}(\Delta\Omega \to -\infty, \delta M^{\pm}) = \lambda^{2+n} \sum_{i=0}^{\infty} r_{[n]}^{(2+n,d_{[n]}^{-}-5i)} \Delta\Omega^{d_{[n]}^{-}-5i} \qquad \forall n \geq 0, \tag{118}$$

where $d_{[n]}^{-} := \frac{n+2}{2}$ for $n \geq 0$ even, and $d_{[n]}^{-} := \frac{n-3}{2}$ for $n \geq 3$ odd. We have verified this up to $n = 8$. Again, since $r_{[1]} = 0$, the solution at 1PLT order is trivial. The coefficients $F_{[n]}^{(i,j)}$ and $r_{[n]}^{(i,j)}$ with $n \in \mathbb{N}$ and $i, j \in \mathbb{Z}$ appearing in the early-time transition-to-plunge solutions are labelled with the powers $i$ of $\lambda$ and $j$ of $\Delta\Omega$ at which they appear in the expansions and in general depend on $\delta M$ and $\delta J$. Some of these coefficients are given explicitly in appendix C.4.

### 4.5 Early-time solution: metric perturbation and self-force

We now compute the early-time behaviour of the metric perturbation mode amplitudes (eqs. (102) and (106)) and the self-force (eqs. (110) and (112)) by substituting the corresponding solutions for the forcing terms (113). We present the early-time solutions up to 5PLT order explicitly. It is straightforward to obtain the 6PLT and 7PLT solutions in the same manner.

The 0PLT, 1PLT and 2PLT solutions are trivial. At 3PLT order we get

$$R_{i\ell m}^{[8]} = \left[ \frac{F_{[0]}^{(3,-1)}}{\Delta\Omega} + \frac{F_{[0]}^{(3,-6)}}{\Delta\Omega^6} + O\left(\Delta\Omega^{-11}\right) \right] R_{i\ell m}^{[8]A} \tag{119}$$

for the metric perturbations and

$$f_{[8]}^{\mu} = \left[ \frac{F_{[0]}^{(3,-1)}}{\Delta\Omega} + \frac{F_{[0]}^{(3,-6)}}{\Delta\Omega^6} + O\left(\Delta\Omega^{-11}\right) \right] f_{[8]A}^{\mu} \tag{120}$$

for the self-force. The 4PLT solutions read

$$R_{i\ell m}^{[9]} = \Delta\Omega^2 R_{i\ell m}^{[9]A} + \left[ -\frac{\left(F_{[0]}^{(3,-1)}\right)^2}{\Delta\Omega^3} - 7\frac{F_{[0]}^{(3,-1)} F_{[0]}^{(3,-6)}}{\Delta\Omega^8} + O\left(\Delta\Omega^{-13}\right) \right] R_{i\ell m}^{[9]B}, \tag{121}$$

$$f_{[9]}^{\mu} = \Delta\Omega^2 f_{[9]A}^{\mu} + \left[ -\frac{\left(F_{[0]}^{(3,-1)}\right)^2}{\Delta\Omega^3} - 7\frac{F_{[0]}^{(3,-1)} F_{[0]}^{(3,-6)}}{\Delta\Omega^8} + O\left(\Delta\Omega^{-13}\right) \right] f_{[9]B}^{\mu}. \tag{122}$$

Finally, at 5PLT order we obtain

$$R_{i\ell m}^{[10]} = R_{i\ell m}^{[10]A} + \left[ 2\frac{\left(F_{[0]}^{(3,-1)}\right)^3}{\Delta\Omega^5} + O\left(\Delta\Omega^{-10}\right) \right] R_{i\ell m}^{[10]B} + \left[ \frac{\left(F_{[0]}^{(3,-1)}\right)^3}{\Delta\Omega^5} + O\left(\Delta\Omega^{-10}\right) \right] R_{i\ell m}^{[10]C}$$

$$+ \left[ F_{[0]}^{(3,-1)} + O\left(\Delta\Omega^{-5}\right) \right] R_{i\ell m}^{[10]D} + \left[ F_{[2]}^{(5,0)} + O\left(\Delta\Omega^{-5}\right) \right] R_{i\ell m}^{[10]E} \tag{123}$$

for the metric perturbations and

$$
f^{\mu}_{[10]} = f^{\mu}_{[10]A} + \left[ 2 \frac{\left(F^{(3,-1)}_{[0]}\right)^3}{\Delta\Omega^5} + O\left(\Delta\Omega^{-10}\right) \right] f^{\mu}_{[10]B} + \left[ \frac{\left(F^{(3,-1)}_{[0]}\right)^3}{\Delta\Omega^5} + O\left(\Delta\Omega^{-10}\right) \right] f^{\mu}_{[10]C}
$$
$$
+ \left[ F^{(3,-1)}_{[0]} + O\left(\Delta\Omega^{-5}\right) \right] f^{\mu}_{[10]D} + \left[ F^{(5,0)}_{[2]} + O\left(\Delta\Omega^{-5}\right) \right] f^{\mu}_{[10]E} \tag{124}
$$

for the self-force.

## 5 Asymptotic match between the inspiral and the transition to plunge

The inspiral and transition-to-plunge regimes overlap in a buffer region exterior to the ISCO, where $r_p > r_*$ and $\Omega < \Omega_*$. Since the two expansions are describing the same motion, they must agree in this overlapping region. In order to compare them, we have re-expanded the post-adiabatic expansion of the inspiral in the near-ISCO limit at fixed $\varepsilon$ in sections 3.4 and 3.5, and computed the early-time behaviour of the transition-to-plunge expansion in sections 4.4 and 4.5. In this section we perform the match between these asymptotic solutions in the buffer region. The asymptotic match of the orbital motion was obtained in [12, 14] using the slow-time formulation. Here we revisit the asymptotic match of the orbital motion and complete the asymptotic match by including the metric perturbation and the self-force. We furthermore discuss *composite solutions* that join the inspiral and transition-to-plunge regimes. The overlapping region where both the inspiral and the transition-to-plunge solutions are valid is described in terms of proper time as $-\lambda^{-5} \ll \tau - \tau_* \ll -\lambda^{-1}$ [14]. In terms of $\Delta\Omega$ we can reformulate this region as $-\lambda^{-2} \ll M\Delta\Omega \ll -\lambda^0$ or, equivalently,

$$
-\frac{\lambda^0}{M} \ll \Omega - \Omega_* \ll -\frac{\lambda^2}{M} . \tag{125}
$$

### 5.1 Orbital motion

The near-ISCO solution of the inspiral motion obtained in section 3.4 is consistent with the following expansions:

$$
r_p = 6M + \sum_{i=0}^{\infty} \lambda^{5i} \sum_{j=0}^{\infty} r^{(5i+2j-2p_{(i)}, j-p_{(i)})}_{(i)} \lambda^{2(j-p_{(i)})} \Delta\Omega^{j-p_{(i)}} , \tag{126}
$$

$$
\frac{d\Omega}{dt} = \sum_{i=0}^{\infty} \lambda^{5+5i} \sum_{j=0}^{\infty} F^{(5+5i+2j-2q_{(i)}, j-q_{(i)})}_{(i)} \lambda^{2(j-q_{(i)})} \Delta\Omega^{j-q_{(i)}} . \tag{127}
$$

The near-ISCO solution up to 2PA order takes exactly that form with $p_{(0)} = -1$, $p_{(1)} = 0$, $p_{(2)} = 3$ and $q_{(0)} = 1$, $q_{(1)} = 2$, $q_{(2)} = 6$. We conjecture that this pattern holds to any $n$PA order with appropriate numbers $p_{(n)}$ and $q_{(n)}$, $n \geq 3$.

The early-time transition-to-plunge solutions can be obtained by summing all contributions given in eqs. (118) and (113),

$$
r_p = 6M + \sum_{n=0}^{\infty} \lambda^{2+n} \sum_{m=0}^{\infty} r^{(2+n, d^-_{[n]}-5m)}_{[n]} \Delta\Omega^{d^-_{[n]}-5m} , \tag{128}
$$

$$
\frac{d\Omega}{dt} = \sum_{n=0}^{\infty} \lambda^{3+n} \sum_{m=0}^{\infty} F^{(3+n, c^-_{[n]}-5m)}_{[n]} \Delta\Omega^{c^-_{[n]}-5m} . \tag{129}
$$

The inspiral and transition-to-plunge solutions listed above can be matched in the overlapping region (125) exterior to the ISCO. We obtain the matching conditions by equating the coefficients of equal powers of $\lambda$ and $\Delta\Omega$, identifying $n = 5i + 2j - 2p_{(i)} - 2$ and $j - p_{(i)} = d^-_{[n]} - 5m$ for the match of the orbital radius and $n = 2 + 5i + 2j - 2q_{(i)}$ and $j - q_{(i)} = c^-_{[n]} - 5m$ for the match of $d\Omega/dt$,

$$r^{(5i+2j-2p_{(i)},\,j-p(i))}_{(i)} = r^{(5i+2j-2p_{(i)},\,j-p(i))}_{[5i+2j-2p_{(i)}-2]}, \tag{130}$$

$$F^{(5+5i+2j-2q_{(i)},\,j-q_{(i)})}_{(i)} = F^{(5+5i+2j-2q_{(i)},\,j-q_{(i)})}_{[2+5i+2j-2q_{(i)}]}. \tag{131}$$

In order to verify these matching conditions, the match of the self-force between the inspiral and the transition to plunge is required. In practice, one proceeds order by order (both in the $\lambda$ and the $\Delta\Omega$ expansion) for the orbital motion, the metric perturbation and the self-force together at the same time to obtain the matching conditions. For the sake of presentation, we will defer the matching of the metric perturbation and the self-force to section 5.3 below. We have explicitly verified eqs. (130) and (131) for all terms involved in the match between the inspiral up to 2PA order and the transition to plunge up to 7PLT order, using the coefficients listed in appendices B.2 and C.4. The structure of the asymptotic match between the inspiral and transition-to-plunge orbital motions is summarized in table 1: the coefficients in the 0PA near-ISCO solution are matched by the leading-order coefficients in the early-time solutions of the even ($2n$PLT, $n \geq 0$) transition-to-plunge orders; the coefficients in the 1PA near-ISCO solution are matched by the leading-order coefficients in the early-time solutions of the odd (($2n + 1$)PLT, $n \geq 0$) transition-to-plunge orders; the coefficients in the 2PA near-ISCO solution are matched by the first-subleading-order coefficients in the early-time solutions of the even transition-to-plunge orders and so forth. In what follows, we label the asymptotic coefficients with the inspiral and transition-to-plunge orders they originate from, that is, $F^{(i,j)}_{(m)}, F^{(i,j)}_{[n]} \to F^{(i,j)}_{(m)/[n]}$ for $i, j \in \mathbb{Z}$ and $m, n \in \mathbb{N}$.

## 5.2  0PA-2PLT and 1PA-7PLT composite solutions

The asymptotic match allows us to write composite solutions, which are valid in the domain $\Omega \leq \Omega_*$ (or, equivalently, $r_p \geq r_*$) and uniformly approximate the exact solution $d\Omega/dt = F^\Omega$ in that region. They are constructed, following standard practice in matched asymptotic expansions, by adding the inspiral and transition-to-plunge expansions truncated at some specific perturbative order and subtracting the common matching values, which would otherwise be counted twice. Similar composite solutions can also be written for any other orbital quantity. We label the composite solution with the highest inspiral and transition-to-plunge orders considered. Relevant composite solutions (because of their smoothness properties as explained below) are

$$F^\Omega_{\text{0PA-2PLT}}(\lambda, \Omega) = \lambda^5 F^\Omega_{(0)}(\Omega) + \lambda^3 F^{\Delta\Omega}_{[0]}\left(\frac{\Omega - \Omega_*}{\lambda^2}\right) + \lambda^5 F^{\Delta\Omega}_{[2]}\left(\frac{\Omega - \Omega_*}{\lambda^2}\right)$$

$$- \lambda^5 \left[\frac{F^{(3,-1)}_{(0)/[0]}}{\Omega - \Omega_*} + F^{(5,0)}_{(0)/[2]}\right], \tag{132}$$

and

$$F^\Omega_{\text{1PA-7PLT}}(\lambda, J^a) = \lambda^5 F^\Omega_{(0)}(\Omega) + \lambda^{10} F^\Omega_{(1)}(J^a) + \lambda^3 \sum_{n=0}^{7} \lambda^n F^{\Delta\Omega}_{[n]}\left(\frac{\Omega - \Omega_*}{\lambda^2}, \delta M^\pm\right)$$

$$- \lambda^5 \left[ \frac{F^{(3,-1)}_{(0)/[0]}}{\Omega - \Omega_*} + F^{(5,0)}_{(0)/[2]} + (\Omega - \Omega_*) F^{(7,1)}_{(0)/[4]} + (\Omega - \Omega_*)^2 F^{(9,2)}_{(0)/[6]} \right] \quad (133)$$

$$- \lambda^{10} \left[ \frac{F^{(6,-2)}_{(1)/[3]}}{(\Omega - \Omega_*)^2} + \frac{F^{(8,-1)}_{(1)/[5]}}{(\Omega - \Omega_*)} + F^{(10,0)}_{(1)/[7]} \right],$$

which neglect terms of order $\lambda^6$ (3PLT) and $\lambda^{11}$ (8PLT), respectively. Sufficiently near the ISCO, the subtracted terms cancel the inspiral terms, leaving the correct transition-to-plunge approximation; sufficiently far from the ISCO, the subtracted terms cancel the transition-to-plunge terms, leaving the correct inspiral approximation. In this way, the composite solutions join the inspiral and transition-to-plunge regimes without the need to switch from one approximation scheme to the other at some radius exterior to the ISCO. In practice, we would only need to switch at the ISCO between one such inspiral/transition-to-plunge composite solution and a (not defined in this paper) transition-to-plunge/plunge composite solution.

The behaviour of the composite solution $F^\Omega_{\text{1PA-7PLT}}$ can be summarized as follows: close to the ISCO, the 0PA and 1PA forcing terms are approximated by their near-ISCO solutions (64b)

Table 1: Visualization of the matching conditions (130) and (131) between inspiral and transition to plunge for the orbital radius $r_p$ and the rate of change $d\Omega/dt$.

| | 0PA | 1PA | 2PA | $\cdots$ |
|---|---|---|---|---|
| 0PLT | $r^{(2,1)}_{(0)} = r^{(2,1)}_{[0]}$<br>$F^{(3,-1)}_{(0)} = F^{(3,-1)}_{[0]}$ | — | $-$<br>$F^{(3,-6)}_{(2)} = F^{(3,-6)}_{[0]}$ | $\cdots$ |
| 1PLT | — | — | — | $\cdots$ |
| 2PLT | $r^{(4,2)}_{(0)} = r^{(4,2)}_{[2]}$<br>$F^{(5,0)}_{(0)} = F^{(5,0)}_{[2]}$ | — | $r^{(4,-3)}_{(2)} = r^{(4,-3)}_{[2]}$<br>$F^{(5,-5)}_{(2)} = F^{(5,-5)}_{[2]}$ | $\cdots$ |
| 3PLT | — | $r^{(5,0)}_{(1)} = r^{(5,0)}_{[3]}$<br>$F^{(6,-2)}_{(1)} = F^{(6,-2)}_{[3]}$ | — | $\cdots$ |
| 4PLT | $r^{(6,3)}_{(0)} = r^{(6,3)}_{[4]}$<br>$F^{(7,1)}_{(0)} = F^{(7,1)}_{[4]}$ | — | $r^{(6,-2)}_{(2)} = r^{(6,-2)}_{[4]}$<br>$F^{(7,-4)}_{(2)} = F^{(7,-4)}_{[4]}$ | $\cdots$ |
| 5PLT | — | $r^{(7,1)}_{(1)} = r^{(7,1)}_{[5]}$<br>$F^{(8,-1)}_{(1)} = F^{(8,-1)}_{[5]}$ | — | $\cdots$ |
| 6PLT | $r^{(8,4)}_{(0)} = r^{(8,4)}_{[6]}$<br>$F^{(9,2)}_{(0)} = F^{(9,2)}_{[6]}$ | — | $r^{(8,-1)}_{(2)} = r^{(8,-1)}_{[6]}$<br>$F^{(9,-3)}_{(2)} = F^{(9,-3)}_{[6]}$ | $\cdots$ |
| 7PLT | — | $r^{(9,2)}_{(1)} = r^{(9,2)}_{[7]}$<br>$F^{(10,0)}_{(1)} = F^{(10,0)}_{[7]}$ | — | $\cdots$ |
| $\vdots$ | $\vdots$ | $\vdots$ | $\vdots$ | $\ddots$ |

and (65b). The divergent and constant terms in those expansions are exactly cancelled by the subtracted terms in eq. (133), while the terms proportional to positive powers of $\Delta\Omega$ go to zero. The composite solution $F^\Omega_{\text{1PA-7PLT}}$ then reduces to the transition-to-plunge solution in the near-ISCO limit. Considering the transition-to-plunge motion up to 7PLT order is necessary and sufficient to obtain a solution that is regular at the ISCO, cancelling all divergent and constant terms in the 1PA inspiral (65b). For the purpose of extending a 0PA inspiral beyond the ISCO, only the 2PLT order is required; that is, we need to build $F^\Omega_{\text{0PA-2PLT}}$. Considering now the early-time limit ($\Delta\Omega \to -\infty$), the terms proportional to negative powers of $\Delta\Omega$ in the early-time transition-to-plunge solution (129) become negligible, while constant and divergent terms are again cancelled by the subtracted terms. At early times, the composite solution is then only given by the inspiral terms. With this construction, both the $F^\Omega_{\text{0PA-2PLT}}$ and $F^\Omega_{\text{1PA-7PLT}}$ composite solutions are $\mathcal{C}^0$ functions at the ISCO (and smooth elsewhere), ensuring that $\Omega$ is $\mathcal{C}^1$ and $\phi_p$ is $\mathcal{C}^2$ there. Higher differentiability can be obtained by adding further PLT orders. We display the behaviour of the 0PA-2PLT composite solution for two different mass ratios in figure 2.

A caveat to this approach is that the early-time limit $\Delta\Omega = (\Omega - \Omega_*)/\lambda^2 \to -\infty$ is formally a small-mass-ratio limit $\lambda \to 0$ since $\Omega \to 0$ in the early inspiral while $\Omega_* \simeq 0.068/M$ is finite. For a mass ratio sufficiently close to 1, at early times the transition-to-plunge part of the composite solution will become numerically comparable to the inspiral part, spoiling the numerical accuracy of the composite solution. To see this, note that the $\varepsilon\, F^{(3,-1)}_{(0)/[0]}/(\Omega - \Omega_*)$ and $\varepsilon\, F^{(5,0)}_{(0)/[2]}$ terms in the composite solution (132) will, by design, cancel the early-time contribution from the transition-to-plunge solution, but this cancellation is not exact: it leaves a residue of $\varepsilon^3 F^{(3,-6)}_{(2)/[0]}/(\Omega - \Omega_*)^6$, $\varepsilon^3 F^{(5,-5)}_{(2)/[2]}/(\Omega - \Omega_*)^5$, and further subleading terms from the early-time transition-to-plunge solutions (114) and (116), which for sufficiently large mass ratios become comparable with the inspiral term $\varepsilon\, F^\Omega_{(0)}$. We can obtain the value of $\varepsilon$ where this occurs as follows. Let us take as a benchmark the early inspiral at $r_{(0)} = r^{\text{early}}_{(0)} := 20M$ (equivalent to $\Omega = \Omega_{\text{early}} := 1/(40\sqrt{5}M)$) and require that the transition-to-plunge terms are smaller than the inspiral terms by 1%,

$$\left| \frac{F^{(3,-6)}_{(2)/[0]} \frac{\varepsilon^3}{(\Omega-\Omega_*)^6} + F^{(5,-5)}_{(2)/[2]} \frac{\varepsilon^3}{(\Omega-\Omega_*)^5}}{\varepsilon\, F^\Omega_{(0)}(\Omega)} \right|_{\Omega=\Omega_{\text{early}}} < 0.01 \,. \tag{134}$$

Using the explicit numerical values listed in appendix D.2, we find that this holds as long as $\varepsilon \lesssim 1/180$, which makes the 0PA-2PLT composite solution numerically inaccurate in the most interesting range of mass ratios for ground-based detectors. We have kept the leading-order residues of both $\lambda^3 F^{\Delta\Omega}_{[0]}$ and $\lambda^5 F^{\Delta\Omega}_{[2]}$ in the numerator of eq. (134), and not only the term $\propto 1/(\Omega-\Omega_*)^5$, which could naively be considered the dominant term at early times: since $\Omega$ monotonically increases from 0 to $\Omega_* \simeq 0.068/M$, it is actually always true (at least outside the ISCO) that $\left|1/(\Omega-\Omega_*)^6\right| > \left|1/(\Omega-\Omega_*)^5\right|$. We have excluded the subleading residues of order $\varepsilon^5$ and higher, which we could have considered without affecting our evaluation. In conclusion, the composite solution should not be trusted for intermediate mass ratios and comparable-mass systems. When comparing against NR simulations in section 6, we will limit our analysis to pure transition-to-plunge waveforms, leaving the methods of meshing the inspiral and transition-to-plunge approximations for future work.

Previously, we estimated the frequencies at which the inspiral approximation breaks down, in the sense that omitted terms become more important than included ones; those breakdown frequencies were given in eqs. (67) and (68). We now consider a different question. Rather than estimating how near to the ISCO we can trust the inspiral approximation, we consider how near to the ISCO we should be in order for the transition-to-plunge approximation to be

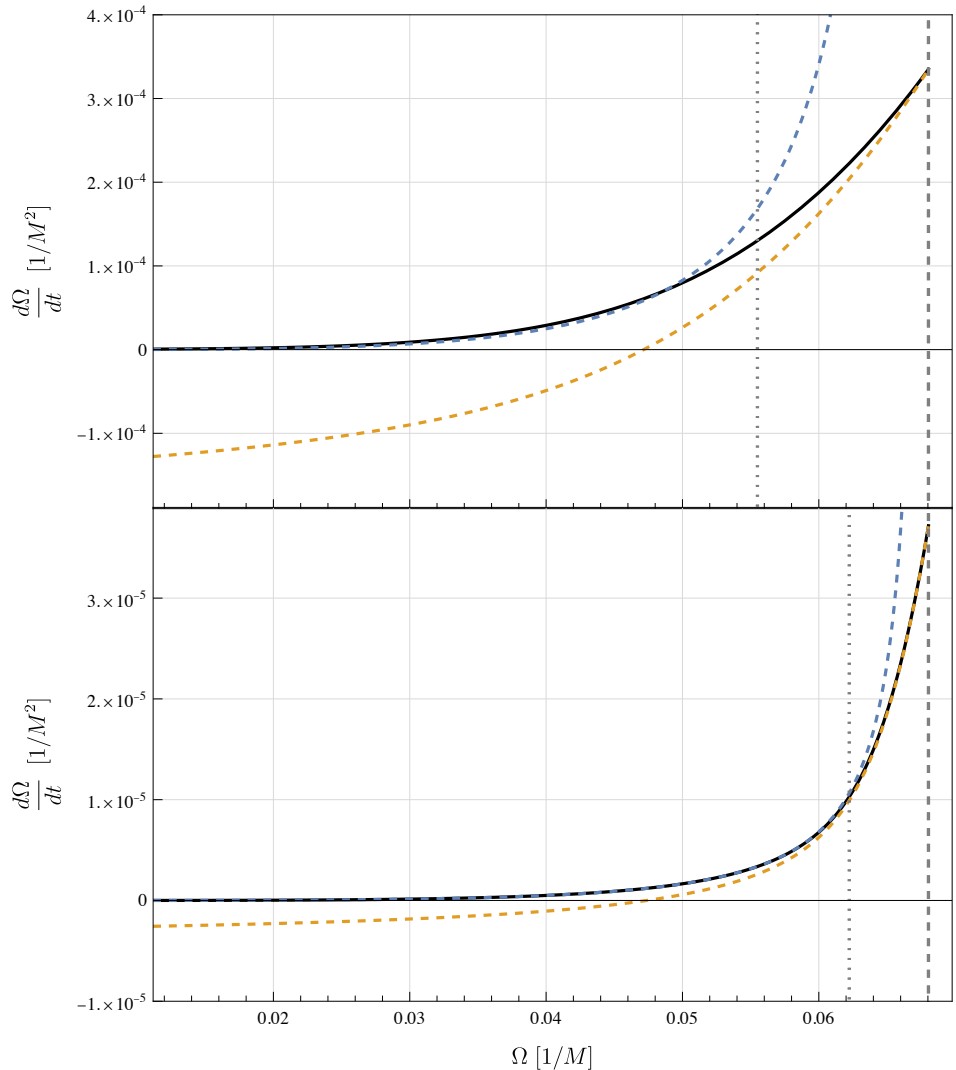

Figure 2: Rate of change of the orbital frequency from $\Omega = 1/(40\sqrt{5}M)$, corresponding to an orbital radius $r_{(0)} = 20M$, to $\Omega = \Omega_*$. The considered mass ratios are $\varepsilon = 1/10$ (top panel) and $\varepsilon = 1/500$ (bottom panel). The adiabatic rate of change, $\varepsilon F^{\Omega}_{(0)}$, given by the dashed blue curve, diverges at the ISCO frequency, marked by the vertical dashed gray line. The rate of change in the transition-to-plunge approximation to 2PLT order, $\lambda^3 F^{\Delta\Omega}_{[0]} + \lambda^5 F^{\Delta\Omega}_{[2]}$, is displayed as a dashed orange curve. The composite solution (132) in black reduces to the inspiral solution at small frequencies (this is only true in the lower panel, see the discussion around eq. (134)) and to the transition-to-plunge motion close to the ISCO. The vertical dotted gray line marks the lower boundary of the region in which the transition-to-plunge curve represents a better approximation than the inspiral curve. As expected, this region becomes narrower as the mass ratio decreases.

superior to the inspiral approximation. Concretely, we compute the critical frequency beyond which the 7PLT transition-to-plunge motion approximates the exact solution better than the 1PA inspiral motion: $\Omega^{\text{1PA-7PLT}}_{\text{crit}} = \Omega_* + \lambda^c \Delta\Omega_c + O(\lambda^{2c})$ such that $\Delta\Omega_c < 0$ and $0 < c < 2$ (which

is an intermediate scaling between the inspiral, $c = 0$, and the transition-to-plunge motion, $c = 2$). The behaviour of the inspiral and transition-to-plunge solutions in terms of the critical frequency is summarized below:

$$\Omega < \Omega_{\text{crit}}^{\text{1PA-7PLT}} : \quad \left| \lambda^5 F_{(0)}^{\Omega} + \lambda^{10} F_{(1)}^{\Omega} - F_{\text{1PA-7PLT}}^{\Omega} \right| < \left| \lambda^3 \sum_{n=0}^{7} \lambda^n F_{[n]}^{\Delta\Omega} - F_{\text{1PA-7PLT}}^{\Omega} \right|, \tag{135a}$$

$$\Omega > \Omega_{\text{crit}}^{\text{1PA-7PLT}} : \quad \left| \lambda^5 F_{(0)}^{\Omega} + \lambda^{10} F_{(1)}^{\Omega} - F_{\text{1PA-7PLT}}^{\Omega} \right| > \left| \lambda^3 \sum_{n=0}^{7} \lambda^n F_{[n]}^{\Delta\Omega} - F_{\text{1PA-7PLT}}^{\Omega} \right|, \tag{135b}$$

$$\Omega = \Omega_{\text{crit}}^{\text{1PA-7PLT}} : \quad \left| \lambda^5 F_{(0)}^{\Omega} + \lambda^{10} F_{(1)}^{\Omega} - F_{\text{1PA-7PLT}}^{\Omega} \right| = \left| \lambda^3 \sum_{n=0}^{7} \lambda^n F_{[n]}^{\Delta\Omega} - F_{\text{1PA-7PLT}}^{\Omega} \right|. \tag{135c}$$

We can obtain the critical exponent $c$ and the correction $\Delta\Omega_c$ by imposing the condition (135c). Since both the inspiral and transition-to-plunge solutions need to be simultaneously valid, the critical frequency lies in the matching region. We can therefore approximate the inspiral and transition-to-plunge solutions with their near-ISCO and early-time behaviours, respectively. At leading order, eq. (135c) then gives

$$\lambda^{15-6c} \left| \Delta\Omega_c^{-6} F_{(2)/[0]}^{(3,-6)} \right| = \lambda^{5+3c} \left| \Delta\Omega_c^3 F_{(0)/[8]}^{(11,3)} \right|. \tag{136}$$

Solving this equation for $c$ and $\Delta\Omega_c$ and recalling that $\lambda = \varepsilon^{1/5}$, we find the critical frequency

$$\Omega_{\text{crit}}^{\text{1PA}-7\text{PLT}} = \Omega_* - \varepsilon^{2/9} \left| \frac{F_{(2)/[0]}^{(3,-6)}}{F_{(0)/[8]}^{(11,3)}} \right|^{1/9} + O(\varepsilon^{4/9}). \tag{137}$$

The critical exponent is $c = 10/9$, lying within the chosen range $0 < c < 2$. If we instead consider a 0PA-2PLT motion, the critical frequency becomes

$$\Omega_{\text{crit}}^{\text{0PA}-2\text{PLT}} = \Omega_* - \varepsilon^{2/7} \left| \frac{F_{(2)/[0]}^{(3,-6)}}{F_{(0)/[4]}^{(7,1)}} \right|^{1/7} + O(\varepsilon^{4/7}). \tag{138}$$

We give the numerical values of the coefficients appearing in the expressions above in appendix D.2. Figure 3 shows the behaviour of the critical frequencies as functions of the mass ratio in the range where the the composite solution is a good approximation of the exact solution (see the discussion around eq. (134)) and can therefore be used in deriving the critical frequencies from eq. (135c). For all mass ratios, $\Omega_{\text{crit}}^{\text{1PA}-7\text{PLT}} < \Omega_{\text{crit}}^{\text{0PA}-2\text{PLT}} < \Omega_*$: as more perturbative terms are added to the transition-to-plunge motion the description becomes more accurate, extending its region of validity to smaller frequencies earlier in the inspiral. As the mass ratio increases, the region where the transition-to-plunge approximation is more accurate than the inspiral one becomes larger. This points to the fact that the transition-to-plunge approximation becomes crucial for modelling intermediate-mass-ratio and nearly comparable-mass binaries within self-force theory, indicating the importance of including transition-to-plunge effects over an increasingly large frequency interval for larger $\varepsilon$. This behaviour is already expected from the scaling around the ISCO of the orbital quantities such as the radius $r_p - r_* \sim \varepsilon^{2/5}$ and the frequency $\Omega - \Omega_* \sim \varepsilon^{2/5}$.

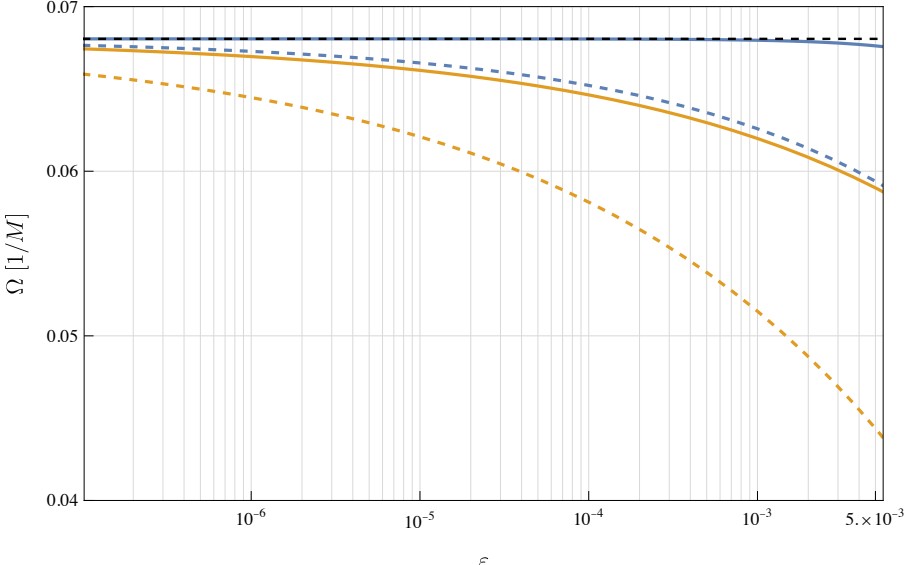

Figure 3: Critical frequencies as functions of the mass ratio for the 0PA-2PLT motion (eq. (138), dashed blue curve) and the 1PA-7PLT motion (eq. (137), dashed orange curve). The horizontal dashed line marks the ISCO frequency. The solid blue and orange curves show the 0PA and 1PA breakdown frequencies (67) and (68), respectively. The inspiral (resp. transition-to-plunge) approximation most accurately describes the motion below (resp. above) the dashed curves. We have restricted the plot to the range of mass ratios where the composite solution faithfully represents the exact solution, following the bound imposed by eq. (134). Importantly, the transition to plunge takes over before the inspiral approximation breaks down.

## 5.3 Metric perturbation and self-force

We now obtain the asymptotic match for the metric perturbation and the self-force. We can write the near-ISCO solution of the inspiral metric perturbation mode amplitudes by adding up eqs. (69), (71) and (73) in the post-adiabatic expansion,

$$
\begin{aligned}
\varepsilon R_{i\ell m}(\Omega \to \Omega_*, \delta M^\pm, r) = {} & \lambda^5 \left. R_{i\ell m}^{(1)} \right|_* + \lambda^7 \Delta\Omega \left. \partial_\Omega R_{i\ell m}^{(1)} \right|_* + \lambda^8 \frac{F_{(0)}^{(3,-1)}}{\Delta\Omega} \left. R_{i\ell m}^{(2)B} \right|_* \\
& + \lambda^9 \left[ \frac{1}{2}\Delta\Omega^2 \left. \partial_\Omega^2 R_{i\ell m}^{(1)} \right|_* - \frac{\left(F_{(0)}^{(3,-1)}\right)^2}{\Delta\Omega^3} \left. R_{i\ell m}^{(3)E} \right|_* \right] \\
& + \lambda^{10} \left[ \left. R_{i\ell m}^{(2)A} \right|_* + F_{(0)}^{(3,-1)} \left. \partial_\Omega R_{i\ell m}^{(2)B} \right|_* + F_{(0)}^{(5,0)} \left. R_{i\ell m}^{(2)B} \right|_* \right] \\
& + O_{\Delta\Omega}(\lambda^{11}) + O_\Omega(\varepsilon^4).
\end{aligned}
\tag{139}
$$

Here $O_\Omega(\varepsilon)$ and $O_{\Delta\Omega}(\varepsilon)$ (or, equivalently, $O_\Omega(\lambda)$ and $O_{\Delta\Omega}(\lambda)$) refer to the limit $\varepsilon \to 0$ in the inspiral expansion at fixed $\Omega$ and in the near-ISCO expansion at fixed $\Delta\Omega$, respectively. Combining the structure of the metric perturbations in the transition-to-plunge regime (eqs. (102) and (106)) with eqs. (119), (121) and (123), we find that the early-time transition-to-plunge

solution is given by

$$\varepsilon R_{i\ell m}(\Delta\Omega\to-\infty,\delta M^{\pm},r) = \lambda^5 R_{i\ell m}^{[5]} + \lambda^7 \Delta\Omega R_{i\ell m}^{[7]A} + \lambda^8 \left[ \frac{F_{[0]}^{(3,-1)}}{\Delta\Omega} + O\left(\Delta\Omega^{-6}\right) \right] R_{i\ell m}^{[8]A}$$

$$+ \lambda^9 \left[ \Delta\Omega^2 R_{i\ell m}^{[9]A} - \frac{\left(F_{[0]}^{(3,-1)}\right)^2}{\Delta\Omega^3} R_{i\ell m}^{[9]B} + O\left(\Delta\Omega^{-8}\right) \right] \tag{140}$$

$$+ \lambda^{10} \left[ R_{i\ell m}^{[10]A} + F_{[0]}^{(3,-1)} R_{i\ell m}^{[10]D} + F_{[2]}^{(5,0)} R_{i\ell m}^{[10]E} + O(\Delta\Omega^{-5}) \right]$$

$$+ O_{\Delta\Omega}(\lambda^{11}).$$

We recall that all the mode amplitudes on the right-hand side of these equations are functions of $\delta M$, $\delta J$ and $r$, while the $F_{(m)/[n]}^{(i,j)}$ coefficients in general depend on $\delta M$ and $\delta J$. Comparing the coefficients of equal powers of $\lambda$ and $\Delta\Omega$ in eqs. (139) and (140) and recalling the results of table 1, we obtain the following matching conditions for the metric perturbation mode amplitudes:

$$R_{i\ell m}^{[5]} = R_{i\ell m}^{(1)}\Big|_*, \tag{141a}$$

$$R_{i\ell m}^{[7]A} = \partial_\Omega R_{i\ell m}^{(1)}\Big|_*, \tag{141b}$$

$$R_{i\ell m}^{[8]A} = R_{i\ell m}^{(2)B}\Big|_*, \tag{141c}$$

$$R_{i\ell m}^{[9]A} = \frac{1}{2}\partial_\Omega^2 R_{i\ell m}^{(1)}\Big|_*, \qquad R_{i\ell m}^{[9]B}(r) = R_{i\ell m}^{(3)E}\Big|_*, \tag{141d}$$

$$R_{i\ell m}^{[10]A} = R_{i\ell m}^{(2)A}\Big|_*, \qquad R_{i\ell m}^{[10]D} = \partial_\Omega R_{i\ell m}^{(2)B}\Big|_*, \qquad R_{i\ell m}^{[10]E} = R_{i\ell m}^{(2)B}\Big|_*. \tag{141e}$$

We have also verified that the mode amplitudes involved in these matching conditions actually satisfy the same field equations. The subleading terms in the $\Delta\Omega\to-\infty$ expansions at each order in $\lambda$ in eq. (140) match with terms that originate from subleading post-adiabatic orders in eq. (139). In analogy to what we have done for the orbital motion, we can write a composite solution also for the mode amplitudes of the metric perturbation:

$$\varepsilon R_{i\ell m}^{1PA-5PLT} = \lambda^5 R_{i\ell m}^{(1)} + \lambda^{10} R_{i\ell m}^{(2)} + \lambda^5 \sum_{n=0}^{5} \lambda^n R_{i\ell m}^{[5+n]}$$

$$- \lambda^5 \left[ R_{i\ell m}^{(1)}\Big|_* + (\Omega-\Omega_*)\,\partial_\Omega R_{i\ell m}^{(1)}\Big|_* + \frac{(\Omega-\Omega_*)^2}{2}\,\partial_\Omega^2 R_{i\ell m}^{(1)}\Big|_* \right] \tag{142}$$

$$- \lambda^{10} \left[ \frac{F_{(0)/[0]}^{(3,-1)}}{\Omega-\Omega_*} R_{i\ell m}^{(2)B}\Big|_* + R_{i\ell m}^{(2)A}\Big|_* + F_{(0)/[0]}^{(3,-1)}\,\partial_\Omega R_{i\ell m}^{(2)B}\Big|_* + F_{(0)/[2]}^{(5,0)} R_{i\ell m}^{(2)B}\Big|_* \right].$$

This composite solution behaves analogously to the one in eq. (133), reducing to the inspiral and transition-to-plunge approximations in the early-time and near-ISCO limits, respectively. Considering the transition to plunge up to 5PLT order is necessary and sufficient to obtain a composite solution that is regular at the ISCO. Fewer transition-to-plunge terms are required compared to the composite solution (133) for the rate of change of the orbital frequency, which is due to the milder divergence close to the ISCO of the inspiral quantities here.

We now turn to the self-force. We obtain the near-ISCO solution of the inspiral self-force by appropriately summing the contributions in eqs. (70), (72) and (74),

$$
\begin{aligned}
f^\mu(\Omega \to \Omega_*, \delta M^\pm) = {} & \lambda^5 \left. f^\mu_{(1)}\right|_* + \lambda^7 \Delta\Omega \left. \partial_\Omega f^\mu_{(1)}\right|_* + \lambda^8 \frac{F^{(3,-1)}_{(0)}}{\Delta\Omega} \left. f^\mu_{(2)B}\right|_* \\
& + \lambda^9 \left[ \frac{1}{2}\Delta\Omega^2 \left. \partial^2_\Omega f^\mu_{(1)}\right|_* - \frac{\left(F^{(3,-1)}_{(0)}\right)^2}{\Delta\Omega^3} \left. f^\mu_{(3)E}\right|_* \right] \\
& + \lambda^{10} \left[ \left. f^\mu_{(2)A}\right|_* + F^{(3,-1)}_{(0)} \left. \partial_\Omega f^\mu_{(2)B}\right|_* + F^{(5,0)}_{(0)} \left. f^\mu_{(2)B}\right|_* \right] \\
& + O_{\Delta\Omega}(\lambda^{11}) + O_\Omega(\varepsilon^4).
\end{aligned}
$$

(143)

Using the structure of the self-force in the transition-to-plunge regime (eqs. (110) and (112)) together with eqs. (120), (122) and (124), we find that the early-time transition-to-plunge solution reads

$$
\begin{aligned}
f^\mu(\Delta\Omega \to -\infty, \delta M^\pm) = {} & \lambda^5 f^\mu_{[5]} + \lambda^7 \Delta\Omega f^\mu_{[7]A} + \lambda^8 \left[ \frac{F^{(3,-1)}_{[0]}}{\Delta\Omega} + O\left(\Delta\Omega^{-6}\right) \right] f^\mu_{[8]A} \\
& + \lambda^9 \left[ \Delta\Omega^2 f^\mu_{[9]A} - \frac{\left(F^{(3,-1)}_{[0]}\right)^2}{\Delta\Omega^3} f^\mu_{[9]B} + O\left(\Delta\Omega^{-8}\right) \right] \\
& + \lambda^{10} \left[ f^\mu_{[10]A} + F^{(3,-1)}_{[0]} f^\mu_{[10]D} + F^{(5,0)}_{[2]} f^\mu_{[10]E} + O(\Delta\Omega^{-5}) \right] \\
& + O_{\Delta\Omega}(\lambda^{11}).
\end{aligned}
$$

(144)

Comparing the coefficients of equal powers of $\lambda$ and $\Delta\Omega$ in eqs. (143) and (144) and recalling the results of table 1, we obtain the following matching conditions for the self-force:

$$
f^\mu_{[5]} = \left. f^\mu_{(1)}\right|_*,
$$

(145a)

$$
f^\mu_{[7]A} = \left. \partial_\Omega f^\mu_{(1)}\right|_*,
$$

(145b)

$$
f^\mu_{[8]A} = \left. f^\mu_{(2)B}\right|_*,
$$

(145c)

$$
f^\mu_{[9]A} = \frac{1}{2} \left. \partial^2_\Omega f^\mu_{(1)}\right|_*, \qquad f^\mu_{[9]B} = \left. f^\mu_{(3)E}\right|_*,
$$

(145d)

$$
f^\mu_{[10]A} = \left. f^\mu_{(2)A}\right|_*, \qquad f^\mu_{[10]D} = \left. \partial_\Omega f^\mu_{(2)B}\right|_*, \qquad f^\mu_{[10]E} = \left. f^\mu_{(2)B}\right|_*.
$$

(145e)

Again, the subleading terms in the $\Delta\Omega \to -\infty$ expansions at each order in $\lambda$ in eq. (144) match with terms that originate from subleading post-adiabatic orders in eq. (143). Considering the third-order self-force in deriving eq. (143) is, however, enough to determine the self-force matching conditions needed for the asymptotic match between the inspiral and the transition-to-plunge approximations truncated at 2PA and 7PLT order, respectively (see table 1). All relevant self-force matching conditions, including those for $f^\mu_{[11]}$ and $f^\mu_{[12]}$, are summarized in appendix D.1.

The matching conditions (141) are particularly useful since they allow us to determine some of the metric perturbations in the transition-to-plunge regime from inspiral quantities

without needing to solve any additional field equations. With the quasi-circular inspiral in Schwarzschild spacetime computed to 1PA order [8,52,53] (meaning the functions $R_{i\ell m}^{(1)}$, $R_{i\ell m}^{(2)A}$ and $R_{i\ell m}^{(2)B}$ are known), we can determine all transition-to-plunge mode amplitudes up to 5PLT order with the exception of $R_{i\ell m}^{[9]B}$, $R_{i\ell m}^{[10]B}$ and $R_{i\ell m}^{[10]C}$. In order to obtain these missing terms one needs to solve the field equations directly in the transition-to-plunge regime (or, equivalently, solve additional equations in the inspiral expansion and obtain the terms of interest through the asymptotic match). By virtue of the matching conditions (145), the same is true also for the self-force. As an example, let us consider the self-force in the transition-to-plunge regime through 3PLT order,

$$f^\mu = \lambda^5 f_{[5]}^\mu(\delta M^\pm) + \lambda^7 \Delta\Omega f_{[7]A}^\mu(\delta M^\pm) + \lambda^8 F_{[0]}^{\Delta\Omega} f_{[8]A}^\mu(\delta M^\pm) + O(\lambda^4). \tag{146}$$

The $\Delta\Omega$-independent quantities can be obtained from the matching conditions (145) rather than deriving them from the metric perturbations using the results in eqs. (110) and (111). The $\Delta\Omega$-dependent factors, which are in general combinations of the forcing terms $F_{[n]}^{\Delta\Omega}$ ($n \geq 0$) and their $\Delta J^a$ derivatives, can be obtained within the transition-to-plunge expansion by solving eq. (86).

# 6 2PLT motion and waveforms

Post-adiabatic waveforms for the inspiral regime have already been generated using the formalism reviewed herein [8]. We have now further developed the formalism to include the transition to plunge. Given the matching conditions between the inspiral and the transition to plunge, all numerical self-force data necessary to model waveforms up to second post-leading transition-to-plunge (2PLT) order is already available and can be readily computed using the Teukolsky package within the Black Hole Perturbation Toolkit (BHPToolkit) [54]. Further numerical work is required to extract the self-force data at 3PLT order and beyond. In this section, we will limit the production of explicit waveforms to 2PLT order.

## 6.1 Overview of the model

The 2PLT dynamics is governed by the following two ordinary differential equations:

$$\frac{d\Omega(t,\lambda)}{dt} = F_{2\text{PLT}}^\Omega(\lambda, \Omega(t,\lambda)), \tag{147}$$

$$\frac{d\phi_p(t,\lambda)}{dt} = \Omega(t,\lambda), \tag{148}$$

where the driving force is given by

$$F_{2\text{PLT}}^\Omega(\lambda, \Omega) := \lambda^3 F_{[0]}^{\Delta\Omega}\left(\frac{\Omega - \Omega_*}{\lambda^2}\right) + \lambda^5 F_{[2]}^{\Delta\Omega}\left(\frac{\Omega - \Omega_*}{\lambda^2}\right), \tag{149}$$

and is displayed in figure 4. We notice that $F_{2\text{PLT}}^\Omega \leq 0$ in some frequency range outside the ISCO. This breakdown at early times is specific to the 2PLT model and is possibly overcome as further PLT orders are added. We will limit our 2PLT model to the range of frequencies where $F_{2\text{PLT}}^\Omega > 0$ when producing waveforms. We also define the 0PLT driving force as

$$F_{0\text{PLT}}^\Omega(\lambda, \Omega) := \lambda^3 F_{[0]}^{\Delta\Omega}\left(\frac{\Omega - \Omega_*}{\lambda^2}\right). \tag{150}$$

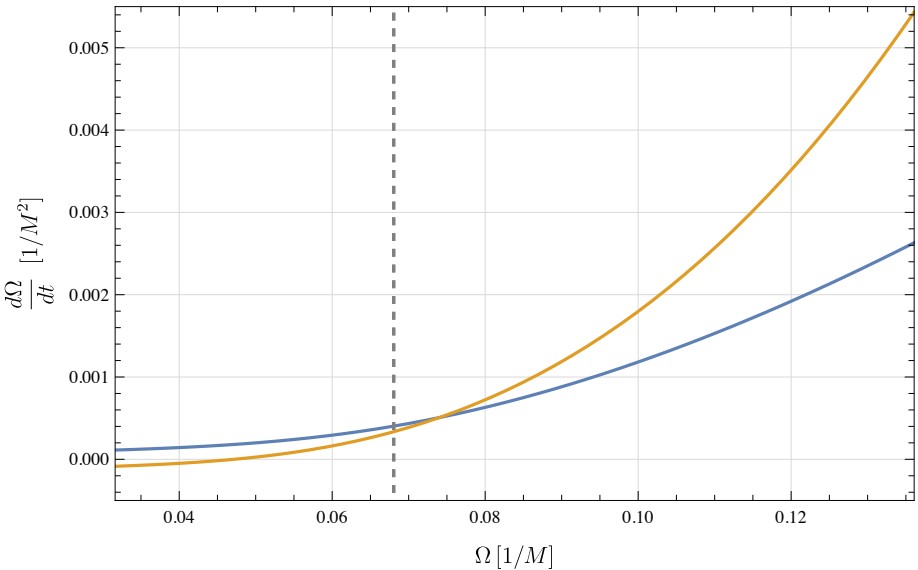

Figure 4: Rate of change of the orbital frequency from $\Omega = 1/(10\sqrt{10}M)$ to the light-ring frequency $\Omega_{\mathrm{LR}} := 1/(3\sqrt{6}M)$. The vertical dashed line marks the ISCO frequency. The considered mass ratio is $\varepsilon = 1/10$. The rate of change to 2PLT order (149) is displayed in orange. For reference, we also plot the 0PLT rate of change (150) in blue.

The transition-to-plunge driving forces $F_{[0]}^{\Delta\Omega}$ and $F_{[2]}^{\Delta\Omega}$ are expressed in terms of the self-force in eqs. (83) and (86) (with the source (C.3)). By virtue of those equations, in order to evolve the orbital frequency in eq. (147), the only required inputs are the self-force terms $f_{[5]}^t$ and $f_{[7]}^t = \Delta\Omega f_{[7]A}^t$. These can be obtained from the first-order inspiral self-force and its derivative at the ISCO, $f_{(1)}^t(\Omega_*)$ and $\partial_\Omega f_{(1)}^t(\Omega_*)$, using the matching conditions (145a) and (145b). Standard balance laws [9,55] allow us to compute the force $f_{(1)}^t(\Omega)$, in turn, from the 0PA energy flux to infinity ($\mathcal{F}_{0\mathrm{PA}}^\infty$) and down the black hole horizon ($\mathcal{F}_{0\mathrm{PA}}^{\mathcal{H}}$) as

$$f_{(1)}^t(\Omega) = -\frac{U_{(0)}(\Omega)}{f(r_{(0)}(\Omega))}\left[\mathcal{F}_{0\mathrm{PA}}^\infty(\Omega) + \mathcal{F}_{0\mathrm{PA}}^{\mathcal{H}}(\Omega)\right]. \tag{151}$$

The fluxes are readily obtained using the BHPToolkit's Teukolsky package. We list the numerical values of $f_{[5]}^t$ and $f_{[7]A}^t$ in appendix D.2.

Given a phase-space trajectory $(\Omega(t,\lambda), \phi_p(t,\lambda))$, the waveform is obtained from the metric perturbation at future null infinity. Analogously to the motion, for the metric perturbation we write

$$h_{\mu\nu}(\lambda, \Omega, \phi_p, x^i) = \sum_{i\ell m}\frac{a_{i\ell}}{r}R_{i\ell m}^{2\mathrm{PLT}}(\lambda, \Omega, r)e^{-im\phi_p}Y_{\mu\nu}^{i\ell m}(x^i), \tag{152}$$

where the 2PLT mode amplitudes are given by

$$R_{i\ell m}^{2\mathrm{PLT}}(\lambda, \Omega, r) := \lambda^5 R_{i\ell m}^{[5]} + \lambda^7 \Delta\Omega R_{i\ell m}^{[7]A} = \lambda^5\left[R_{i\ell m}^{(1)}\Big|_* + (\Omega - \Omega_*)\,\partial_\Omega R_{i\ell m}^{(1)}\Big|_*\right]. \tag{153}$$

In order to derive these expressions we have used the matching conditions (141a) and (141b). Note that as a first approximation we have set $\delta M = \delta J = 0$. Their contribution is numerically subdominant during the inspiral [8,18] and it is safe to assume this will also be the case during the transition to plunge since the rate of change of $\delta M$ and $\delta J$ remains of order $\lambda^5$. The strain

is expressed in terms of the two GW polarizations as the limit $r \to \infty$ of the expression

$$r(h_+ - ih_\times) = r h_{\mu\nu} \bar{m}^\mu \bar{m}^\nu = \sum_{\ell \geq 2} \sum_{m=-\ell}^{\ell} r h_{\bar{m}\bar{m}}^{\ell m} {}_{-2}Y_{\ell m}(\theta, \phi), \tag{154}$$

where $\bar{m} = \frac{1}{\sqrt{2}}(0, 0, 1, -i\csc\theta)$ and ${}_{-2}Y_{\ell m}$ is a spin-weighted spherical harmonic. For convenience, we define $h_{\ell m} := \lim_{r\to\infty}(r h_{\bar{m}\bar{m}}^{\ell m})$ and write the asymptotic $\ell m$ mode of the waveform as

$$h_{\ell m}(\varepsilon, \Omega, \phi_p) = H_{\ell m}(\varepsilon, \Omega)e^{-im\phi_p}. \tag{155}$$

In terms of the Barack-Lousto-Sago coefficients, the complex amplitude $H_{\ell m}$ is given by [4]

$$H_{\ell m} = \frac{1}{2\sqrt{(\ell+2)(\ell+1)\ell(\ell-1)}}[R_{7\ell m}(\varepsilon, \Omega, \infty) + i R_{10\ell m}(\varepsilon, \Omega, \infty)]. \tag{156}$$

We can compute the numerical inputs for $H_{\ell m}$ from the outputs of the BHPToolkit's Teukolsky package, just as we did for the 2PLT driving forces. We consider here only the oscillatory modes ($m \neq 0$); quasistationary modes ($m = 0$) are related to the displacement memory effect and require a separate treatment. Using the relationships between 0PA Teukolsky amplitudes and 0PA metric perturbation amplitudes at infinity, as given in eq. (420a) of [4], we write

$$H_{\ell m} = H_{\ell m}^{2\mathrm{PLT}}(\varepsilon, \Omega) = 2\varepsilon \left\{ \left.\frac{{}_{-2}C_{\ell m}^{\mathrm{up}}(\Omega)}{(m\Omega)^2}\right|_* + (\Omega - \Omega_*)\,\partial_\Omega \left(\left.\frac{{}_{-2}C_{\ell m}^{\mathrm{up}}(\Omega)}{(m\Omega)^2}\right)\right|_* \right\}. \tag{157}$$

Here ${}_{-2}C_{\ell m}^{\mathrm{up}}(\Omega)$ are the mode amplitudes that are output by the Teukolsky package after expressing $\Omega$ as a function of the radius $r_{(0)}$ using eq. (37). Note that to 2PLT order, the amplitude $H_{\ell m}^{2\mathrm{PLT}}$ coincides with the first two terms in a Taylor series expansion of $H_{\ell m}^{0\mathrm{PA}}$ around the ISCO frequency.

In summary, the 2PLT waveform is given by eq. (155) with (157), where $\phi_p$ and $\Omega$ are solutions to eqs. (147) and (148). Recall that because of our choice of hyperboloidal slicing, time $t$ along the worldline is identified with retarded time $u$ along future null infinity; we can hence simply replace $t$ with $u$ in eqs. (147) and (148). Also recall that, as stressed earlier in this paper, a key advantage of the multiscale formulation is that it enables rapid waveform generation by dividing the problem into a (slow) offline step and a (fast) online step. Here the offline step consists of computing all the necessary functions of $\Omega$ (driving forces and waveform amplitudes).[2] The online step consists of solving eqs. (147) and (148) and substituting the result into eq. (155).[3] Concretely, for our 2PLT waveforms, in the offline stage we use the BHPToolkit's Teukolsky package to first compute the amplitudes ${}_{-2}C_{\ell m}^{\mathrm{up}}$ and their associated GW fluxes $\mathcal{F}_{0\mathrm{PA}}^{\infty/\mathcal{H}}$ as functions of $\Omega$. Since these are 0PA inspiral quantities, they are identical to the amplitudes and fluxes from geodesic circular orbits of frequency $\Omega$, which are provided directly as outputs of the Teukolsky package's function `TeukolskyPointParticleMode`. We use `TeukolskyPointParticleMode` to compute the amplitudes and fluxes on a densely sampled grid around the ISCO, for a set of spherical harmonic modes $\ell = 2, \ldots, \ell_{\max}$, $m = -\ell, \ldots, \ell$. We include contributions up to $\ell_{\max} = 30$ in the calculation of the 0PA fluxes. This data is then stored as interpolating functions of $\Omega$ (or of orbital radius $r_{(0)}(\Omega)$). For the fluxes, only

---

[2]In practice, the field equations are solved in units with $M = 1$, and outputs are stored in those units. Equivalently, stored functions of $\Omega$ are actually functions of the dimensionless quantity $M\Omega$ or $r_{(0)}/M$. This is relevant when comparing to numerical relativity simulations, for example, which are in units with $m_p + M = 1$.

[3]For data analysis purposes, the online step would also involve summing over modes of the waveform. For generic orbits, this is the slowest part of the online step [7], but it is not a major consideration for the quasicircular orbits we consider here.

the total fluxes (summed over $\ell$ and $m$) are interpolated. For the waveform amplitudes, we interpolate each mode up to $\ell = 5, m = 5$; the higher-$\ell$ modes contribute very little to the amplitude (even though, through the flux, they have a significant cumulative impact on the waveform phase). As explained above eq. (151), all the driving forces appearing in eq. (147) can be obtained from the fluxes $\mathcal{F}_{0PA}^{\infty/\mathcal{H}}$. The 0PLT and 2PLT driving forces are obtained from the fluxes by solving the differential equations (83) and (86) (with source (C.3)); we solve these in advance and store the solutions as interpolating functions of $\Delta\Omega$. We stress that the entirety of the offline stage is completed without involving $\phi_p$ and without specifying a mass ratio or initial conditions. In the online step, we solve eq. (147) after specifying $\varepsilon$, $\Omega(0)$, and $\phi_p(0)$. These choices are detailed in the following subsections.

## 6.2 Qualitative comparison with numerical relativity waveforms

In this subsection we construct waveform templates using the procedure we have just described. We then compare our waveforms with those of NR simulations from the SXS catalogue [31] with large mass ratio $q := 1/\varepsilon = M/m_p$ ranging from $q = 1$ to $q = 10$. We have chosen simulations of binary black hole mergers with low eccentricity ($\lesssim 10^{-3}$) and small dimensionless spins of the individual black holes ($\lesssim 10^{-7}$). In view of these comparisons, we re-expand the relevant quantities for our waveform generation in powers of the symmetric mass ratio $\nu := Mm_p/(M+m_p)^2 = \varepsilon/(1+\varepsilon)^2$ at fixed total mass $M_{\text{tot}} := M + m_p = M(1+\varepsilon)$. Inverting the relation between $\nu$ and $\varepsilon$ gives $\varepsilon = (1-2\nu-\sqrt{1-4\nu})/(2\nu)$, which, in the small-mass-ratio expansion, leads to $\varepsilon = \nu + 2\nu^2 + O(\nu^3)$. At the orders we are interested in we can simply substitute $\varepsilon \to \nu$ and $M \to M_{\text{tot}}$. Functions of $\Omega$ stored in units with $M = 1$ can then be used without change and interpreted as being in units with $M_{\text{tot}} = 1$.

We consider two different scenarios for our self-force waveforms. We list them below, together with the details on the metric perturbation and the evolution equations for the orbital frequency and phase. In both cases, $\phi_p$ is obtained from $d\phi_p/dt = \Omega$, and $\Delta\Omega = (\Omega-\Omega_*)/\nu^{2/5}$. We define $H_{\ell m}^{[n]}$ from $R_{i\ell m}^{[n]}$ via eq. (156).

- **Model: 0PLT.** The leading-order transition-to-plunge approximation, but including the 2PLT amplitude correction (the 0PLT amplitude would only be given by a constant, $\nu H_{\ell m}^{[5]}$):

$$\frac{d\Omega}{dt} = \nu^{3/5} F_{[0]}^{\Delta\Omega}(\Delta\Omega), \qquad h_{\ell m} = \nu\left[H_{\ell m}^{[5]} + \nu^{2/5} H_{\ell m}^{[7]}(\Delta\Omega)\right]e^{-im\phi_p}. \qquad (158)$$

- **Model: 2PLT.** The transition to plunge through 2PLT order:

$$\frac{d\Omega}{dt} = \nu^{3/5} F_{[0]}^{\Delta\Omega}(\Delta\Omega) + \nu F_{[2]}^{\Delta\Omega}(\Delta\Omega), \qquad h_{\ell m} = \nu\left[H_{\ell m}^{[5]} + \nu^{2/5} H_{\ell m}^{[7]}(\Delta\Omega)\right]e^{-im\phi_p}. \qquad (159)$$

We compute the evolution of the orbital frequency and phase in the two scenarios described above. We start at the ISCO frequency $\Omega(t = 0) = 1/(6\sqrt{6}M_{\text{tot}})$ (and choose $\phi_p(t = 0) = 0$) and integrate eqs. (158) and (159) backward and forward in time until, respectively, an initial frequency $\Omega_i \approx 0.49/M_{\text{tot}}$ and a final frequency at the light-ring, $\Omega_f = 1/(3\sqrt{6}M_{\text{tot}})$. We have chosen the lower bound such that, for all the considered mass ratios, the rate of change of the orbital frequency to 2PLT order remains positive (see figure 4). Physically, the orbital frequency increases until the light ring where it starts to decrease before vanishing at the horizon. This does not happen in our transition-to-plunge model, and including the final plunge is necessary to correctly capture the dynamics in this late stage. We therefore cut off the integration at the light ring and leave the modelling of the plunge and its hybridization with the transition-to-plunge expansion to future work.

In figures 5 to 9 we compare the self-force waveforms to the chosen NR simulations for the dominant $(\ell, m) = (2, 2)$ mode, which we write as $h_{22} = |h_{22}| e^{-i\Phi_{22}}$, where $\Phi_{22} := -\arg(h_{22})$. For each comparison, we align the two waveforms in phase at the ISCO, that is, at the NR time $t_*$ such that the waveform frequency defined as $\omega_{22} := 1/2 \, d\Phi_{22}/dt$ is $\omega_{22}(t_*) = 1/(6\sqrt{6}M_{\text{tot}})$. We notice that to the left of the ISCO the 2PLT model covers a much larger range of the NR waveform compared to the 0PLT model, while the opposite is true to the right of the ISCO. We can explain this feature by looking at figure 4, which displays the rate of change of the orbital frequency for the specific case of $q = 10$ (but the general behaviour remains valid also for other mass ratios): to the left of the ISCO the frequency increases more rapidly in the 0PLT model, meaning the lower bound $\Omega_{\text{init}}$ is reached earlier. This trend is inverted to the right of the ISCO, where the 2PLT rate of change is much larger than the 0PLT one. In the region where the 0PLT and 2PLT waveforms overlap, the 2PLT model performs significantly better in the comparison with the NR simulations.

During the transition to plunge, the phase admits an expansion of the form

$$\phi_p = \frac{1}{\lambda} \left[ \phi_p^{[0]}(\Delta\Omega) + \lambda^2 \phi_p^{[2]}(\Delta\Omega) + O(\lambda^3) \right], \tag{160}$$

consistently with eqs. (5) and (78). The 2PLT model reduces the orbital phase error from $O(\lambda)$ to $O(\lambda^2)$. From eq. (160) we also deduce that on a fixed-$\Delta\Omega$ interval the total accumulated phase (like the total elapsed time) scales as $1/\lambda$, increasing with $q$. We have highlighted such a fixed-$\Delta\Omega$ interval in figures 5 to 9 with blue and orange colored regions, which contain an increasing portion of phase as $q$ increases.

These results, although preliminary, are promising: given the improvement between the 0PLT and 2PLT models, we expect the comparison with NR to markedly improve as we proceed to significantly higher PLT orders. The orders considered so far only included dissipative 0PA information, while starting from 3PLT order the model will also include post-adiabatic effects. Furthermore, we expect accuracy to significantly improve once we incorporate the final plunge, given that the peak merger amplitude occurs roughly when the particle crosses the light ring at $r = 3M_{\text{tot}}$ [10], which occurs at a frequency outside the expected domain of validity of our transition-to-plunge expansion.

## 6.3 Quantitative comparisons with numerical relativity and surrogate waveforms

In the previous subsection we qualitatively compared our model with NR waveforms for quasi-circular and non-spinning black hole binaries with different mass ratios for the $(l, m) = (2, 2)$ mode. In this section we quantitatively assess the accuracy of our model's underlying ingredients (its amplitudes and the dynamics that determines its phasing) as functions of frequency and mass ratio.

We again compare to SXS simulations, but we now also consider the surrogate model BHPTNRSur1dq1e4 [32, 33]. This model is built from a 0PA inspiral, a "generalized" leading-order (Ori-Thorne) transition to plunge [34], and a geodesic plunge. We discuss the dynamics and waveform-generation mechanism of this model in the next section, but here we only note that it includes both a transition and a plunge. The full BHPTNRSur1dq1e4 model includes calibration parameters that allow it to mimic NR waveforms in the comparable-mass regime, but to better assess our own sources of error we compare only to the uncalibrated BHPTNR-Sur1dq1e4 model. We specifically opt to compare against this model because its construction and expected accuracy are most similar to our own (since it is based on a 0PA inspiral and a transition-to-plunge model), while other inspiral-merger-ringdown models are largely indistinguishable from NR for the systems and mass ratios we consider here. When comparing to SXS simulations, we work in units of $M_{\text{tot}} = 1$ (the units used in the SXS output data) and

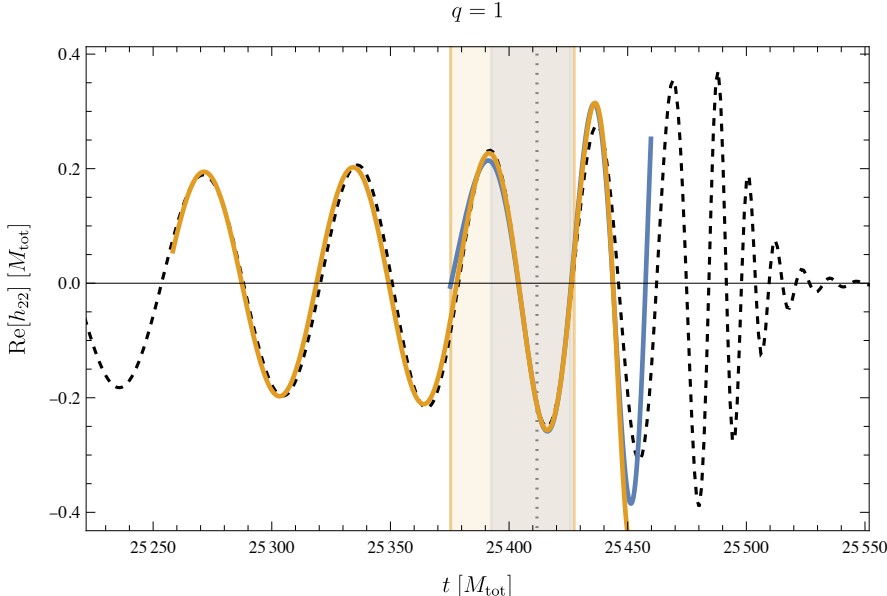

Figure 5: Gravitational waveforms ($(\ell, m) = (2, 2)$ mode) for a non-spinning compact binary with mass ratio $q = 1$. We compare the waveform corresponding to our 0PLT and 2PLT models (displayed as blue and orange curves, respectively) with the NR simulation SXS:BBH:1132 [56] of the same binary (plotted as a dashed black curve). The waveforms are aligned in frequency and phase at the vertical dotted gray line, where $\omega_{22} = 1/(6\sqrt{6}M_{\text{tot}})$. The blue and orange colored regions cover a portion of, respectively, the 0PLT and 2PLT self-force waveform that corresponds to an interval of fixed $\Delta\Omega$ of width $0.04/M_{\text{tot}}$. This interval was chosen such that the phase has the expected behaviour with increasing $q$, far enough from the 2PLT rate of change becoming negative.

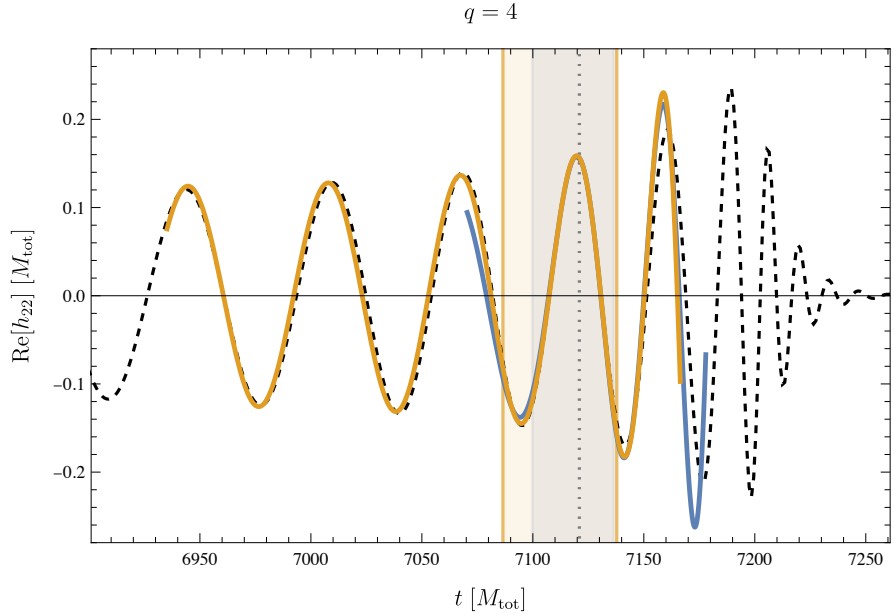

Figure 6: Same as figure 5 for a mass ratio $q = 4$. The NR simulation is SXS:BBH:1220 [57].

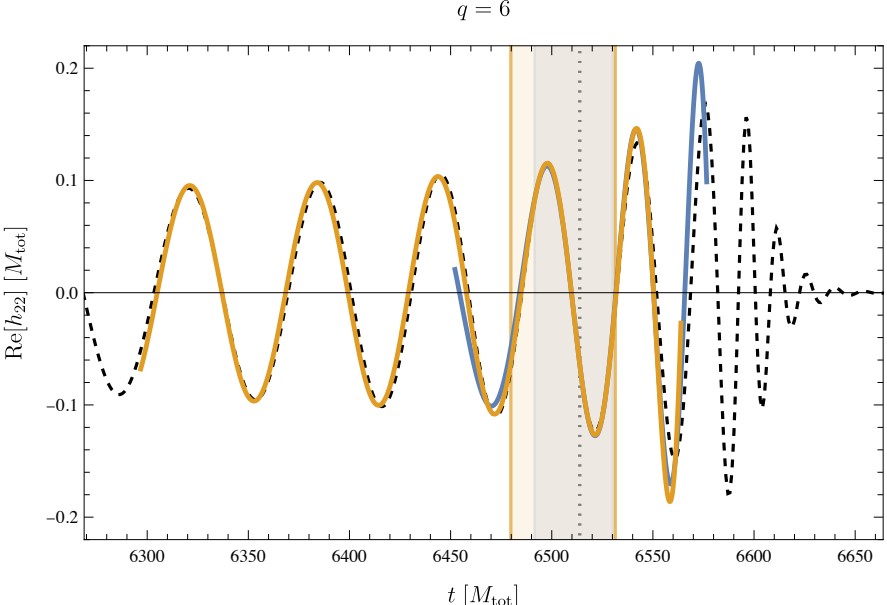

Figure 7: Same as figure 5 for a mass ratio $q = 6$. The NR simulation is SXS:BBH:0181 [58].

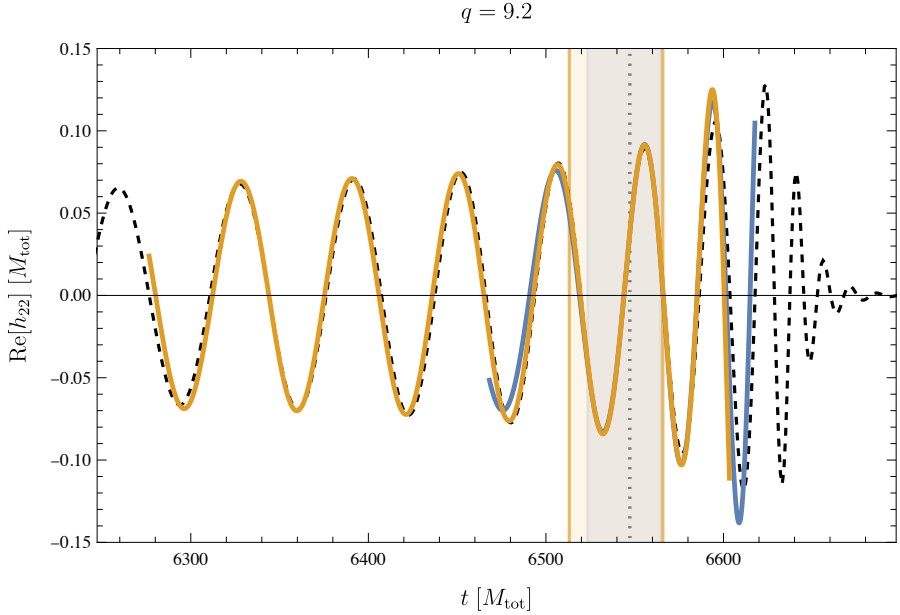

Figure 8: Same as figure 5 for a mass ratio $q = 9.2$. The NR simulation is SXS:BBH:1108 [59].

with $\nu$ as our expansion parameter, as explained in the previous section. When comparing to BHPTNRSur1dq1e4, we work in units of $M = 1$ and with $\varepsilon$ as our expansion parameter (the conventions of the uncalibrated BHPTNRSur1dq1e4).

Because our formalism is based on functions of $\Omega$ (or $\Delta\Omega$), in this section we directly compare functions of frequency rather than functions of time. This also provides a clearer view of our model's intrinsic properties rather than involving arbitrary choices of initial time and phase, as was done for the SXS waveform comparisons in the previous section. For an invariant comparison of observables, we must use the waveform frequency rather than the orbital

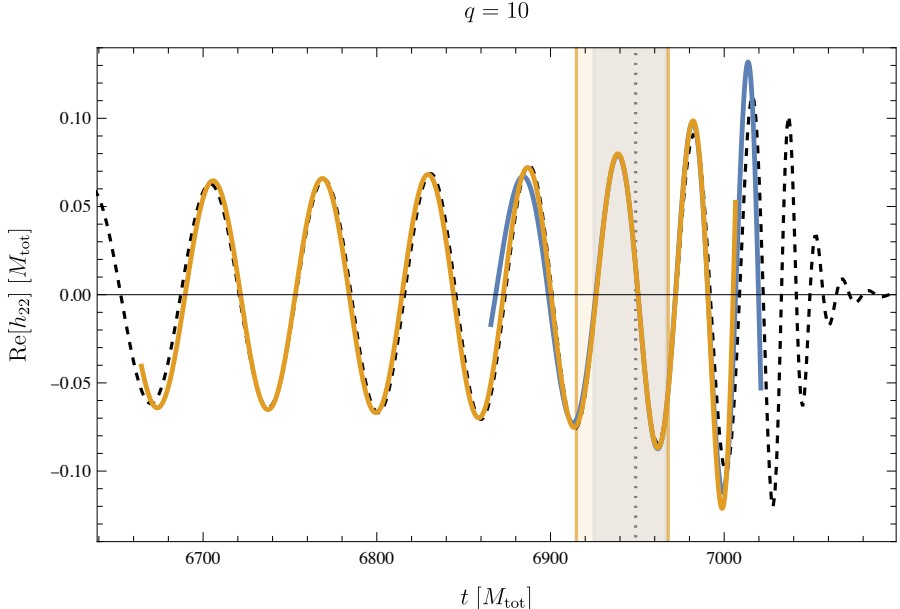

Figure 9: Same as figure 5 for a mass ratio $q = 10$. The NR simulation is SXS:BBH:1107 [60].

frequency (and indeed, this is the only frequency we can extract from the SXS and BHPTNR-Sur1dq1e4 simulations). Let us first write the $\ell m$ mode of the waveform as $h_{\ell m} = |h_{\ell m}| e^{-i\Phi_{\ell m}}$, defining the $\ell m$-mode waveform phase as $\Phi_{\ell m} := -\arg(h_{\ell m})$. Substituting our transition-to-plunge expansion of $h_{\ell m}$, we find the waveform phase is given by

$$
\begin{aligned}
\Phi_{\ell m} = m\phi_p &- \arctan\left(\frac{\mathrm{Im}\left(H_{\ell m}^{[5]}\right)}{\mathrm{Re}\left(H_{\ell m}^{[5]}\right)}\right) \\
&- \lambda^2 \frac{\mathrm{Re}\left(H_{\ell m}^{[5]}\right)\mathrm{Im}\left(H_{\ell m}^{[7]}\right) - \mathrm{Im}\left(H_{\ell m}^{[5]}\right)\mathrm{Re}\left(H_{\ell m}^{[7]}\right)}{\left|H_{\ell m}^{[5]}\right|^2} \\
&- \lambda^3 \frac{\mathrm{Re}\left(H_{\ell m}^{[5]}\right)\mathrm{Im}\left(H_{\ell m}^{[8]}\right) - \mathrm{Im}\left(H_{\ell m}^{[5]}\right)\mathrm{Re}\left(H_{\ell m}^{[8]}\right)}{\left|H_{\ell m}^{[5]}\right|^2} + O(\lambda^4),
\end{aligned}
\tag{161}
$$

where the corrections are constructed from the real and imaginary parts of

$$
H_{\ell m} = \lambda^5 H_{\ell m}^{[5]} + \lambda^7 H_{\ell m}^{[7]} + \lambda^8 H_{\ell m}^{[8]} + \dots
$$

We next define the $\ell m$-mode waveform frequency as $\omega_{\ell m} := \frac{1}{m}\frac{d\Phi_{\ell m}}{dt}$. Using eqs. (78) and (79) when taking the time derivative of $\Phi_{\ell m}$, we obtain

$$
\begin{aligned}
\omega_{\ell m} = \Omega_* &+ \lambda^2 \Delta\Omega - \lambda^3 \frac{\mathrm{Re}\left(H_{\ell m}^{[5]}\right)\mathrm{Im}\left(H_{\ell m}^{[7]A}\right) - \mathrm{Im}\left(H_{\ell m}^{[5]}\right)\mathrm{Re}\left(H_{\ell m}^{[7]A}\right)}{m\left|H_{\ell m}^{[5]}\right|^2} F_{[0]}^{\Delta\Omega} \\
&- \lambda^4 \frac{\mathrm{Re}\left(H_{\ell m}^{[5]}\right)\mathrm{Im}\left(H_{\ell m}^{[8]A}\right) - \mathrm{Im}\left(H_{\ell m}^{[5]}\right)\mathrm{Re}\left(H_{\ell m}^{[8]A}\right)}{m\left|H_{\ell m}^{[5]}\right|^2} F_{[0]}^{\Delta\Omega}\partial_{\Delta\Omega}F_{[0]}^{\Delta\Omega} + O(\lambda^5),
\end{aligned}
\tag{162}
$$

where we have used the fact that $H_{\ell m}^{[7]} = \Delta\Omega \, H_{\ell m}^{[7]A}$ and $H_{\ell m}^{[8]} = F_{[0]}^{\Delta\Omega} H_{\ell m}^{[8]A}$, which follows directly from eqs. (102) and (156). We now define $\Delta\omega_{\ell m} := (\omega_{\ell m} - \Omega_*)/\lambda^2$, and invert the equation above to obtain $\Delta\Omega$ as a function of $\Delta\omega_{\ell m}$,

$$
\begin{aligned}
\Delta\Omega = \Delta\omega_{\ell m} + \lambda \, &\frac{\mathrm{Re}\left(H_{\ell m}^{[5]}\right)\mathrm{Im}\left(H_{\ell m}^{[7]A}\right) - \mathrm{Im}\left(H_{\ell m}^{[5]}\right)\mathrm{Re}\left(H_{\ell m}^{[7]A}\right)}{m\left|H_{\ell m}^{[5]}\right|^2} F_{[0]}^{\Delta\Omega} \\
&+ \lambda^2 \Bigg[ \frac{\left(\mathrm{Re}\left(H_{\ell m}^{[5]}\right)\mathrm{Im}\left(H_{\ell m}^{[7]A}\right) - \mathrm{Im}\left(H_{\ell m}^{[5]}\right)\mathrm{Re}\left(H_{\ell m}^{[7]A}\right)\right)^2}{m^2\left|H_{\ell m}^{[5]}\right|^4} \\
&+ \frac{\mathrm{Re}\left(H_{\ell m}^{[5]}\right)\mathrm{Im}\left(H_{\ell m}^{[8]A}\right) - \mathrm{Im}\left(H_{\ell m}^{[5]}\right)\mathrm{Re}\left(H_{\ell m}^{[8]A}\right)}{m\left|H_{\ell m}^{[5]}\right|^2}\Bigg] F_{[0]}^{\Delta\Omega} \partial_{\Delta\Omega} F_{[0]}^{\Delta\Omega} + O(\lambda^3),
\end{aligned}
\tag{163}
$$

where all functions on the right-hand side are now functions of $\Delta\omega_{\ell m}$. Finally, taking the time derivative of eq. (162) and using eq. (163), we obtain the rate of change of the $\ell m$ waveform frequency as a function of $\Delta\omega_{\ell m}$,

$$
\begin{aligned}
\frac{d\omega_{\ell m}}{dt} = \lambda^3 &F_{[0]}^{\Delta\omega}(\Delta\omega_{\ell m}) + \lambda^5 F_{[2]}^{\Delta\omega}(\Delta\omega_{\ell m}) + O(\lambda^6) \\
:=\lambda^3 &F_{[0]}^{\Delta\Omega} + \lambda^5 \Bigg[ F_{[2]}^{\Delta\Omega} - \Bigg( \frac{\left(\mathrm{Re}\left(H_{\ell m}^{[5]}\right)\mathrm{Im}\left(H_{\ell m}^{[7]A}\right) - \mathrm{Im}\left(H_{\ell m}^{[5]}\right)\mathrm{Re}\left(H_{\ell m}^{[7]A}\right)\right)^2}{2m^2\left|H_{\ell m}^{[5]}\right|^4} \\
&+ \frac{\mathrm{Re}\left(H_{\ell m}^{[5]}\right)\mathrm{Im}\left(H_{\ell m}^{[8]A}\right) - \mathrm{Im}\left(H_{\ell m}^{[5]}\right)\mathrm{Re}\left(H_{\ell m}^{[8]A}\right)}{m\left|H_{\ell m}^{[5]}\right|^2} \Bigg) \left(F_{[0]}^{\Delta\Omega}\right)^2 \partial_{\Delta\Omega}^2 F_{[0]}^{\Delta\Omega} \Bigg] + O(\lambda^6).
\end{aligned}
\tag{164}
$$

We emphasise the somewhat surprising result that $H_{\ell m}^{[8]}$, which represents a 3PLT term in the waveform amplitude, enters at the same order as $F_{[2]}^{\Delta\Omega}$ in the waveform's frequency evolution. In our comparisons, we will include all the contributions to the above equations at the displayed orders *except* contributions proportional to $H_{\ell m}^{[8]}$, which would need to be extracted from second-order calculations in the inspiral. We highlight the potential importance of these omitted $H_{\ell m}^{[8]}$ terms below.

We extract $\omega_{\ell m}$ (and its time derivative) from the SXS and BHPTNRSur1dq1e4 waveforms by applying a second-order central finite difference scheme to the GW phase. Forward and backward finite difference schemes were applied at the end points of the data sets. It was found that more efficient cubic splines were inaccurate as a method of differentiating the data. In the case of the SXS simulations, a Savitzky–Golay filter was applied to the frequency and frequency evolution data to reduce high-frequency noise. This was done using Python's `scipy.signal savgol_filter` function with the options set to the following: `window=101`, `order=6`, `deriv=0`, `delta=0.01`, `mode='interp'`. The filter was applied to data starting after junk radiation and before the common horizon time in the frequency range $-0.05 \le \Delta\omega_{22} \le 0.25$, the minimum frequency range of the SXS simulations chosen to compare with, determined by the $q = 20$ simulation. In the case of the BHPTNRSur1dq1e4 datasets, small portions of the data were removed where "stitching" occurs, to avoid very high-frequency noise when

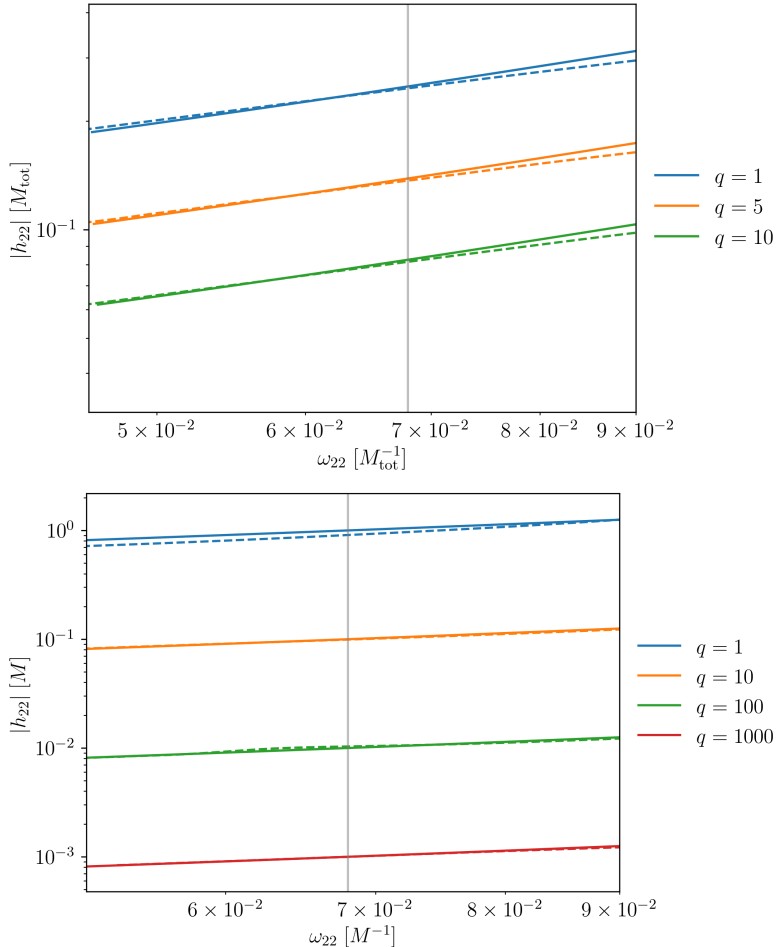

Figure 10: Comparisons of the waveform amplitude $|h_{22}|$ as a function of the waveform frequency $\omega_{22}$ for a series of mass ratios. Top: comparison between data from SXS simulations SXS:BBH:0180 ($q = 1$), SXS:BBH:0056 ($q = 5$) and SXS:BBH:1107 ($q = 10$) (dashed lines) and our 2PLT waveform model in eq. (165) with the replacement $\varepsilon \to \nu$ (solid lines). Bottom: comparison between data from the uncalibrated BHPTNRSur1dq1e4 model (dashed lines) and our 2PLT waveform model in eq. (165) (solid lines).

differentiating [32]. See more details on how the BHPTNRSur1dq1e4 model is implemented in section 7.1. Each of these details can be seen explicitly in the supplementary material.

We now turn to the comparisons. Figure 10 compares the amplitude $|h_{22}|$ as a function of the waveform frequency $\omega_{22}$ for several mass ratios. Using eqs. (159) and (163), we see the amplitude of the 2PLT model is simply given by

$$\left|h_{\ell m}^{\text{2PLT}}\right| = \varepsilon \left|H_{\ell m}^{[5]} + (\omega_{\ell m} - \Omega_*) H_{\ell m}^{[7]A}\right|. \tag{165}$$

The above equation translates to a roughly linear growth with the waveform frequency. The top panel of figure 10 compares our 2PLT model to data from SXS simulations, showing that the 2PLT amplitudes capture the behaviour of the NR amplitudes for $\omega_{22} < \Omega_*$ for all mass ratios, but with large disagreement at higher frequencies. This inaccuracy could be due to our omission of higher-order terms or due to our omission of the final plunge. We can gain some insight into the source of error by comparing to BHPTNRSur1dq1e4 in the bottom panel of figure 10, which shows good agreement between the 2PLT amplitudes and those of the

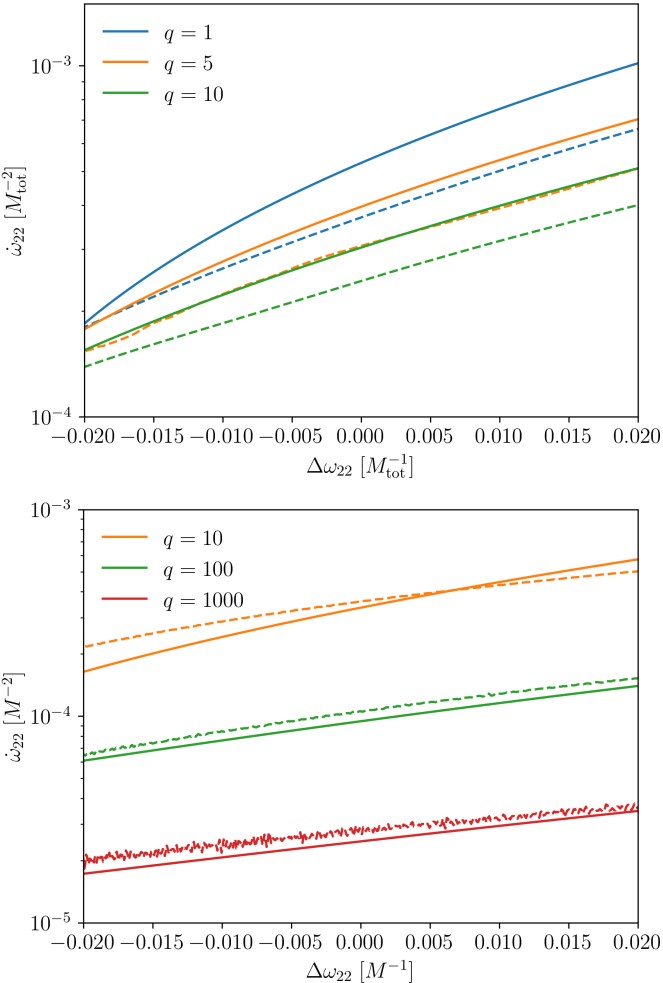

Figure 11: Comparisons of the rate of change of the waveform frequency, $\dot{\omega}_{22}$, as a function of $\Delta\omega_{22}$ for a series of mass ratios. Top: comparison between data from SXS simulations SXS:BBH:0180 ($q = 1$), SXS:BBH:0056 ($q = 5$) and SXS:BBH:1107 ($q = 10$) (dashed lines) and our 2PLT model defined in eq. (164) (solid lines), with the replacement $\lambda \rightarrow \nu^{1/5}$. Bottom: comparison between the uncalibrated BHTNR-Sur1dq1e4 model (dashed lines) and our 2PLT model (164) (solid lines).

uncalibrated BHPTNRSur1dq1e4 model across all mass ratios, despite the fact that BHPTNR-Sur1dq1e4 incorporates the final plunge. This suggests that in the frequency range shown in these figures, our omission of the plunge cannot account for our error, and the dominant source of error is likely our omission of higher-order terms in the transition-to-plunge expansion.

We next compare the underlying dynamics that drives the waveform phasing. This is entirely encoded in the evolution of the waveforms frequency, $\dot{\omega}_{\ell m} := d\omega_{\ell m}/dt$. The comparisons against SXS simulations and the BHPTNRSur1dq1e4 model are shown in the top and bottom panels of figure 11, respectively. The comparison against SXS simulations, in the top panel, shows that the accuracy of our 2PLT dynamics is significantly lower than the accuracy of our waveform amplitudes. To compare between figure 11 and figure 10, note that the frequency interval in figure 11 is significantly narrower than in figure 10: the interval $\Delta\omega_{22} \in (-0.02, 0.02)$ in figure 11 corresponds to a frequency interval ranging from $\omega_{22} \in (0.057, 0.080)$ for $q = 1$ to $\omega_{22} \in (0.061, 0.075)$ for $q = 10$. We can again question whether our error stems primarily from our omission of higher-order terms or from our omis-

sion of the plunge. However, in this case we see substantial differences even at $\Delta\omega_{22} = 0$, indicating that higher-order terms in the dynamics are important regardless of the impact of the plunge. In the bottom panel of figure 11, we also notice a significant difference between our results and those of BHPTNRSur1dq1e4. This difference does not appear to converge to zero with increasing $q$. Such lack of convergence between the two models might be due to the particular numerical details of BHPTNRSur1dq1e4 or of our extraction of $\dot\omega_{22}$. However, we will explore this in more detail in the next section, where we perform an isolated comparison with the underlying transition-to-plunge model used in BHPTNRSur1dq1e4; in that comparison, we do observe that the two models converge as expected for $q \to \infty$.

Since we are working with asymptotic approximations, the reduction in error with increasing $q$ (or, equivalently, decreasing $\varepsilon$, $\lambda$ and $\nu$) is ultimately the most crucial test of our formulation. The validity of our method rests on the assumption that for all finite $n \geq 0$, the error (i.e., the difference between an $n$PLT model and an exact solution) scales appropriately with $\lambda$. Assuming sufficiently small numerical errors and sufficiently small eccentricity in the NR simulations, our formalism implies $\left|\dot\omega_{\ell m}^{n\text{PLT}} - \dot\omega_{\ell m}^{\text{NR}}\right| = O(\lambda^{n+4})$ for all $n \geq 2$. We now test that criterion by examining the residual between NR and the 0PLT/2PLT dynamics as a function of mass ratio at a fixed value of $\Delta\omega_{22}$. From eq. (164) we first define

$$F_{0\text{PLT}}^{\omega}(\nu, \Delta\omega_{\ell m}) := \nu^{3/5} F_{[0]}^{\Delta\omega}(\Delta\omega_{\ell m}),\tag{166}$$

$$F_{2\text{PLT}}^{\omega}(\nu, \Delta\omega_{\ell m}) := \nu^{3/5} F_{[0]}^{\Delta\omega}(\Delta\omega_{\ell m}) + \nu F_{[2]}^{\Delta\omega}(\Delta\omega_{\ell m}),\tag{167}$$

noting again that we omit $H_{\ell m}^{[8]}$ terms in $F_{[2]}^{\Delta\omega}$. In principle, at fixed $\Delta\omega_{22}$ and for sufficiently small $\nu$, the residual $\left|\dot\omega_{\ell m}^{\text{NR}} - F_{0\text{PLT}}^{\omega}\right|$ should approach $\nu F_{[2]}^{\Delta\omega} = F_{2\text{PLT}}^{\omega} - F_{0\text{PLT}}^{\omega}$, and $\left|\dot\omega_{\ell m}^{\text{NR}} - F_{2\text{PLT}}^{\omega}\right|$ should scale as a 3PLT term, $O(\nu^{6/5})$. We plot these residuals in figure 12. Although the residuals decrease as we move from the 0PLT to the 2PLT model (see the orange and green dots), the residual $\left|\dot\omega_{\ell m}^{\text{NR}} - F_{0\text{PLT}}^{\omega}\right|$ does not agree well with $\nu F_{[2]}^{\Delta\omega}$. Moreover, the residual $\left|\dot\omega_{\ell m}^{\text{NR}} - F_{2\text{PLT}}^{\omega}\right|$ does not decay more rapidly than the residual $\left|\dot\omega_{\ell m}^{\text{NR}} - F_{0\text{PLT}}^{\omega}\right|$, i.e. the slope of the green points is not steeper than the slope of the orange points. However, both of these findings are expected because we neglect the contribution due to $H_{\ell m}^{[8]}$ in eq. (164). We leave the inclusion of $H_{\ell m}^{[8]}$ in our model, and assessment of its impact, to future work.

# 7 Comparison with other approaches

The preceding section compared our 2PLT model with "exact" NR simulations and with an inspiral-merger-ringdown model that (like our model) is based on black hole perturbation theory. Rather than making further numerical comparisons, we now examine how our overarching approach differs from other approaches used in inspiral-merger-ringdown models. We specifically compare to two approaches: (i) the method used by the surrogate model BHPTNRSur1dq1e4 [32, 33], including its underlying treatment of the transition to plunge [34], and (ii) the EOB framework. Both of these approaches are conceptually similar to our own but differ in important ways. In both cases, we suggest how our approach might be used to improve models that are based on these methods.

## 7.1 Comparison with the transition-to-plunge model of BHPTNRSur1dq1e4

We now compare our transition-to-plunge expansion with the approach used in the surrogate model BHPTNRSur1dq1e4 [32, 33].

The most fundamental difference between our approach and BHPTNRSur1dq1e4's is that we adopt a multiscale expansion of the Einstein equations, while BHPTNRSur1dq1e4 follows

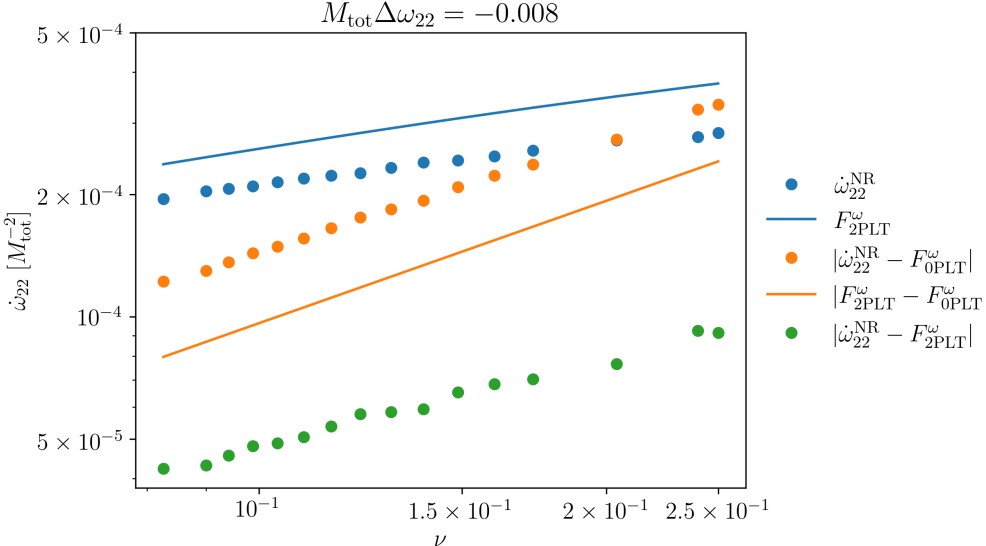

Figure 12: Waveform frequency evolution and residuals across mass ratios at a fixed value of $\Delta\omega_{22} = -0.008/M_{\text{tot}}$, well within the transition-to-plunge regime. Blue points: $\dot{\omega}_{22}$ calculated from SXS data. Blue line: $F^{\omega}_{2\text{PLT}}$, for which in this plot we have excluded the term containing the 3PLT mode amplitude $H^{[8]}_{\ell m}$ in eq. (164). Orange points: residual of the SXS data after subtracting $F^{\omega}_{0\text{PLT}}$. Orange line: the 2PLT contribution $\nu F^{\Delta\omega}_{[2]}$. Green points: the residual after subtracting $F^{\omega}_{2\text{PLT}}$ from the SXS data. Sixteen SXS simulations were used, with mass ratios varying from $q = 1$ to $q = 10$.

a type of iterative approach to solving the field equations. As stressed earlier in sections 2 and 6.1, in the multiscale approach we solve field equations on a grid of parameter values ($J^a$ for the inspiral and $\Delta J^a$ for the transition to plunge) as an offline step, and the online waveform generation is then a rapid process of evolving through the parameter space. This is the basis of the Fast EMRI Waveforms framework [7,61] and of the 0PA waveforms in [5,62], for example. It is also the only extant method of generating 1PA waveforms, and its extension to generic orbits in Kerr at 1PA (and higher) order is well understood [4]. But nearly all work on this method has been restricted to the inspiral phase, and none at all has been done for the transition to plunge. The iterative approach of BHPTNRSur1dq1e4 instead begins with leading-order, geodesic motion, solves a field equation with that motion as a source, uses the outputs (fluxes and self-forces, for example) to determine a corrected, evolving trajectory, and then solves field equations with that new trajectory. This is the basis for schemes in [63–66], which BHPTNRSur1dq1e4 builds on. An advantage of this approach is that once an inspiral-transition-plunge trajectory is known, it is straightforward to construct time-domain inspiral-merger-ringdown waveforms using a time-domain Teukolsky solver. A disadvantage is that waveform generation in this approach is slow because solving the field equations with an evolving, non-geodesic source is expensive. BHPTNRSur1dq1e4 overcomes that limitation by simulating a large bank of time-domain waveforms and then constructing a surrogate model for those waveforms; the surrogate model can then generate waveforms rapidly. Another potential disadvantage is that no framework has been developed to extend the iterative method to second order, and it is unclear whether surrogate models can be straightforwardly built for generic binary configurations.

Beyond this difference in overall strategy, our approach also differs from BHPTNRSur1dq1e4 in its construction of the transition-to-plunge trajectory, even at low or-

ders in our expansion. BHPTNRSur1dq1e4 uses the "generalized Ori-Thorne" model of Apte and Hughes [34], which corresponds to a transition-to-plunge expansion truncated at leading order in the small-mass-ratio expansion, with refinements that remove pathologies from the Ori-Thorne model. The Apte-Hughes model builds a full trajectory by switching between the adiabatic inspiral, the transition to plunge and the geodesic plunge at some times $t_i$ and $t_f$ outside and inside the ISCO, respectively. These sharp jumps from one regime to another lead to discontinuities (at $t_i$ and $t_f$) in orbital quantities such as orbital frequency, energy $E = -u_t$, and azimuthal angular momentum $L_z = u_\phi$. This is precisely what is meant by the "stitching" in the BHPTNRSur1dq1e4 model mentioned in section 6.2. These discontinuities are resolved by introducing different smoothing procedures dubbed "Model 1" and "Model 2", which we review below.

In addition to removing discontinuities, the Apte-Hughes model corrects an inconsistency in the original Ori-Thorne model [11]. The Ori-Thorne model keeps the orbital frequency fixed at its ISCO value during the full transition-to-plunge regime, trivially relating the corrections to the orbital energy and angular momentum through $\delta E = \Omega_* \delta L_z$. As noted by Kesden [49], this is in conflict with the normalization of the four-velocity; the orbital frequency and orbital radius change by a comparable amount ($\sim \lambda^2$) over the transition to plunge, making it inconsistent to freeze one while the other evolves. Even though it is not explicitly mentioned, this incorrect feature of the Ori-Thorne model is bypassed in the Apte-Hughes treatment by allowing the corrections to the orbital energy and angular momentum to evolve independently from the Ori-Thorne constraint. This makes the Apte-Hughes model consistent at 0PLT order.

Since our analysis has centred on frequency evolution throughout this paper, in our comparison we consider the Apte-Hughes orbital frequency, which we can write as[4]

$$\Omega(\lambda, \hat{\bar{\tau}}) = \frac{f(r_p(\lambda, \hat{\bar{\tau}}))L_z(\lambda, \hat{\bar{\tau}})}{E(\lambda, \hat{\bar{\tau}})r_p(\lambda, \hat{\bar{\tau}})^2} ; \tag{168}$$

cf. eq. (4.9) of [34]. Here we have introduced a slow-time variable $\hat{\bar{\tau}} := \lambda(\bar{\tau} - \bar{\tau}_*)$, where $\bar{\tau}$ is Mino time, related to coordinate time via $d\bar{\tau} = dt/(r_p^2 U)$, and where $\bar{\tau}_*$ is the Mino time at which the particle crosses the ISCO. $r_p$ has the form $r_p = 6M + \lambda^2 \delta R(\hat{\bar{\tau}})$, where $\delta R$ satisfies a Painlevé differential equation derived by Ori and Thorne [11] (and independently by Buonanno and Damour [10]). $E$ and $L_z$ have the forms $E_* + \delta E(\lambda, \hat{\bar{\tau}})$ and $L_{z*} + \delta L_z(\lambda, \hat{\bar{\tau}})$. In the original Ori-Thorne treatment, $\delta E = \lambda^4 \frac{dE}{d\bar{\tau}}\big|_* \hat{\bar{\tau}}$ and the analogue for $\delta L_z$, corresponding to the leading flux-driven changes around the ISCO. In order to remove the discontinuities at $t_i$, mentioned above, the Apte-Hughes Model 1 and Model 2 modify these transition-to-plunge expansions of $E$ and $L_z$ by adding corrections corresponding to higher-order terms in a Taylor series around the ISCO (as well as a correction to values at the ISCO, in Model 1). The coefficients in this extended Taylor series are fixed by requiring $E$ and $L_z$ to be $C^1$ at $t_i$.

Apte and Hughes' addition of these new terms in $E$ and $L_z$, and their method of fixing them, is (self-admittedly) ad hoc. However, it is conceivable that the additional terms implicitly mimic higher-order PLT terms, effectively raising the Apte-Hughes model beyond 0PLT order. To explore that possibility, we consider the Apte-Hughes frequency evolution in more detail. We restrict our analysis to their Model 2, which was indicated as the preferred choice in [34] and used in BHPTNRSur1dq1e4. We perform the comparison with our model using Mino time. The orbital frequency in the Apte-Hughes Model 2 is obtained by substituting the expansions of the orbital radius, energy and angular momentum,

$$r_p = 6M + \lambda^2 \delta R(\bar{\tau}), \tag{169}$$

---

[4]Note that here $\lambda = \varepsilon^{1/5}$, while in [34] $\lambda$ is Mino time and the mass ratio is denoted as $\eta$.

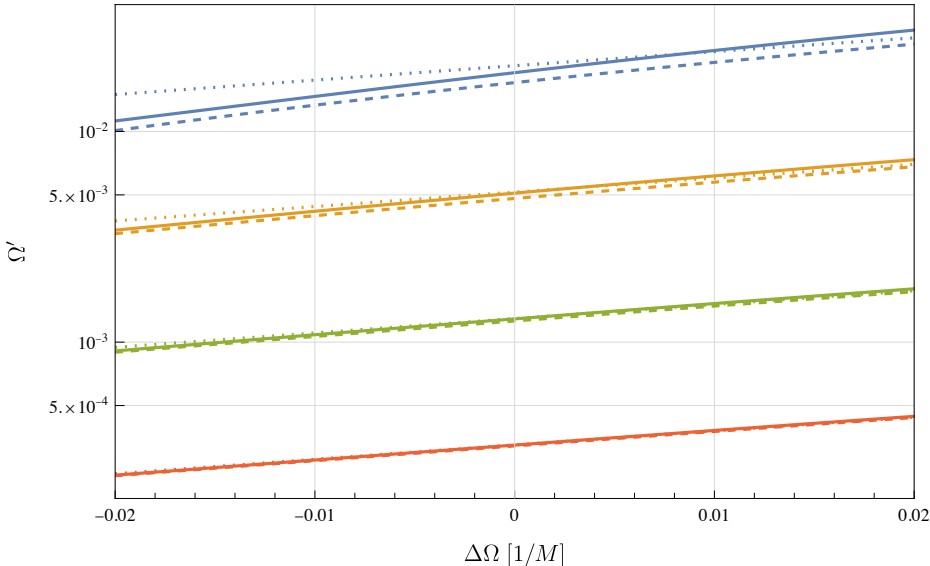

Figure 13: Rate of change of the orbital frequency with respect to Mino time, $\Omega'$, obtained from the Apte-Hughes Model 2 (solid curves) and our 0PLT (dotted curves) and 2PLT (dashed curves) approximations in eq. (172). The scale on the y-axis is logarithmic. The considered mass ratios are $\varepsilon = 10^{-1}$ (blue), $\varepsilon = 10^{-2}$ (orange), $\varepsilon = 10^{-3}$ (green), $\varepsilon = 10^{-4}$ (red). With decreasing mass ratio all curves converge to the 0PLT approximation, as expected.

$$E^{\text{M2}} = \frac{2\sqrt{2}}{3} + \lambda^4 \left.\frac{dE}{d\bar{\tau}}\right|_* \hat{\bar{\tau}} + \frac{1}{2}\left(\frac{\hat{\bar{\tau}}}{\lambda}\right)^2 \mathcal{E}_2^{\text{M2}} + \frac{1}{6}\left(\frac{\hat{\bar{\tau}}}{\lambda}\right)^3 \mathcal{E}_3^{\text{M2}}, \tag{170}$$

$$L_z^{\text{M2}} = 2\sqrt{3}M + \lambda^4 \left.\frac{dL_z}{d\bar{\tau}}\right|_* \hat{\bar{\tau}} + \frac{1}{2}\left(\frac{\hat{\bar{\tau}}}{\lambda}\right)^2 \mathcal{L}_2^{\text{M2}} + \frac{1}{6}\left(\frac{\hat{\bar{\tau}}}{\lambda}\right)^3 \mathcal{L}_3^{\text{M2}}, \tag{171}$$

into eq. (168). The values of the coefficients $\mathcal{E}_2^{\text{M2}}$, $\mathcal{E}_3^{\text{M2}}$, $\mathcal{L}_2^{\text{M2}}$ and $\mathcal{L}_3^{\text{M2}}$ were provided to us by the authors of [34] and depend on the mass ratio. It is then straightforward to compute the rate of change of the orbital frequency with respect to Mino time, $\Omega' := d\Omega/d\bar{\tau}$, which we plot as a function of $\Delta\Omega$ in figure 13 (solid curves). In order to compare with the transition-to-plunge expansion to 2PLT order, we convert eq. (78) to Mino time as

$$\frac{d\Omega}{d\bar{\tau}} = r_p^2 U \frac{d\Omega}{dt} = \lambda^3 \left(36\sqrt{2}F_{[0]}^{\Delta\Omega}\right) + \lambda^5 \left(36\sqrt{2}F_{[2]}^{\Delta\Omega} + 12\sqrt{2}r_{[0]}F_{[0]}^{\Delta\Omega} + 36U_{[0]}F_{[0]}^{\Delta\Omega}\right) + O(\lambda^6). \tag{172}$$

We display the 0PLT and 2PLT approximations in figure 13 as dotted and dashed curves, respectively. We notice that as the mass ratio decreases, both the 2PLT approximation and Model 2 asymptotically converge to the 0PLT approximation.

We now look at the transition-to-plunge expansion of the orbital frequency in more detail. From eqs. (4.5) and (4.6) of [34] we extract the mass-ratio scaling of Apte-Hughes' Model 2 parameters: $\mathcal{E}_2^{\text{M2}} \sim \mathcal{L}_2^{\text{M2}} \sim \lambda^8$ and $\mathcal{E}_3^{\text{M2}} \sim \mathcal{L}_3^{\text{M2}} \sim \lambda^{11}$. Substituting eqs. (169), (170) and (171)

into eq. (168), we then obtain

$$
\Omega = \Omega_* - \lambda^2 \frac{\delta R}{24\sqrt{6}M^2} + \lambda^4 \frac{2\sqrt{3}\left(1-54M^2\right)\frac{dE}{d\hat{\tau}}\big|_* \hat{\hat{\tau}} + 3\sqrt{6}\,\delta R^2}{2592M^3}
$$
$$
+ \lambda^6 \frac{108M^2\left(\sqrt{2}\bar{\mathcal{L}}_2^{\mathrm{M2}} - 3\sqrt{3}M\bar{\mathcal{E}}_2^{\mathrm{M2}}\right)\hat{\hat{\tau}}^2 - 3\sqrt{3}\left(1-54M^2\right)\frac{dE}{d\hat{\tau}}\big|_* \delta R\,\hat{\hat{\tau}} - 2\sqrt{6}\,\delta R^3}{15552M^4} \quad (173)
$$
$$
+ O(\lambda^8),
$$

where $\bar{\mathcal{L}}_2^{\mathrm{M2}} := \mathcal{L}_2^{\mathrm{M2}}/\lambda^8$ and $\bar{\mathcal{E}}_2^{\mathrm{M2}} := \mathcal{E}_2^{\mathrm{M2}}/\lambda^8$. We can compare this expression for $\Omega$ with an analogous one obtainable within our framework. For easier comparison with eq. (173) we compute the transition-to-plunge expansion of the orbital frequency in the slow-time formulation, that is, we expand all orbital quantities in integer powers of $\lambda$ at fixed $\hat{\hat{\tau}}$. We obtain, up to 3PLT order,

$$
\Omega = \Omega_* - \lambda^2 \frac{r_{[0]}}{24\sqrt{6}M^2} + \lambda^4 \frac{5Mr_{[0]}^2 + r_{[0]}'' - 24M^2 r_{[2]}}{576\sqrt{6}M^4} - \lambda^5 \frac{54M^2 f_{[5]}^r + r_{[3]}}{24\sqrt{6}M^2}
$$
$$
+ \lambda^6 \frac{70Mr_{[0]}^3 + 36r_{[0]}\left(r_{[0]}'' - 20M^2 r_{[2]}\right) + 9\left(192M^3 r_{[4]} + 3(r_{[0]}')^2 - 8Mr_{[2]}''\right)}{41472\sqrt{6}M^5} + O(\lambda^7).
$$
$$
(174)
$$

Here, $r_{[n]}$ are the coefficients of the expansion of the orbital radius at fixed $\hat{\hat{\tau}}$,

$$
r_p = 6M + \lambda^2 \sum_{n=0}^{\infty} \lambda^n r_{[n]}(\hat{\hat{\tau}}),
$$

and primed quantities are differentiated with respect to $\hat{\hat{\tau}}$. $f_{[5]}^r$ is given by the first-order inspiral self-force evaluated at the ISCO-crossing time (after a matching procedure analogous to the one carried out in section 5.3). Finally, we expand eq. (174) around $\hat{\hat{\tau}} = 0$,

$$
\Omega = \Omega_* + \lambda^2\left[c_{(2,3)}\hat{\hat{\tau}}^3 + O(\hat{\hat{\tau}}^8)\right] + \lambda^4\left[c_{(4,1)}\hat{\hat{\tau}} + O(\hat{\hat{\tau}}^6)\right] + \lambda^5\left[c_{(5,0)} + O(\hat{\hat{\tau}}^5)\right]
$$
$$
+ \lambda^6\left[c_{(6,4)}\hat{\hat{\tau}} + O(\hat{\hat{\tau}}^9)\right] + O(\lambda^7). \quad (175)
$$

The coefficients $c_{(m,n)}$, $m, n \in \mathbb{N}$, labelled with the powers of $\lambda$ and $\hat{\hat{\tau}}$ with which they appear in the expansion, are constructed from ISCO quantities only. They can be easily obtained by solving the equations of motion for the radial corrections $r_{[n]}$, $n \geq 0$, in the limit $\hat{\hat{\tau}} \to 0$ (explicitly, we find $r_{[0]} \sim \hat{\hat{\tau}}^3 + O(\hat{\hat{\tau}}^8)$, $r_{[2]} \sim \hat{\hat{\tau}}^6 + O(\hat{\hat{\tau}}^{11})$, $r_{[3]} \sim \hat{\hat{\tau}}^5 + O(\hat{\hat{\tau}}^{10})$ and $r_{[4]} \sim \hat{\hat{\tau}}^4 + O(\hat{\hat{\tau}}^9)$). By comparing eqs. (174) and (175) with eq. (173), we again confirm that the transition-to-plunge description of Apte and Hughes is equivalent to our 0PLT approximation since $r_{[0]}$ and $\delta R$ solve the same Painlevé transcendental equation. Although Model 2 does not include $r_{[2]}$, it correctly captures the general behaviour (linear in $\hat{\hat{\tau}}$ at leading order) of the 2PLT order, though the explicit time dependence is distinct. The two models begin to significantly differ at 3PLT order (and in general all odd PLT orders, which are absent in Apte-Hughes' Model 2), where our expansion starts to include conservative effects.

Our analysis in this section shows that the ability of the correction terms in Apte and Hughes' model to consistently mimic higher PLT orders is limited. This leads us to conclude that the inclusion of higher-order PLT terms should improve the performance of BHPTNR-Sur1dq1e4 and similar models. We note, in particular, that 2PLT terms are easily calculated from 0PA fluxes and therefore can be easily incorporated.

The Apte-Hughes model describes the transition to plunge for all black hole spins, while the results in this paper are limited to a Schwarzschild primary. It is important to mention that the multiscale, phase-space framework presented here remains unchanged and naturally carries over to quasi-circular and equatorial orbits in Kerr spacetime (results limited to the orbital motion in the slow-time formulation were presented in [12–14]). Given the availability of adiabatic inspirals in Kerr spacetime, 2PLT waveforms can be computed within a framework equivalent to the one we have presented in this paper (see [67]). Moving to higher PLT orders presents the same challenges that are being tackled in constructing 1PA inspiral waveforms in Kerr spacetime (e.g., the construction of the second-order source) [68].

## 7.2 Comparison with the effective one-body framework

The EOB waveform-generation framework is conceptually very similar to our own. The key inputs (a Hamiltonian, waveform amplitudes, and radiation-reaction forces) are calculated as functions on phase space, and the waveform is then computed by solving ordinary differential equations to find phase-space trajectories. EOB also provides an important point of comparison for several reasons:

1. The EOB dynamics is known to exactly recover 0PLT dynamics in the small-mass-ratio limit [10]. It is important to assess whether EOB also captures subleading PLT dynamics.

2. Quasicircular EOB waveforms are based on a quasicircular approximation in which $d\Omega/dt$ (or $dr_p/dt$) is explicitly set to zero in the calculation of waveform amplitudes and fluxes, and in the late inspiral and plunge this treatment of the amplitudes is corrected through non-quasicircular (NQC) factors that are calibrated to numerical waveforms [24, 69, 70]. It is conceivable that these NQC corrections are related to the subleading terms involving $d\Omega/dt$ and $d\Delta\Omega/dt$ that appear throughout our multiscale expansion.

3. The inclusion of NQC corrections has been crucial for achieving high accuracy with EOB waveforms [71, 72], but their importance is significantly reduced by using self-force data to improve EOB's treatment of energy fluxes [73]. It is possible that EOB can be further improved, and its reliance on NQC corrections can be further reduced, by using information from the self-forced transition to plunge.

Motivated by these considerations, in this section we show that EOB can, in principle, reproduce our results for the transition-to-plunge dynamics at least to 3PLT order in the small-mass-ratio limit. However, we also pinpoint how EOB implementations can miss certain effects related to evolution of $\Omega$ (or $r_p$), and we highlight possible ways our results could be used to inform EOB models.

The EOB formalism begins by mapping the two-body dynamics onto the effective dynamics of a particle in a deformed Schwarzschild background, with the deformation parametrized by the symmetric mass ratio $\nu$. We consider the EOB Hamiltonian [10, 22, 74]

$$H_{\text{EOB}} := \frac{M_{\text{tot}}}{m_p}\sqrt{1 + 2\nu\left[\sqrt{A(r_p)\left(1 + \frac{p_r^2 m_p^2}{\mu^2 B(r_p)} + \frac{p_\phi^2 m_p^2}{\mu^2 r_p^2} + Q(r_p, p_r)\right)} - 1\right]}, \qquad (176)$$

where the total mass $M_{\text{tot}}$, the reduced mass $\mu$ and the symmetric mass ratio $\nu$ are defined as

$$M_{\text{tot}} := M + m_p = M(1 + \varepsilon), \quad \mu := \frac{M m_p}{M_{\text{tot}}} = \frac{M\varepsilon}{1 + \varepsilon}, \quad \nu := \frac{\mu}{M_{\text{tot}}} = \frac{\varepsilon}{(1 + \varepsilon)^2}. \qquad (177)$$

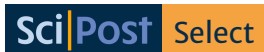 Select

To facilitate the comparison with the self-force dynamics, we have rescaled the EOB Hamiltonian and the momenta $p_\mu$ with the mass of the secondary $m_p = M\varepsilon$ (though we recognize this is unnatural in the EOB framework, where one would normally use $\mu$ for such rescaling). The EOB potentials expanded in powers of $\nu$ are given by [75, 76]

$$A(r_p) = 1 - \frac{2M_{\text{tot}}}{r_p} + \nu\, a(M_{\text{tot}}/r_p) + \nu^2 a_2(M_{\text{tot}}/r_p) + O(\nu^3), \tag{178}$$

$$B(r_p) = \frac{1}{A(r_p)}\left[1 + \nu\, d(M_{\text{tot}}/r_p) + \nu^2 d_2(M_{\text{tot}}/r_p) + O(\nu^3)\right], \tag{179}$$

$$Q(r_p, p_r) = \nu\, q(M_{\text{tot}}/r_p) p_r^4 + O(\nu^2, p_r^6). \tag{180}$$

The potentials $B$ and $Q$ only affect eccentric orbits and do not enter the comparison we consider here until high order (e.g., we have found the potential $d$ affects the transition-to-plunge dynamics at 5PLT order). At the orders we consider explicitly here, we will only encounter the potential $\nu a(M_{\text{tot}}/r_p)$, and we can replace $\nu$ with $\varepsilon$ in that term.

The system's Hamilton equations read [10]

$$\frac{dr_p}{dt} = \frac{\partial H_{\text{EOB}}}{\partial p_r}, \tag{181a}$$

$$\frac{d\phi_p}{dt} := \Omega = \frac{\partial H_{\text{EOB}}}{\partial p_\phi}, \tag{181b}$$

$$\frac{dp_r}{dt} = -\frac{\partial H_{\text{EOB}}}{\partial r_p} + F_r, \tag{181c}$$

$$\frac{dp_\phi}{dt} = F_\phi. \tag{181d}$$

The radiation-reaction forces appearing in the Hamilton equations are given by [23, 73]

$$F_r = -\frac{\mathcal{F}^{\text{EOB}}}{\Omega}\frac{p_r}{p_\phi}, \qquad F_\phi = -\frac{\mathcal{F}^{\text{EOB}}}{\Omega}, \tag{182}$$

where $\mathcal{F}^{\text{EOB}}$ is the energy flux per unit secondary mass [73]

$$\mathcal{F}^{\text{EOB}} = \frac{1}{8\pi m_p}\sum_{\ell,m}(m\Omega)^2 |H_{\ell m}|^2. \tag{183}$$

Before we examine the transition to plunge, it will be instructive to consider the simpler case of the inspiral, comparing the EOB dynamics with the self-force inspiral expansion of section 3. To make this comparison, we perturbatively expand the EOB equations in integer powers of $\varepsilon$ at fixed mechanical parameters $J^a$. We refer to reference [77] for a more detailed treatment of this expansion of the EOB dynamics. In practice, we substitute $r_p$ and $dJ^a/dt$ using, respectively, eqs. (33) and (35), and additionally expand

$$p_r(\varepsilon, J^a) = \varepsilon\left[p_r^{(0)}(\Omega) + \varepsilon\, p_r^{(1)}(J^a) + O(\varepsilon^2)\right], \tag{184}$$

$$p_\phi(\varepsilon, J^a) = p_\phi^{(0)}(\Omega) + \varepsilon\, p_\phi^{(1)}(J^a) + O(\varepsilon^2), \tag{185}$$

$$\mathcal{F}^{\text{EOB}}(\varepsilon, J^a) = \varepsilon\, \mathcal{F}^{\text{EOB}}_{(1)}(\Omega) + \varepsilon^2 \mathcal{F}^{\text{EOB}}_{(2)}(J^a) + O(\varepsilon^3). \tag{186}$$

The radiation-reaction forces (182) then admit the following expansions:

$$F_r(\varepsilon, J^a) = \varepsilon^2 F_r^{(2)}(J^a) + O(\varepsilon^3), \tag{187a}$$

$$F_\phi(\varepsilon, J^a) = \varepsilon\, F_\phi^{(1)}(\Omega) + \varepsilon^2 F_\phi^{(2)}(J^a) + O(\varepsilon^3)\,. \tag{187b}$$

At adiabatic order, the expanded Hamilton equations (181) give

$$r_{(0)} = \frac{M}{(M\Omega)^{2/3}}\,, \tag{188}$$

$$F_{(0)}^\Omega = -\frac{3\Omega^{4/3}}{M^{2/3} U_{(0)} D} F_\phi^{(1)}\,. \tag{189}$$

The radiation-reaction force $F_\phi$ defined from eq. (181d) is related to the azimuthal self-force by $F_\phi = \frac{f_\phi}{U}$, which at adiabatic order reduces to $F_\phi^{(1)} = \frac{f_\phi^{(1)}}{U_{(0)}} = \frac{f(r_{(0)})}{\Omega U_{(0)}} f_{(1)}^t$, where the last equality derives from the orthogonality condition $u_\mu f^\mu = 0$. Substituting this into eq. (189) allows us to correctly reproduce the 0PA result in eq. (37). The 1PA correction to the orbital radius reads

$$r_{(1)} = \frac{M}{6(M\Omega)^{2/3}} \left[ 6 - 4U_{(0)} + 8(M\Omega)^{2/3} U_{(0)} - a'\left((M\Omega)^{2/3}\right) \right]\,. \tag{190}$$

This is a gauge-dependent quantity. While the EOB framework we discuss here is set up in the Schwarzschild gauge, our formalism uses the Schwarzschild gauge only at the background level, without specifying a gauge for the metric perturbations appearing in the self-force (61). Equating eqs. (190) and the 1PA result (39a) corresponds to adopting the EOB gauge for the self-force dynamics. The 1PA rate of change of the orbital frequency then reproduces the self-force result in eq. (39b), after substituting the radiation-reaction force as $F_\phi^{(2)} = \frac{1}{U_{(0)}} f_\phi^{(2)} - \frac{U_{(1)}}{U_{(0)}^2} f_\phi^{(1)}$ and using the orthogonality condition to write $f_\phi^{(2)}$ in terms of the $t$ and $r$ components of the first- and second-order self-force,

$$f_\phi^{(2)} = \frac{1}{\Omega} \left[ f(r_{(0)}) f_{(2)}^t + f'(r_{(0)}) r_{(1)} f_{(1)}^t - \frac{\partial_\Omega r_{(0)} F_{(0)}^\Omega}{f(r_{(0)})} f_{(1)}^r \right]\,.$$

This analysis shows that the EOB dynamics can reproduce the 1PA self-force dynamics when expanded in the same small-mass-ratio, multiscale form. Referring back to our enumerated items at the beginning of the section, we can now judge that the EOB dynamics *without* NQCs already correctly encodes the effects of nonzero $d\Omega/dt$ during the inspiral. These terms arise from substituting expansions in powers of $\varepsilon$ at fixed $\Omega$ into the left-hand side of Hamilton's equations (and applying the chain rule). In fact, the multiscale expansion in the small-mass-ratio limit is roughly equivalent to the post-adiabatic expansion used in EOB [78]. Like our inspiral expansion, the post-adiabatic EOB expansion breaks down at an ISCO due to the behaviour of $d\Omega/dt$ in this approximation; such breakdown can be seen by eye from the plots in reference [78].

However, there are several caveats to this conclusion:

1. Exactly reproducing the 1PA self-force dynamics requires fixing the EOB potential $a$ in eq. (178) to agree with its value in self-force theory. This has been done in a gauge-invariant way in references [75, 79, 80], for example.

2. We have equated the EOB radiation-reaction force $F_\phi$ to the local self-force $f_\phi/U$. Since $F_\phi$ in EOB is obtained from energy fluxes via eq. (182), we have implicitly assumed $f_\phi$ is linked to fluxes through an energy balance law of the form (182). This relationship is not known to be true at second order in self-force theory.

3. Even assuming energy balance, the EOB dynamics will only reproduce the 1PA dynamics if the fluxes $\mathcal{F}_{(1)}^{\text{EOB}}$ and $\mathcal{F}_{(2)}^{\text{EOB}}$ are made to agree with their values from self-force theory.

Since the EOB fluxes are calculated from waveform amplitudes via eq. (183), reproducing the self-force fluxes requires reproducing the amplitudes $H_{\ell m}^{(1)}$ and $H_{\ell m}^{(2)}$. This calibration has been done in reference [73], for example. Importantly, the amplitudes $H_{\ell m}^{(2)}$ contain contributions directly proportional to $\dot{\Omega}$ (through $\dot{\Omega}$ terms that appear as sources in the second-order Einstein equation). These contributions must be consistently accounted for in the EOB inspiral amplitudes. They also appear to be distinct from NQC corrections: the terms that appear in $H_{\ell m}^{(2)}$ are linear in $\dot{\Omega}$, while NQC corrections to the amplitudes appear as a multiplicative factor that depends only on even powers of $p_r$ ($\sim \dot{\Omega}$) [73]. This hints that, at least in some variants of EOB, NQC corrections might serve to mask inaccuracies in the inspiral dynamics rather than serving to correct fundamental inadequacies of a quasicircular approximation.

4. Equation (183) for the energy flux is itself not exact. First, it must include fluxes into the primary black hole (or into both black holes for comparable-mass binaries). Second, if we write the waveform modes in terms of real amplitudes, as $h_{\ell m} = |H_{\ell m}(J^a)|e^{-i\Phi_{\ell m}}$, then the exact flux to future null infinity (per unit secondary mass) is

$$\mathcal{F} = \frac{1}{8\pi m_p} \sum_{\ell m} |\dot{h}_{\ell m}|^2 = \frac{1}{8\pi m_p} \sum_{\ell m} \left[ (m\omega_{\ell m})^2 |H_{\ell m}|^2 + \left( \frac{dJ^a}{dt} \partial_{J^a} |H_{\ell m}| \right)^2 \right], \qquad (191)$$

where $\omega_{\ell m} := \frac{1}{m}\dot{\Phi}_{\ell m}$. This is the physical energy flux, proportional to minus the integral over the sphere at infinity of the square of the time derivative of the shear. Note the numerically dominant piece of the second term comes from $\dot{\Omega}\partial_{\Omega}|H_{\ell m}|$.

Equation (191) differs from the flux formula (183) in two ways. First, it involves factors of the waveform mode frequency $\omega_{\ell m}$, which differs from $\Omega$ by small but non-negligible $O(\varepsilon)$ corrections proportional to $\dot{\Omega}$, in analogy with eq. (162); see Appendix B of reference [18]. Second, eq. (191) depends explicitly on $(\dot{\Omega})^2$. However, the terms proportional to $(\dot{\Omega})^2$ are suppressed by $O(\varepsilon^2)$ because $\dot{\Omega} = O(\varepsilon)$, meaning this second correction is not relevant to the 1PA dynamics.

We now turn to the transition-to-plunge expansion of the EOB dynamics, keeping in mind our findings from the inspiral. To compare with the results we have obtained within the self-force formalism in section 4, we consider the near-ISCO scaling of the orbital frequency (75) and expand the orbital radius and the rates of change of $\Delta J^a$ according to eqs. (76), (78) and (79). The momenta $p_r$ and $p_\phi$ and the EOB energy flux are also expanded as

$$p_r(\lambda, \Delta J^a) = \lambda^3 \sum_{n=0}^{2} \lambda^n p_r^{[n]}(\Delta\Omega) + \lambda^3 \sum_{n=3}^{\infty} \lambda^n p_r^{[n]}(\Delta J^a), \qquad (192)$$

$$p_\phi(\lambda, \Delta J^a) = \sum_{n=0}^{4} \lambda^n p_\phi^{[n]}(\Delta\Omega) + \sum_{n=5}^{\infty} \lambda^n p_\phi^{[n]}(\Delta J^a), \qquad (193)$$

$$\mathcal{F}^{\text{EOB}}(\lambda, \Delta J^a) = \lambda^5 \mathcal{F}_{[5]}^{\text{EOB}} + \lambda^7 \mathcal{F}_{[7]}^{\text{EOB}}(\Delta\Omega) + \sum_{n=8}^{\infty} \lambda^n \mathcal{F}_{[n]}^{\text{EOB}}(\Delta J^a). \qquad (194)$$

Substituting these expansions into eq. (182), we find that the radiation-reaction forces admit the following expansions:

$$F_r(\lambda, \Delta J^a) = \lambda^8 F_r^{[8]}(\Delta J^a) + O(\lambda^9), \qquad (195a)$$

$$F_\phi(\lambda, \Delta J^a) = \lambda^5 F_\phi^{[5]} + \lambda^7 F_\phi^{[7]}(\Delta\Omega) + \lambda^8 F_\phi^{[8]}(\Delta J^a) + O(\lambda^9). \qquad (195b)$$

We can relate the radiation-reaction forces (195) to the $t$ and $r$ components of the transition-to-plunge self-force (80) by again using $F_\phi = \frac{f_\phi}{U}$ and the orthogonality condition $u^\mu f_\mu = 0$:

$$F_\phi^{[5]} = 4\sqrt{3}M f_{[5]}^t, \tag{196a}$$

$$F_\phi^{[7]} = 4M\left(\sqrt{3}f_{[7]}^t - 30\sqrt{2}M\Delta\Omega f_{[5]}^t\right), \tag{196b}$$

$$F_\phi^{[8]} = 4\sqrt{3}M\left(f_{[8]}^t + 54\sqrt{6}M^2 F_{[0]}^{\Delta\Omega}f_{[5]}^r\right). \tag{196c}$$

We obtain the equations of motion for the forcing terms $F_{[n]}^{\Delta\Omega}$ from the expanded Hamilton equations (181). After using the relations (196), we have checked up to 3PLT order that $F_{[0]}^{\Delta\Omega}$ satisfies eq. (83), while the subleading terms $F_{[2]}^{\Delta\Omega}$ and $F_{[3]}^{\Delta\Omega}$ obey eq. (86) with sources (C.3) and (C.4), respectively.

From this analysis, we conclude that the EOB formalism correctly captures the dynamics of the transition to plunge at least to 3PLT order in the small-mass-ratio limit. However, all of the caveats we listed for the inspiral are even more pronounced for the transition to plunge. In particular, precise balance laws between the local force and asymptotic flux have not been derived for the transition-to-plunge expansion, and corrections to the flux formula (183) could be important. The exact flux formula (191) in the transition-to-plunge regime becomes

$$\mathcal{F} = \frac{1}{8\pi m_p}\sum_{\ell m}\left[(m\omega_{\ell m})^2|H_{\ell m}|^2 + \left(\frac{d\Delta J^a}{dt}\partial_{\Delta J^a}|H_{\ell m}|\right)^2\right]. \tag{197}$$

As shown in eq. (161), the waveform frequency $\omega_{\ell m}$ differs from the orbital frequency $\Omega$ by an amount of order $\varepsilon^{3/5}$ (3PLT), which could be substantially more significant than the analogous difference in the inspiral. Moreover, the scalings $\frac{d\Delta\Omega}{dt} \sim \varepsilon^{1/5}$ and $\partial_{\Delta J^a}|H_{\ell m}| \sim \varepsilon^{2/5}$ suggest that the second term in eq. (197) is only suppressed by a factor $\varepsilon^{6/5}$ relative to the first term, again making it more significant than the analogous correction in the inspiral.

These considerations suggest that EOB inspiral-merger-ringdown waveforms might be further improved using self-force waveform amplitudes (or fluxes) during the transition to plunge. However, this might require carrying self-force calculations to 3PLT order and higher. Conversely, the fact that EOB dynamics encodes the transition-to-plunge dynamics (given correct radiation-reaction forces) suggests that our transition-to-plunge results might be resummable into a simpler form.

## 8 Conclusion

We have established from first principles a framework based on multiscale expansions that enables fast waveform generation during the transition-to-plunge stage of asymmetric compact binary coalescences. Our framework builds on the waveform-generation formalism presented in [4, 8, 9] for the inspiral, combined with the scheme of matched asymptotic expansions between the inspiral and the transition to plunge developed in [12–14, 48]. Our framework is complete for non-spinning, quasi-circular and equatorial binaries, and in particular takes into account the dynamical change of the background mass and spin.

We have considered the coupled problem of the orbital motion and the field equations in the inspiral and transition-to-plunge regimes separately, and have shown that the solutions to the two problems can be asymptotically matched in a buffer region exterior to the ISCO where the two regimes overlap. Using the GSF data that is currently available, we have built the simplest transition-to-plunge waveforms using a 0PLT and a 2PLT model, which takes into

account the transition-to-plunge dynamics to leading and first non-vanishing subleading order in the multiscale expansion, respectively. We have left the hybridization of our transition-to-plunge waveforms with the inspiral and the plunge at, respectively, early and late times to future work. We have performed extensive comparisons between our GSF models and NR simulations, the BHPTNRSur1dq1e4 surrogate model and the EOB formalism. We found that our models lead to a promising route for capturing both the orbital dynamics and the waveforms of asymmetric binaries; see, e.g., figures 5 and 12. These numerical results validate our GSF model and encourage the numerical implementation of higher-order PLT models. They also motivate the analytic development and numerical implementation of more complete models that include the primary spin and arbitrary orbital dynamics including eccentricity and spin precession. This is left for future work.

Finally, we conclude by estimating how the model should improve as higher PLT orders are included. For the inspiral, the GSF community has focused specifically on achieving 1PA accuracy because the orbital phase (and hence the GW phase) has an expansion of the form [3]

$$\phi_p = \frac{1}{\varepsilon}\left[\phi_p^{(0)}(\Omega) + \varepsilon\phi_p^{(1)}(\Omega, \delta M^{\pm}) + O(\varepsilon^2)\right]. \tag{198}$$

The validity of this expansion can be seen straightforwardly from the structure of the equations that determine $\phi_p$ during the inspiral, reproduced here for convenience:

$$\frac{d\phi_p}{dt} = \Omega, \tag{199}$$

$$\frac{d\Omega}{dt} = \varepsilon\left[F_{(0)}^{\Omega}(\Omega) + \varepsilon F_{(1)}^{\Omega}(\Omega, \delta M^{\pm}) + O(\varepsilon^2)\right]. \tag{200}$$

Equation (198) shows that, on any fixed frequency interval (e.g., the LISA frequency band), the phase error of a 1PA model scales linearly with $\varepsilon$ in the $\varepsilon \to 0$ limit. A 1PA model can therefore guarantee sufficiently small phase errors for sufficiently small $\varepsilon$. During the transition to plunge, the phase instead admits an expansion of the form

$$\phi_p = \frac{1}{\varepsilon^{1/5}}\left[\phi_p^{[0]}(\Delta\Omega) + \varepsilon^{2/5}\phi_p^{[2]}(\Delta\Omega) + O(\varepsilon^{3/5})\right], \tag{201}$$

as can be seen from the structure of the transition-to-plunge equations

$$\frac{d\phi_p}{dt} = \Omega_* + \varepsilon^{2/5}\Delta\Omega, \tag{202}$$

$$\frac{d\Delta\Omega}{dt} = \varepsilon^{1/5}\left[F_{[0]}^{\Delta\Omega}(\Delta\Omega) + \varepsilon^{2/5}F_{[2]}^{\Delta\Omega}(\Delta\Omega) + O(\varepsilon^{3/5})\right]. \tag{203}$$

This suggests that phase errors vanish in the $\varepsilon \to 0$ limit *even for a 0PLT model*. For a 7PLT model (being the highest PLT order considered in this paper) the phase error scales as $\varepsilon^{7/5}$, formally even smaller than the phase error accumulated over a 1PA inspiral. Such scaling estimates should be used with caution for the transition to plunge because the low fractional powers of $\varepsilon$ mean that subsequent terms can easily compete with each other. However, when combined with our promising results at 0PLT and 2PLT order, these estimates suggest that a higher-order PLT model should be highly accurate for mass ratios in the range $\lesssim 1/10$, which is of interest for ground-based detectors.

## Acknowledgments

We gratefully thank Maarten van de Meent, Niels Warburton and Barry Wardell for insightful discussions during the course of this project. We also thank Barry Wardell for providing densely

sampled 0PA numerical flux data. We are particularly grateful to Scott Hughes for providing comparison 0PLT data that led us to discover an error in our numerical implementation. This work makes use of the Black Hole Perturbation Toolkit.

**Funding information**   G.C. and L.K. acknowledge the support of the PRODEX Experiment Arrangement 4000129178 between ESA and ULB. A.P. and L.K. acknowledge the support of a UKRI Frontier Research Grant (as selected by the ERC) under the Horizon Europe Guarantee scheme [grant number EP/Y008251/1]. A.P. additionally acknowledges the support of a Royal Society University Research Fellowship. G.C. is Senior Research Associate of the F.R.S.-FNRS and acknowledges support from the FNRS research credit J.0036.20F.

# A   Material for section 2

## A.1   Barack-Lousto-Sago basis of tensor spherical harmonics

The Barack-Lousto-Sago harmonics [41, 81, 82] in Schwarzschild coordinates $(t, r, \theta, \phi)$ are given by

$$
Y^{1\ell m}_{\mu\nu} = \frac{Y^{\ell m}}{\sqrt{2}}\begin{pmatrix} 1 & 0 & 0 & 0 \\ 0 & f^{-2} & 0 & 0 \\ 0 & 0 & 0 & 0 \\ 0 & 0 & 0 & 0 \end{pmatrix}, \quad Y^{2\ell m}_{\mu\nu} = \frac{Y^{\ell m}}{\sqrt{2}f}\begin{pmatrix} 0 & 1 & 0 & 0 \\ 1 & 0 & 0 & 0 \\ 0 & 0 & 0 & 0 \\ 0 & 0 & 0 & 0 \end{pmatrix},
$$

$$
Y^{3\ell m}_{\mu\nu} = \frac{Y^{\ell m}}{\sqrt{2}}\begin{pmatrix} f & 0 & 0 & 0 \\ 0 & -f^{-1} & 0 & 0 \\ 0 & 0 & 0 & 0 \\ 0 & 0 & 0 & 0 \end{pmatrix}, \quad Y^{4\ell m}_{\mu\nu} = \frac{r\,Y^{\ell m}}{\sqrt{2}\gamma_1}\begin{pmatrix} 0 & 0 & \partial_\theta & \partial_\phi \\ 0 & 0 & 0 & 0 \\ \partial_\theta & 0 & 0 & 0 \\ \partial_\phi & 0 & 0 & 0 \end{pmatrix},
$$

$$
Y^{5\ell m}_{\mu\nu} = \frac{r\,Y^{\ell m}}{\sqrt{2}\gamma_1 f}\begin{pmatrix} 0 & 0 & 0 & 0 \\ 0 & 0 & \partial_\theta & \partial_\phi \\ 0 & \partial_\theta & 0 & 0 \\ 0 & \partial_\phi & 0 & 0 \end{pmatrix}, \quad Y^{6\ell m}_{\mu\nu} = \frac{r^2 Y^{\ell m}}{\sqrt{2}}\begin{pmatrix} 0 & 0 & 0 & 0 \\ 0 & 0 & 0 & 0 \\ 0 & 0 & 1 & 0 \\ 0 & 0 & 0 & s^2 \end{pmatrix}, \tag{A.1}
$$

$$
Y^{7\ell m}_{\mu\nu} = \frac{r^2 Y^{\ell m}}{\sqrt{2}\gamma_2}\begin{pmatrix} 0 & 0 & 0 & 0 \\ 0 & 0 & 0 & 0 \\ 0 & 0 & D_2 & D_1 \\ 0 & 0 & D_1 & -s^2 D_2 \end{pmatrix}, \quad Y^{8\ell m}_{\mu\nu} = \frac{r\,Y^{\ell m}}{\sqrt{2}\gamma_1}\begin{pmatrix} 0 & 0 & s^{-1}\partial_\phi & -s\,\partial_\theta \\ 0 & 0 & 0 & 0 \\ s^{-1}\partial_\phi & 0 & 0 & 0 \\ -s\,\partial_\theta & 0 & 0 & 0 \end{pmatrix},
$$

$$
Y^{9\ell m}_{\mu\nu} = \frac{r\,Y^{\ell m}}{\sqrt{2}\gamma_1 f}\begin{pmatrix} 0 & 0 & 0 & 0 \\ 0 & 0 & s^{-1}\partial_\phi & -s\,\partial_\theta \\ 0 & s^{-1}\partial_\phi & 0 & 0 \\ 0 & -s\,\partial_\theta & 0 & 0 \end{pmatrix}, \quad Y^{10\ell m}_{\mu\nu} = \frac{r^2 Y^{\ell m}}{\sqrt{2}\gamma_2}\begin{pmatrix} 0 & 0 & 0 & 0 \\ 0 & 0 & 0 & 0 \\ 0 & 0 & s^{-1}D_1 & -s\,D_2 \\ 0 & 0 & -s\,D_2 & -s\,D_1 \end{pmatrix},
$$

where $s := \sin\theta$, $f := 1 - 2M/r$, $D_1 := 2(\partial_\theta - \cot\theta)\partial_\phi$, $D_2 := \partial_\theta^2 - \cot\theta\,\partial_\theta - s^{-2}\partial_\phi^2$ and $Y^{\ell m} = Y^{\ell m}(\theta, \phi)$ are the standard scalar spherical harmonics. The coefficients $\gamma_1$ and $\gamma_2$ are defined as

$$
\gamma_1 := \ell(\ell+1), \qquad \gamma_2 := (\ell-1)\ell(\ell+1)(\ell+2). \tag{A.2}
$$

Here we use the definition of $Y^{3\ell m}_{\mu\nu}$ from [41] rather than [81]; the two differ by a factor of $f$. The harmonics $Y^{i\ell m}_{\mu\nu}$ are orthogonal with respect to a certain inner product, which satisfies

$$
\oint dS\,\eta^{\alpha\mu}\eta^{\beta\nu}Y^{i\ell m}_{\mu\nu}Y^{*j\ell'm'}_{\alpha\beta} = \kappa_i\delta_{ij}\delta_{\ell\ell'}\delta_{mm'}, \tag{A.3}
$$

with $dS = \sin\theta\, d\theta\, d\phi$ the surface element on the unit sphere,

$$\eta^{\mu\nu} := \mathrm{diag}\left(1, f^2, r^{-2}, r^{-2}\sin^{-2}\theta\right), \tag{A.4}$$

and

$$\kappa_i := \begin{cases} f^2 & \text{for } i = 3, \\ 1 & \text{otherwise}. \end{cases} \tag{A.5}$$

Equation (A.4) corrects a typo appearing in [81], as previously noted in [9, 83].

The normalization factors $a_{i\ell}$ appearing in eq. (18) are defined as

$$a_{i\ell} := \frac{1}{\sqrt{2}} \begin{cases} 1 & \text{for } i = 1, 2, 3, 6, \\ 1/\sqrt{\gamma_1} & \text{for } i = 4, 5, 8, 9, \\ 1/\sqrt{\gamma_2} & \text{for } i = 7, 10. \end{cases} \tag{A.6}$$

## A.2 The matrix operator $\mathcal{M}^{ij}$

The radial operators appearing in eq. (26) are given by

$$\mathcal{M}_r^{1j} = \left\{ \frac{f}{2r^2}\left(1 - \frac{4M}{r}\right), 0, \frac{f'f^2}{2}\partial_r - \frac{f^2}{2r^2}\left(1 - \frac{4M}{r}\right), 0, -\mathcal{M}_r^{11}, -\frac{f^2}{2r^2}\left(1 - \frac{6M}{r}\right), 0, 0, 0, 0 \right\},$$

$$\mathcal{M}_r^{2j} = \left\{ -\frac{f'f}{2}\left(\partial_r + \frac{1}{r}\right), \frac{f}{2}\left(f'\partial_r + \frac{f}{r^2}\right), \frac{f'f^2}{2}\left(\partial_r + \frac{1}{r}\right), -\frac{f^2}{2r^2}, \frac{f'f}{2r}, \frac{f'f^2}{r}, 0, 0, 0, 0 \right\},$$

$$\mathcal{M}_r^{3j} = \left\{ -\frac{f}{2r^2}, 0, \frac{f}{2r^2}\left(1 - \frac{4M}{r}\right), 0, \frac{f}{2r^2}, \frac{f}{2r^2}\left(1 - \frac{4M}{r}\right), 0, 0, 0, 0 \right\},$$

$$\mathcal{M}_r^{4j} = \left\{ 0, -\frac{\gamma_1 f}{2r^2}, 0, \frac{f'f}{4}\left(\partial_r - \frac{3}{r}\right), -\frac{f'f}{4}\left(\partial_r + \frac{2}{r}\right), -\frac{\gamma_1 f'f}{4r}, \frac{f'f}{4r}, 0, 0, 0 \right\},$$

$$\mathcal{M}_r^{5j} = \left\{ -\frac{\gamma_1 f}{2r^2}, 0, \frac{\gamma_1 f^2}{2r^2}, 0, \frac{f}{r^2}\left(1 - \frac{9M}{2r}\right), \frac{\gamma_1 f}{2r^2}\left(1 - \frac{3M}{r}\right), -\frac{f}{2r^2}\left(1 - \frac{3M}{r}\right), 0, 0, 0 \right\},$$

$$\mathcal{M}_r^{6j} = \left\{ -\frac{f}{2r^2}, 0, \frac{f}{2r^2}\left(1 - \frac{4M}{r}\right), 0, \frac{f}{2r^2}, \frac{f}{2r^2}\left(1 - \frac{4M}{r}\right), 0, 0, 0, 0 \right\},$$

$$\mathcal{M}_r^{7j} = \left\{ 0, 0, 0, 0, -\frac{\gamma_2 f}{2\gamma_1 r^2}, 0, -\frac{f}{2r^2}, 0, 0, 0 \right\},$$

$$\mathcal{M}_r^{8j} = \left\{ 0, 0, 0, 0, 0, 0, 0, \frac{f'f}{4}\left(\partial_r - \frac{3}{r}\right), -\frac{f'f}{4}\left(\partial_r + \frac{2}{r}\right), \frac{f'f}{4r} \right\},$$

$$\mathcal{M}_r^{9j} = \left\{ 0, 0, 0, 0, 0, 0, 0, 0, \frac{f}{r^2}\left(1 - \frac{9M}{2r}\right), -\frac{f}{2r^2}\left(1 - \frac{3M}{r}\right) \right\},$$

$$\mathcal{M}_r^{10j} = \left\{ 0, 0, 0, 0, 0, 0, 0, 0, -\frac{\gamma_2 f}{2\gamma_1 r^2}, -\frac{f}{2r^2} \right\}, \tag{A.7}$$

where $f' = 2M/r^2$ and $\gamma_1$ and $\gamma_2$ are defined in eq. (A.2) above. The radial derivatives are taken at fixed $(\Delta)J^a$ and $\phi_p$. All components of the radial matrix $\mathcal{M}_t^{ij}$ vanish except

$$\mathcal{M}_t^{13} = \mathcal{M}_t^{23} = -\frac{ff'H}{2},$$

$$-\frac{1}{2}\mathcal{M}_t^{21} = \frac{1}{2}\mathcal{M}_t^{22} = \mathcal{M}_t^{44} = -\mathcal{M}_t^{45} = \mathcal{M}_t^{88} = -\mathcal{M}_t^{89} = \frac{f'(1-H)}{4}. \tag{A.8}$$

The matrix operator $\mathcal{M}^{ij} = \mathcal{M}_r^{ij} + \mathcal{M}_t^{ij}(\partial_t)_r$ agrees with the one given in eqs. (A1)-(A10) of [41] with $(\partial_v)_u = \frac{f}{2}(\partial_r)_{(\Delta)J^a, \phi_p} + \frac{1-H}{2}\left(-im\Omega + F^{(\Delta)J^a}\partial_{(\Delta)J^a}\right)$ and $(\partial_r)_t$ replaced using eq. (27b).

### A.3 Harmonic decomposition of the Lorenz gauge condition

We list the operators $Z_{raj}$ and the components of the radial vector $Z_{taj}$ for $a = 1, 2, 3, 4$ and $j = 1, \ldots, 10$, which appear in the harmonic decomposition of the Lorenz gauge condition (32):

$$Z_{r12} = Z_{r21} = -(\partial_x)_{(\Delta)J^a} - \frac{f}{r}, \tag{A.9a}$$

$$Z_{r14} = Z_{r25} = Z_{r37} = Z_{r410} = \frac{f}{r}, \tag{A.9b}$$

$$Z_{r23} = f(\partial_x)_{(\Delta)J^a} + \frac{f^2}{r}, \tag{A.9c}$$

$$Z_{r26} = \frac{2f^2}{r}, \tag{A.9d}$$

$$Z_{r35} = Z_{r49} = -(\partial_x)_{(\Delta)J^a} - \frac{2f}{r}, \tag{A.9e}$$

$$Z_{r36} = -\ell(\ell+1)\frac{f}{r}, \tag{A.9f}$$

and

$$Z_{t11} = Z_{t22} = Z_{t34} = Z_{t48} = 1, \tag{A.10a}$$

$$Z_{t12} = Z_{t21} = Z_{t35} = Z_{t49} = H, \tag{A.10b}$$

$$Z_{t13} = f, \tag{A.10c}$$

$$Z_{t23} = -fH. \tag{A.10d}$$

All other terms vanish. With these explicit expression, the conditions (32) agree with eqs. (A13)-(A16) of [41] upon substituting $\partial_t$ and $\partial_r$ in that reference with eq. (27).

## B Material for section 3

### B.1 Inspiral expansion of the point-particle stress-energy tensor

After performing the inspiral expansion of the first-order term in eq. (21), we find that the mode amplitudes of the stress-energy tensor at 1PA order are explicitly given by

$$t_{i\ell m}^{(2)} = -\frac{1}{4}\mathcal{E}_{(0)}\alpha_{i\ell m}^{(2)} \begin{cases} Y_{\ell m}^*(\frac{\pi}{2}, 0) & i = 1, \ldots, 7, \\ \partial_\theta Y_{\ell m}^*(\frac{\pi}{2}, 0) & i = 8, 9, 10, \end{cases} \tag{B.1}$$

where the coefficients $\alpha^{(2)}_{i\ell m}$ read (dropping the $\ell$ and $m$ indices)

$$\alpha^{(2)}_1 = -\frac{(r_{(0)}-8M)f_{(0)}r_{(1)}}{r_{(0)}^3}, \quad \alpha^{(2)}_2 = -\frac{2f_{(0)}(\partial_\Omega r_{(0)})F^\Omega_{(0)}}{r_{(0)}},$$

$$\alpha^{(2)}_3 = -\frac{(r_{(0)}-6M)r_{(1)}}{r_{(0)}^3}, \quad \alpha^{(2)}_4 = \frac{8imM\Omega r_{(1)}}{r_{(0)}^2}, \quad \alpha^{(2)}_5 = -2im\Omega(\partial_\Omega r_{(0)})F^\Omega_{(0)},$$

$$\alpha^{(2)}_6 = \frac{\Omega^2 r_{(1)}}{f_{(0)}}, \quad \alpha^{(2)}_7 = \frac{[\ell(\ell+1)-2m^2]\Omega^2 r_{(1)}}{f_{(0)}}, \quad \alpha^{(2)}_8 = \frac{\alpha^{(1)}_4}{im},$$

$$\alpha^{(2)}_9 = \frac{\alpha^{(1)}_5}{im}, \quad \alpha^{(2)}_{10} = 2im\,\alpha^{(1)}_6. \tag{B.2}$$

The coefficients of $F^\Omega_{(0)}$ in these terms are the explicit expressions of $t^{(2)B}_{i\ell m}$ in eq. (46b), while the remaining terms give $t^{(2)A}_{i\ell m}$.

## B.2 Coefficients of the near-ISCO inspiral solutions

This appendix contains explicit expressions for some of the coefficients appearing in the near-ISCO solutions of the inspiral motion up to 2PA order. All self-force and forcing terms are evaluated at the ISCO frequency. The coefficients appearing at the lowest orders in the near-ISCO solution of the orbital radius (eqs. (64a), (65a) and (66a)) are given by

$$r^{(2,1)}_{(0)} = -24\sqrt{6}M^2, \qquad r^{(4,2)}_{(0)} = 720M^3, \tag{B.3}$$

$$r^{(6,3)}_{(0)} = -3840\sqrt{6}M^4, \qquad r^{(8,4)}_{(0)} = 126720M^5, \tag{B.4}$$

$$r^{(5,0)}_{(1)} = -54M^2 f^r_{(1)}, \tag{B.5}$$

$$r^{(7,1)}_{(1)} = 54M^2\left(14\sqrt{6}Mf^r_{(1)} - \partial_\Omega f^r_{(1)}\right), \tag{B.6}$$

$$r^{(9,2)}_{(1)} = -27M^2\left(1560M^2 f^r_{(1)} - 28\sqrt{6}M\partial_\Omega f^r_{(1)} + \partial_\Omega^2 f^r_{(1)}\right), \tag{B.7}$$

$$r^{(4,-3)}_{(2)} = \frac{9}{4}\sqrt{\frac{3}{2}}\left(f^t_{(1)}\right)^2, \tag{B.8}$$

$$r^{(6,-2)}_{(2)} = -\frac{9}{8}f^t_{(1)}\left(120Mf^t_{(1)} - \sqrt{6}\partial_\Omega f^t_{(1)}\right), \tag{B.9}$$

$$r^{(8,-1)}_{(2)} = \frac{9}{8}f^t_{(1)}\left[12M\left(52\sqrt{6}Mf^t_{(1)} - 3\partial_\Omega f^t_{(1)}\right) - f^r_{(2)B}\right]. \tag{B.10}$$

The coefficients appearing at the lowest orders in the near-ISCO solutions for the forcing terms $F^\Omega_{(n)}$ with $n = 0, 1, 2$ (eqs. (64b), (65b) and (66b)) read

$$F^{(3,-1)}_{(0)} = \frac{f^t_{(1)}}{48M^2}, \tag{B.11}$$

$$F^{(5,0)}_{(0)} = -\frac{1}{48M^2}\left(7\sqrt{6}Mf^t_{(1)} - \partial_\Omega f^t_{(1)}\right), \tag{B.12}$$

$$F^{(7,1)}_{(0)} = \frac{1}{96M^2}\left(100M^2 f^t_{(1)} - 14\sqrt{6}M\partial_\Omega f^t_{(1)} + \partial_\Omega^2 f^t_{(1)}\right), \tag{B.13}$$

$$F_{(0)}^{(9,2)} = \frac{1}{288M^2}\left(1836\sqrt{6}M^3 f_{(1)}^t + 300M^2 \partial_\Omega f_{(1)}^t - 21\sqrt{6}M \partial_\Omega^2 f_{(1)}^t + \partial_\Omega^3 f_{(1)}^t\right), \tag{B.14}$$

$$\begin{aligned}F_{(0)}^{(11,3)} = -\frac{1}{1152M^2}\Big(&243552M^4 f_{(1)}^t - 7344\sqrt{6}M^3 \partial_\Omega f_{(1)}^t - 600M^2 \partial_\Omega^2 f_{(1)}^t \\ &+ 28\sqrt{6}M \partial_\Omega^3 f_{(1)}^t - \partial_\Omega^4 f_{(1)}^t\Big),\end{aligned} \tag{B.15}$$

$$F_{(1)}^{(6,-2)} = -\frac{f_{(1)}^t}{2304M^4}\left(36\sqrt{6}M^2 f_{(1)}^r - 9M \partial_\Omega f_{(1)}^r - f_{(2)B}^t\right), \tag{B.16}$$

$$\begin{aligned}F_{(1)}^{(8,-1)} = \frac{1}{2304M^4}\Big[&48M^2 f_{(2)A}^t - 14\sqrt{6}M f_{(1)}^t f_{(2)B}^t - 162\sqrt{6}M^2 f_{(1)}^t \partial_\Omega f_{(1)}^r \\ &+ f_{(2)B}^t \partial_\Omega f_{(1)}^t + 9M \partial_\Omega f_{(1)}^r \partial_\Omega f_{(1)}^t + 36M^2 f_{(1)}^r\left(108M f_{(1)}^t - \sqrt{6}\partial_\Omega f_{(1)}^t\right) \\ &+ f_{(1)}^t \partial_\Omega f_{(2)B}^t + 9M f_{(1)}^t \partial_\Omega^2 f_{(1)}^r + 432M^3\left(F_{(0)}^{\delta M}\partial_{\delta M}f_{(1)}^r + F_{(0)}^{\delta J}\partial_{\delta J}f_{(1)}^r\right)\Big],\end{aligned} \tag{B.17}$$

$$\begin{aligned}F_{(1)}^{(10,0)} = -\frac{1}{4608M^2}\Big[&672\sqrt{6}M^3 f_{(2)A}^t - 788M^2 f_{(1)}^t f_{(2)B}^t - 16164M^3 f_{(1)}^t \partial_\Omega f_{(1)}^r \\ &+ 28\sqrt{6}M f_{(2)B}^t \partial_\Omega f_{(1)}^t + 324\sqrt{6}M^2 \partial_\Omega f_{(1)}^r \partial_\Omega f_{(1)}^t - 96M^2 \partial_\Omega f_{(2)A}^t \\ &+ 28\sqrt{6}M f_{(1)}^t \partial_\Omega f_{(2)B}^t - 2\partial_\Omega f_{(1)}^t \partial_\Omega f_{(2)B}^t + 288\sqrt{6}M^2 f_{(1)}^t \partial_\Omega^2 f_{(1)}^r \\ &- 18M \partial_\Omega f_{(1)}^t \partial_\Omega^2 f_{(1)}^r - f_{(2)B}^t \partial_\Omega^2 f_{(1)}^t - 9M \partial_\Omega f_{(1)}^r \partial_\Omega^2 f_{(1)}^t \\ &+ 36M^2 f_{(1)}^r\left(2108\sqrt{6}M^2 f_{(1)}^t - 216M \partial_\Omega f_{(1)}^t + \sqrt{6}\partial_\Omega^2 f_{(1)}^t\right) - f_{(1)}^t \partial_\Omega^2 f_{(2)B}^t \\ &- 9M f_{(1)}^t \partial_\Omega^3 f_{(1)}^r + 6048\sqrt{6}M^4\left(F_{(0)}^{\delta M}\partial_{\delta M}f_{(1)}^r + F_{(0)}^{\delta J}\partial_{\delta J}f_{(1)}^r\right) \\ &- 864M^3\Big(\partial_\Omega F_{(0)}^{\delta M}\partial_{\delta M}f_{(1)}^r + \partial_\Omega F_{(0)}^{\delta J}\partial_{\delta J}f_{(1)}^r \\ &+ F_{(0)}^{\delta M}\partial_\Omega \partial_{\delta M}f_{(1)}^r + F_{(0)}^{\delta J}\partial_\Omega \partial_{\delta J}f_{(1)}^r\Big)\Big],\end{aligned} \tag{B.18}$$

$$F_{(2)}^{(3,-6)} = \sqrt{\frac{3}{2}}\frac{\left(f_{(1)}^t\right)^3}{2048M^5}, \tag{B.19}$$

$$F_{(2)}^{(5,-5)} = -\frac{\left(f_{(1)}^t\right)^2}{12288M^5}\left(252M f_{(1)}^t - 5\sqrt{6}\partial_\Omega f_{(1)}^t\right), \tag{B.20}$$

$$\begin{aligned}F_{(2)}^{(7,-4)} = -\frac{f_{(1)}^t}{221184M^6}\Big[&18M f_{(2)B}^r f_{(1)}^t - 324\sqrt{6}M^3\left(f_{(1)}^t\right)^2 - 36\sqrt{6}M\left(\partial_\Omega f_{(1)}^t\right)^2 \\ &+ f_{(1)}^t\left(2f_{(3)E}^t + 27M\left(200M \partial_\Omega f_{(1)}^t - \sqrt{6}\partial_\Omega^2 f_{(1)}^t\right)\right)\Big],\end{aligned} \tag{B.21}$$

$$\begin{aligned}F_{(2)}^{(9,-3)} = \frac{f_{(1)}^t}{221184M^6}\Big[&15552M^4\left(f_{(1)}^r\right)^2 + 1228608M^4\left(f_{(1)}^t\right)^2 + 2f_{(2)B}^t f_{(3)C}^t \\ &+ 2f_{(1)}^t f_{(3)D}^t + 28\sqrt{6}M f_{(1)}^t f_{(3)E}^t + 18M f_{(2)B}^t \partial_\Omega f_{(1)}^r + 18M f_{(3)C}^t \partial_\Omega f_{(1)}^r \\ &+ 162M^2\left(\partial_\Omega f_{(1)}^r\right)^2 - 72\sqrt{6}M^2 f_{(1)}^r\left(f_{(2)B}^t + f_{(3)C}^t + 18M \partial_\Omega f_{(1)}^r\right) \\ &+ 18M f_{(2)B}^r\left(10\sqrt{6}M f_{(1)}^t - \partial_\Omega f_{(1)}^t\right) - 7956\sqrt{6}M^3 f_{(1)}^t \partial_\Omega f_{(1)}^t \\ &- 2f_{(3)E}^t \partial_\Omega f_{(1)}^t - 864M^2\left(\partial_\Omega f_{(1)}^t\right)^2 - 2f_{(1)}^t \partial_\Omega f_{(3)E}^t \\ &- 2268M^2 f_{(1)}^t \partial_\Omega^2 f_{(1)}^t + 18\sqrt{6}\partial_\Omega f_{(1)}^t \partial_\Omega^2 f_{(1)}^t + 9\sqrt{6}M f_{(1)}^t \partial_\Omega^3 f_{(1)}^t\Big].\end{aligned} \tag{B.22}$$

## C  Material for section 4

### C.1  Transition-to-plunge equations

In this appendix we list the transition-to-plunge equations up to 7PLT order. The $n$PLT corrections ($n = 3, \ldots, 7$) to the redshift are algebraically determined as

$$U_{[3]} = 0, \tag{C.1a}$$

$$U_{[4]} = 16\sqrt{2}M^3 \left[ 324M \left( F_{[0]}^{\Delta\Omega} \right)^2 + 5\sqrt{6}\Delta\Omega^3 \right], \tag{C.1b}$$

$$U_{[5]} = 0, \tag{C.1c}$$

$$\begin{aligned}
U_{[6]} = -48\sqrt{2}M^3 \Big[ & 9F_{[0]}^{\Delta\Omega} \left( \sqrt{6}f_{[5]}^t - 24MF_{[2]}^{\Delta\Omega} \right) - 22M\Delta\Omega^4 \\
& + 432M^2 \left( F_{[0]}^{\Delta\Omega} \right)^2 \left( 2\sqrt{6}\Delta\Omega - 27M \left( \partial_{\Delta\Omega}F_{[0]}^{\Delta\Omega} \right)^2 \right) \Big],
\end{aligned} \tag{C.1d}$$

$$U_{[7]} = 1296\sqrt{2}M^4 F_{[0]}^{\Delta\Omega} \left( 8F_{[3]}^{\Delta\Omega} + 3\sqrt{6}f_{[5]}^r \partial_{\Delta\Omega}F_{[0]}^{\Delta\Omega} \right), \tag{C.1e}$$

while for the orbital radius we obtain

$$r_{[3]} = -54M^2 f_{[5]}^r, \tag{C.2a}$$

$$\begin{aligned}
r_{[4]} = 96M^4 \Big[ & 810M \left( F_{[0]}^{\Delta\Omega} \right)^2 - \sqrt{6} \left( 40\Delta\Omega^3 + 27F_{[2]}^{\Delta\Omega} \partial_{\Delta\Omega}F_{[0]}^{\Delta\Omega} \right) \\
& + 27F_{[0]}^{\Delta\Omega} \left( 108M\Delta\Omega\,\partial_{\Delta\Omega}F_{[0]}^{\Delta\Omega} - \sqrt{6}\partial_{\Delta\Omega}F_{[2]}^{\Delta\Omega} \right) \Big],
\end{aligned} \tag{C.2b}$$

$$r_{[5]} = -54M^2 \left[ f_{[7]}^r - 2\sqrt{6}M \left( 7\Delta\Omega f_{[5]}^r - 24M \left( F_{[3]}^{\Delta\Omega} \partial_{\Delta\Omega}F_{[0]}^{\Delta\Omega} + F_{[0]}^{\Delta\Omega} \partial_{\Delta\Omega}F_{[3]}^{\Delta\Omega} \right) \right) \right], \tag{C.2c}$$

$$\begin{aligned}
r_{[6]} = -18M^2 \Big[ & 3f_{[8]}^r + 13824M^4 \left( F_{[0]}^{\Delta\Omega} \right)^2 \left( 10\sqrt{6}\Delta\Omega - 27M \left( \partial_{\Delta\Omega}F_{[0]}^{\Delta\Omega} \right)^2 \right) \\
& - 16M^2 \big( 440M\Delta\Omega^4 + 972M\Delta\Omega F_{[2]}^{\Delta\Omega} \partial_{\Delta\Omega}F_{[0]}^{\Delta\Omega} \\
& - 9\sqrt{6} \left( F_{[4]}^{\Delta\Omega} \partial_{\Delta\Omega}F_{[0]}^{\Delta\Omega} + F_{[2]}^{\Delta\Omega} \partial_{\Delta\Omega}F_{[2]}^{\Delta\Omega} \right) \big) \\
& + 36MF_{[0]}^{\Delta\Omega} \big( 12\sqrt{6}Mf_{[5]}^t - 240M^2 F_{[2]}^{\Delta\Omega} + 5568\sqrt{6}M^3\Delta\Omega^2 \partial_{\Delta\Omega}F_{[0]}^{\Delta\Omega} \\
& - 432M^2\Delta\Omega\,\partial_{\Delta\Omega}F_{[2]}^{\Delta\Omega} + 4\sqrt{6}M\partial_{\Delta\Omega}F_{[4]}^{\Delta\Omega} - \partial_{\Delta\Omega}f_{[7]}^t \big) \Big],
\end{aligned} \tag{C.2d}$$

$$\begin{aligned}
r_{[7]} = -54M^2 \Big[ & f_{[9]}^r - 2M \big( 7\sqrt{6}\Delta\Omega f_{[7]}^r - 6M \big( 65\Delta\Omega^2 f_{[5]}^r - 432M\Delta\Omega F_{[3]}^{\Delta\Omega} \partial_{\Delta\Omega}F_{[0]}^{\Delta\Omega} \\
& + 4\sqrt{6} \left( F_{[5]}^{\Delta\Omega} \partial_{\Delta\Omega}F_{[0]}^{\Delta\Omega} + F_{[3]}^{\Delta\Omega} \partial_{\Delta\Omega}F_{[2]}^{\Delta\Omega} + F_{[2]}^{\Delta\Omega} \partial_{\Delta\Omega}F_{[3]}^{\Delta\Omega} \right) \big) \big) \\
& - 12MF_{[0]}^{\Delta\Omega} \big( 240M^2 F_{[3]}^{\Delta\Omega} + 126\sqrt{6}M^2 f_{[5]}^r \partial_{\Delta\Omega}F_{[0]}^{\Delta\Omega} + 432M^2\Delta\Omega\partial_{\Delta\Omega}F_{[3]}^{\Delta\Omega} \\
& - 4\sqrt{6}M\partial_{\Delta\Omega}F_{[5]}^{\Delta\Omega} + \partial_{\Delta\Omega}f_{[8]}^t \big) \Big].
\end{aligned} \tag{C.2e}$$

The sources appearing in eq. (86) are given by

$$S_{[2]}^{\Delta\Omega} := -\frac{f_{[7]}^t}{432\sqrt{6}M^3} - \frac{\Delta\Omega f_{[5]}^t}{108M^2} + \frac{11}{9}\Delta\Omega^2 F_{[0]}^{\Delta\Omega} + 26\sqrt{6}M \left( F_{[0]}^{\Delta\Omega} \right)^2 \partial_{\Delta\Omega}F_{[0]}^{\Delta\Omega}, \tag{C.3}$$

$$S_{[3]}^{\Delta\Omega} := -\frac{f_{[8]}^t}{432\sqrt{6}M^3} + \frac{F_{[0]}^{\Delta\Omega}f_{[5]}^r}{12M} - \frac{F_{[0]}^{\Delta\Omega}\partial_{\Delta\Omega}f_{[7]}^r}{48\sqrt{6}M^2}, \tag{C.4}$$

$$\begin{aligned}
S_{[4]}^{\Delta\Omega} := &-\frac{1}{2592M^3\left(F_{[0]}^{\Delta\Omega}\right)^2}\Bigg[ 2332800M^5\left(F_{[0]}^{\Delta\Omega}\right)^5 + 3\sqrt{6}f_{[5]}^t\left(F_{[2]}^{\Delta\Omega}\right)^2 \\
&- 2F_{[0]}^{\Delta\Omega}F_{[2]}^{\Delta\Omega}\left(\sqrt{6}f_{[7]}^t + 24\left(M\Delta\Omega\left(f_{[5]}^t + \sqrt{6}MF_{[2]}^{\Delta\Omega}\right) - 54M^3F_{[2]}^{\Delta\Omega}\left(\partial_{\Delta\Omega}F_{[0]}^{\Delta\Omega}\right)^2\right)\right) \\
&+ \left(F_{[0]}^{\Delta\Omega}\right)^2\left(\sqrt{6}f_{[9]}^t + 12M\left(2\Delta\Omega f_{[7]}^t - 3M\left(\Delta\Omega^2\left(7\sqrt{6}f_{[5]}^t - 88MF_{[2]}^{\Delta\Omega}\right)\right.\right.\right. \\
&\left.\left.\left. + 144MF_{[2]}^{\Delta\Omega}\partial_{\Delta\Omega}F_{[0]}^{\Delta\Omega}\partial_{\Delta\Omega}F_{[2]}^{\Delta\Omega}\right)\right)\right) \\
&- 3M\left(F_{[0]}^{\Delta\Omega}\right)^3\left(2560\sqrt{6}M^3\Delta\Omega^3 - 3\left(288M^2\left(\partial_{\Delta\Omega}F_{[2]}^{\Delta\Omega}\right)^2 + \sqrt{6}\partial_{\Delta\Omega}f_{[8]}^r\right.\right. \\
&\left.\left. + 12M\partial_{\Delta\Omega}F_{[0]}^{\Delta\Omega}\left(156Mf_{[5]}^t - \sqrt{6}\partial_{\Delta\Omega}f_{[7]}^t\right)\right)\right) \\
&+ 108\left(F_{[0]}^{\Delta\Omega}\right)^4\left(33984M^5\Delta\Omega\partial_{\Delta\Omega}F_{[0]}^{\Delta\Omega} - \sqrt{6}M^2\left(624M^2\partial_{\Delta\Omega}F_{[2]}^{\Delta\Omega} + \partial_{\Delta\Omega}^2f_{[7]}^t\right)\right)\Bigg], \tag{C.5}
\end{aligned}$$

$$\begin{aligned}
S_{[5]}^{\Delta\Omega} := &-\frac{1}{2592M^3\left(F_{[0]}^{\Delta\Omega}\right)^2}\Bigg[ 6\sqrt{6}f_{[5]}^t F_{[2]}^{\Delta\Omega}F_{[3]}^{\Delta\Omega} - 2F_{[0]}^{\Delta\Omega}\left(\sqrt{6}F_{[2]}^{\Delta\Omega}f_{[8]}^t\right. \\
&+ F_{[3]}^{\Delta\Omega}\left(\sqrt{6}f_{[7]}^t + 24M\Delta\Omega\left(f_{[5]}^t + 2\sqrt{6}MF_{[2]}^{\Delta\Omega}\right) - 2592M^3F_{[2]}^{\Delta\Omega}\left(\partial_{\Delta\Omega}F_{[0]}^{\Delta\Omega}\right)^2\right)\Big) \\
&+ \left(F_{[0]}^{\Delta\Omega}\right)^2\left(\sqrt{6}f_{[10]}^t + 3M\left(9\sqrt{6}f_{[5]}^t f_{[5]}^r + 8\Delta\Omega f_{[8]}^t\right.\right. \\
&+ 96M^2F_{[3]}^{\Delta\Omega}\left(11\Delta\Omega^2 - 18\partial_{\Delta\Omega}F_{[0]}^{\Delta\Omega}\partial_{\Delta\Omega}F_{[2]}^{\Delta\Omega}\right)\Big) \\
&+ 9MF_{[2]}^{\Delta\Omega}\left(24Mf_{[5]}^r - 576M^2\partial_{\Delta\Omega}F_{[0]}^{\Delta\Omega}\partial_{\Delta\Omega}F_{[3]}^{\Delta\Omega} - \sqrt{6}\partial_{\Delta\Omega}f_{[7]}^r\right) \\
&+ 9\sqrt{6}M\left(F_{[0]}^{\delta M}\partial_{\delta M}f_{[5]}^r + F_{[0]}^{\delta J}\partial_{\delta J}f_{[5]}^r\right)\Big) \\
&- 9M\left(F_{[0]}^{\Delta\Omega}\right)^3\left(24Mf_{[7]}^r - 576M^2\partial_{\Delta\Omega}F_{[2]}^{\Delta\Omega}\partial_{\Delta\Omega}F_{[3]}^{\Delta\Omega}\right. \\
&+ 24M\Delta\Omega\left(6\sqrt{6}Mf_{[5]}^r - \partial_{\Delta\Omega}f_{[7]}^r\right) - \sqrt{6}\partial_{\Delta\Omega}f_{[9]}^r + 12\sqrt{6}M\partial_{\Delta\Omega}F_{[0]}^{\Delta\Omega}\partial_{\Delta\Omega}f_{[8]}^t\Big) \\
&- 108\sqrt{6}M^2\left(F_{[0]}^{\Delta\Omega}\right)^4\left(624M^2\partial_{\Delta\Omega}F_{[3]}^{\Delta\Omega} + \partial_{\Delta\Omega}^2f_{[8]}^t\right)\Bigg], \tag{C.6}
\end{aligned}$$

and higher-order terms are straightforwardly computed from the transition-to-plunge expansion of the $t$ component of the equation of motion.

## C.2 Linearized Einstein operators

The linearized Einstein operators $E_{ij\ell m}^{[n]}$, $n = 3, 4, 5, 6, 7$, explicitly read

$$\begin{aligned}
E_{ij\ell m}^{[3]} = &\left[\mathcal{M}_t^{ij} - \delta_{ij}\frac{im}{2}\Omega_*\left(1 - H^2\right) + \frac{\delta_{ij}}{4}\left(\partial_x H + 2H\partial_x\right)\right]F_{[2]}^{\Delta\Omega}\partial_{\Delta\Omega} \\
&- \delta_{ij}\frac{im}{4}\left(1 - H^2\right)F_{[0]}^{\Delta\Omega}\left(1 + 2\Delta\Omega\partial_{\Delta\Omega}\right), \tag{C.7}
\end{aligned}$$

$$
E_{ij\ell m}^{[4]a} = \left[\mathcal{M}_t^{ij} - \delta_{ij}\frac{im}{2}\Omega_*\left(1-H^2\right) + \frac{\delta_{ij}}{4}\left(\partial_x H + 2H\partial_x\right)\right] F_{[3]}^{\Delta\Omega a}\partial_{\Delta\Omega}
$$
$$
- \delta_{ij}\delta_1^a\frac{1-H^2}{4}\left[m^2\Delta\Omega^2 - \left(F_{[0]}^{\Delta\Omega}\frac{\partial F_{[2]}^{\Delta\Omega}}{\partial\Delta\Omega} + F_{[2]}^{\Delta\Omega}\frac{\partial F_{[0]}^{\Delta\Omega}}{\partial\Delta\Omega}\right)\partial_{\Delta\Omega} - 2F_{[0]}^{\Delta\Omega}F_{[2]}^{\Delta\Omega}\partial_{\Delta\Omega}^2\right],
\tag{C.8}
$$

$$
E_{ij\ell m}^{[5]a} = \left[\mathcal{M}_t^{ij} - \delta_{ij}\frac{im}{2}\Omega_*\left(1-H^2\right) + \frac{\delta_{ij}}{4}\left(\partial_x H + 2H\partial_x\right)\right]\left[F_{[4]}^{\Delta\Omega a}\partial_{\Delta\Omega} + F_{[0]}^{\delta M^\pm a}\partial_{\delta M^\pm}\right]
$$
$$
- \delta_{ij}\frac{1-H^2}{4}\left[im\delta_1^a F_{[2]}^{\Delta\Omega}\left(1+2\Delta\Omega\partial_{\Delta\Omega}\right) - \left(F_{[0]}^{\Delta\Omega}\frac{\partial F_{[3]}^{\Delta\Omega}}{\partial\Delta\Omega} + F_{[3]}^{\Delta\Omega a}\frac{\partial F_{[0]}^{\Delta\Omega}}{\partial\Delta\Omega}\right)\partial_{\Delta\Omega}\right.
$$
$$
\left. - 2F_{[0]}^{\Delta\Omega}F_{[3]}^{\Delta\Omega a}\partial_{\Delta\Omega}^2\right],
\tag{C.9}
$$

$$
E_{ij\ell m}^{[6]a} = \left[\mathcal{M}_t^{ij} - \delta_{ij}\frac{im}{2}\Omega_*\left(1-H^2\right) + \frac{\delta_{ij}}{4}\left(\partial_x H + 2H\partial_x\right)\right] F_{[5]}^{\Delta\Omega a}\partial_{\Delta\Omega}
$$
$$
- \delta_{ij}\frac{1-H^2}{4}\left[imF_{[3]}^{\Delta\Omega a}\left(1+2\Delta\Omega\partial_{\Delta\Omega}\right)\right.
$$
$$
- \left(F_{[0]}^{\Delta\Omega}\frac{\partial F_{[4]}^{\Delta\Omega a}}{\partial\Delta\Omega} + F_{[4]}^{\Delta\Omega a}\frac{\partial F_{[0]}^{\Delta\Omega}}{\partial\Delta\Omega} + \delta_1^a F_{[2]}^{\Delta\Omega}\frac{\partial F_{[2]}^{\Delta\Omega}}{\partial\Delta\Omega}\right)\partial_{\Delta\Omega}
$$
$$
\left. - \left(2F_{[0]}^{\Delta\Omega}F_{[4]}^{\Delta\Omega a} + \delta_1^a\left(F_{[2]}^{\Delta\Omega}\right)^2\right)\partial_{\Delta\Omega}^2 - 2F_{[0]}^{\Delta\Omega}F_{[0]}^{\delta M^\pm a}\partial_{\Delta\Omega}\partial_{\delta M^\pm}\right],
\tag{C.10}
$$

$$
E_{ij\ell m}^{[7]ab} = \left[\mathcal{M}_t^{ij} - \delta_{ij}\frac{im}{2}\Omega_*\left(1-H^2\right) + \frac{\delta_{ij}}{4}\left(\partial_x H + 2H\partial_x\right)\right]\left[F_{[6]}^{\Delta\Omega ab}\partial_{\Delta\Omega} + F_{[2]}^{\delta M^\pm(a}\delta_1^{b)}\partial_{\delta M^\pm}\right]
$$
$$
- \delta_{ij}\frac{1-H^2}{4}\left[imF_{[4]}^{\Delta\Omega(a}\left(1+2\Delta\Omega\partial_{\Delta\Omega}\right) - \left(2F_{[0]}^{\Delta\Omega}F_{[5]}^{\Delta\Omega(a} + 2F_{[2]}^{\Delta\Omega}F_{[3]}^{\Delta\Omega(a}\right)\partial_{\Delta\Omega}^2\right.
$$
$$
- \left(F_{[0]}^{\Delta\Omega}\frac{\partial F_{[5]}^{\Delta\Omega(a}}{\partial\Delta\Omega} + F_{[5]}^{\Delta\Omega(a}\frac{\partial F_{[0]}^{\Delta\Omega}}{\partial\Delta\Omega} + F_{[2]}^{\Delta\Omega}\frac{\partial F_{[3]}^{\Delta\Omega(a}}{\partial\Delta\Omega} + F_{[3]}^{\Delta\Omega(a}\frac{\partial F_{[2]}^{\Delta\Omega}}{\partial\Delta\Omega}\right)\partial_{\Delta\Omega}
$$
$$
\left. + 2im\Delta\Omega F_{[0]}^{\delta M^\pm(a}\partial_{\delta M^\pm}\right]\delta_1^{b)}.
\tag{C.11}
$$

Here we have introduced the Kronecker delta $\delta_1^a$ such that $\delta_1^a\delta M^a = 1$. In the last equation, the notation $(ab)$ stands for weighted symmetrization over $ab$: $T^{(ab)} = \frac{1}{2}T^{ab} + \frac{1}{2}T^{ba}$. For $n \geq 5$, $E_{ij\ell m}^{[n]}$ depends upon $F_{[m]}^{\delta M^\pm}$, $0 \leq m \leq n-5$. Higher-order terms can be obtained straightforwardly.

## C.3 Transition-to-plunge expansion of the point-particle stress-energy tensor

The $n$PLT ($n \geq 3$) mode amplitudes of the first-order stress-energy tensor can be obtained from the transition-to-plunge expansion of eq. (21). The mode amplitudes of the stress-energy tensor at 3PLT order, $t_{i\ell m}^{[8]}$, are given by

$$
t_{i\ell m}^{[8]} = F_{[0]}^{\Delta\Omega}t_{i\ell m}^{[8]A} := -F_{[0]}^{\Delta\Omega}\frac{\mathcal{E}_*}{4}\alpha_{i\ell m}^{[8]A}\begin{cases} Y_{\ell m}^*(\frac{\pi}{2},0) & i = 1,\ldots,7, \\ \partial_\theta Y_{\ell m}^*(\frac{\pi}{2},0) & i = 8,9,10, \end{cases}
\tag{C.12}
$$

where (dropping the $\ell$ and $m$ indices)

$$\alpha_{1,3,4,6,7,8,10}^{[8]A} = 0, \quad \alpha_2^{[8]A} = 16\sqrt{\frac{2}{3}}M\delta(r-6M),$$

$$\alpha_5^{[8]A} = 8imM\delta(r-6M), \quad \alpha_9^{[8]A} = 8M\delta(r-6M). \tag{C.13}$$

The mode amplitudes of the first-order stress-energy tensor at 4PLT order, $t_{i\ell m}^{[9]}$, are given by

$$
\begin{aligned}
t_{i\ell m}^{[9]} &= \Delta\Omega^2 t_{i\ell m}^{[9]A} + F_{[0]}^{\Delta\Omega}\left(\partial_{\Delta\Omega}F_{[0]}^{\Delta\Omega}\right)t_{i\ell m}^{[9]B} \\
&:= -\frac{\mathcal{E}_*}{4}\left[\Delta\Omega^2\alpha_{i\ell m}^{[9]A} + F_{[0]}^{\Delta\Omega}\left(\partial_{\Delta\Omega}F_{[0]}^{\Delta\Omega}\right)\alpha_{i\ell m}^{[9]B}\right]\begin{cases} Y_{\ell m}^*(\frac{\pi}{2},0) & i = 1,\dots,7, \\ \partial_\theta Y_{\ell m}^*(\frac{\pi}{2},0) & i = 8,9,10, \end{cases}
\end{aligned}
\tag{C.14}
$$

where (dropping again the $\ell$ and $m$ indices)

$$\alpha_1^{[9]A} = -\frac{8}{9}M\left[5\delta + 60M\delta' - 144M^2\delta''\right], \quad \alpha_{2,5,9}^{[9]A} = 0,$$

$$\alpha_3^{[9]A} = -\frac{16}{3}M\left[\delta + 9M\delta' - 36M^2\delta''\right],$$

$$\alpha_4^{[9]A} = -\frac{16}{3}\sqrt{\frac{2}{3}}imM\left[\delta + 3M\delta' - 36M^2\delta''\right],$$

$$\alpha_6^{[9]A} = \frac{1}{3}M\left[5\delta + 36M\delta' + 144M^2\delta''\right], \tag{C.15}$$

$$\alpha_7^{[9]A} = \frac{1}{3}M\left[\ell(\ell+1) - 2m^2\right]\left[5\delta + 36M\delta' + 144M^2\delta''\right],$$

$$\alpha_8^{[9]A} = -\frac{16}{3}\sqrt{\frac{2}{3}}M\left[\delta + 3M\delta' - 36M^2\delta''\right],$$

$$\alpha_{10}^{[9]A} = \frac{2}{3}imM\left[5\delta + 36M\delta' + 144M^2\delta''\right],$$

and

$$\alpha_1^{[9]B} = -16\sqrt{6}M^2\left[\delta - 12M\delta'\right], \quad \alpha_{2,5,9}^{[9]B} = 0, \quad \alpha_3^{[9]B} = 288\sqrt{6}M^3\delta',$$

$$\alpha_4^{[9]B} = -96imM^2\left[\delta - 6M\delta'\right], \quad \alpha_6^{[9]B} = -18\sqrt{6}M^2\left[\delta - 4M\delta'\right],$$

$$\alpha_7^{[9]B} = -18\sqrt{6}M^2\left[\ell(\ell+1) - 2m^2\right]\left[\delta - 4M\delta'\right], \tag{C.16}$$

$$\alpha_8^{[9]B} = -96M^2\left[\delta - 6M\delta'\right], \quad \alpha_{10}^{[9]B} = -36\sqrt{6}imM^2\left[\delta - 4M\delta'\right].$$

Note that we have introduced $\delta := \delta(r-6M)$, $\delta' := \delta'(r-6M)$ and $\delta'' := \delta''(r-6M)$ to shorten expressions.

## C.4 Coefficients of the early-time transition-to-plunge solutions

Up to 7PLT order, the first coefficients in the early-time solutions for the orbital radius (118) are given by

$$r_{[0]}^{(2,1)} = -24\sqrt{6}M^2, \quad r_{[0]}^{(2,1-5i)} = 0, \ \forall i \geq 1, \tag{C.17}$$

$$r_{[1]}^{(3,-1)} = 0, \qquad r_{[1]}^{(3,-1-5i)} = 0, \ \forall\, i \geq 1, \tag{C.18}$$

$$r_{[2]}^{(4,2)} = 720 M^3, \qquad r_{[2]}^{(4,-3)} = \frac{9}{4}\sqrt{\frac{3}{2}}\left(f_{[5]}^{t}\right)^2, \tag{C.19}$$

$$r_{[3]}^{(5,0)} = -54 M^2 f_{[5]}^{r}, \qquad r_{[3]}^{(5,-5i)} = 0, \ \forall\, i \geq 1, \tag{C.20}$$

$$r_{[4]}^{(6,3)} = -3840\sqrt{6} M^4, \qquad r_{[4]}^{(6,-2)} = -\frac{9}{8} f_{[5]}^{t}\left(120 M f_{[5]}^{t} - \sqrt{6} f_{[7]A}^{t}\right), \tag{C.21}$$

$$r_{[5]}^{(7,1)} = 54 M^2 \left(14\sqrt{6} M f_{[5]}^{r} - f_{[7]A}^{r}\right), \tag{C.22}$$

$$r_{[6]}^{(8,4)} = 126720 M^5, \qquad r_{[6]}^{(8,-1)} = \frac{9}{8} f_{[5]}^{t}\left[12 M\left(52\sqrt{6} M f_{[5]}^{t} - 3 f_{[7]A}^{t}\right) - f_{[8]A}^{r}\right], \tag{C.23}$$

$$r_{[7]}^{(9,2)} = -27 M^2 \left(1560 M^2 f_{[5]}^{r} - 28 M\sqrt{6} f_{[7]A}^{r} + 2 f_{[9]A}^{r}\right). \tag{C.24}$$

The first coefficients in the early-time solutions of the $F_{[n]}^{\Delta\Omega}$ forcing terms (113) up to 7PLT order read

$$F_{[0]}^{(3,-1)} = \frac{f_{[5]}^{t}}{48 M^2}, \qquad F_{[0]}^{(3,-6)} = \sqrt{\frac{3}{2}}\frac{\left(f_{[5]}^{t}\right)^3}{2048 M^5}, \tag{C.25}$$

$$F_{[1]}^{(4,-3)} = 0, \qquad F_{[1]}^{(4,-3-5i)} = 0, \ \forall\, i \geq 0, \tag{C.26}$$

$$F_{[2]}^{(5,0)} = -\frac{1}{48 M^2}\left(7\sqrt{6} M f_{[5]}^{t} - f_{[7]A}^{t}\right), \tag{C.27}$$

$$F_{[2]}^{(5,-5)} = -\frac{\left(f_{[5]}^{t}\right)^2}{12288 M^5}\left(252 M f_{[5]}^{t} - 5\sqrt{6} f_{[7]A}^{t}\right), \tag{C.28}$$

$$F_{[3]}^{(6,-2)} = -\frac{f_{[5]}^{t}}{2304 M^4}\left(36\sqrt{6} M^2 f_{[5]}^{r} - 9 M f_{[7]A}^{r} - f_{[8]A}^{t}\right), \tag{C.29}$$

$$F_{[4]}^{(7,1)} = \frac{1}{48 M^2}\left(50 M^2 f_{[5]}^{t} - 7\sqrt{6} M f_{[7]A}^{t} + f_{[9]A}^{t}\right), \tag{C.30}$$

$$\begin{aligned} F_{[4]}^{(7,-4)} = \frac{f_{[5]}^{t}}{110592 M^6}\Big[ &18\sqrt{6} M\left(f_{[7]A}^{t}\right)^2 + 162\sqrt{6} M^3\left(f_{[5]}^{t}\right)^2 \\ &- f_{[5]}^{t}\left(f_{[9]B}^{t} + 9 M\left(f_{[8]A}^{r} - 3\sqrt{6} f_{[9]A}^{t} + 300 M f_{[7]A}^{t}\right)\right)\Big], \end{aligned} \tag{C.31}$$

$$\begin{aligned} F_{[5]}^{(8,-1)} = \frac{1}{2304 M^4}\Big[ &48 M^2 f_{[10]A}^{t} + f_{[7]A}^{t}\left(f_{[10]E}^{t} + 9 M\left(f_{[7]A}^{r} - 4\sqrt{6} M f_{[5]}^{r}\right)\right) \\ &+ f_{[5]}^{t}\left(f_{[10]D}^{t} + M\left(18 f_{[9]A}^{r} - 7\sqrt{6} f_{[8]A}^{t} - 7\sqrt{6} f_{[10]E}^{t}\right.\right. \\ &\left.\left. - 162\sqrt{6} M f_{[7]A}^{r} + 3888 M^2 f_{[5]}^{r}\right)\right) + 432 M^3\left(F_{[0]}^{\delta M}\partial_{\delta M} f_{[5]}^{r} + F_{[0]}^{\delta J}\partial_{\delta J} f_{[5]}^{r}\right)\Big], \end{aligned} \tag{C.32}$$

$$F_{[6]}^{(9,2)} = \frac{1}{288 M^2}\left(1836\sqrt{6} M^3 f_{[5]}^{t} + 300 M^2 f_{[7]A}^{t} - 42\sqrt{6} M f_{[9]A}^{t} + 6 f_{[11]F}^{t}\right), \tag{C.33}$$

$$
\begin{aligned}
F_{[6]}^{(9,-3)} = \frac{f_{[5]}^{t}}{110592 M^6} \Big[ & f_{[5]}^{t} f_{[11]D}^{t} - f_{[5]}^{t} f_{[11]E}^{t} + 7\sqrt{6} M f_{[5]}^{t} f_{[9]B}^{t} + 9M f_{[7]A}^{r} f_{[11]A}^{t} \\
& + 7\sqrt{6} M f_{[5]}^{t} f_{[11]B}^{t} + 27\sqrt{6} M f_{[5]}^{t} f_{[11]F}^{t} + 81 M^2 \left( f_{[7]A}^{r} \right)^2 \\
& + 90\sqrt{6} M^2 f_{[5]}^{t} f_{[8]A}^{r} - 432 M^2 \left( f_{[7]A}^{t} \right)^2 - 2268 M^2 f_{[5]}^{t} f_{[9]A}^{t} \\
& - 36\sqrt{6} M^2 f_{[5]}^{r} f_{[11]A}^{t} - 648\sqrt{6} M^3 f_{[5]}^{r} f_{[7]A}^{r} + 7776 M^4 \left( f_{[5]}^{r} \right)^2 \\
& + 614304 M^4 \left( f_{[5]}^{t} \right)^2 + f_{[8]A}^{t} \left( f_{[11]A}^{t} + 9M \left( f_{[7]A}^{r} - 4\sqrt{6} M f_{[5]}^{r} \right) \right) \\
& - f_{[7]A}^{t} \left( f_{[11]B}^{t} + 9M \left( f_{[8]A}^{r} - 2\sqrt{6} \left( f_{[9]A}^{t} - 221 M^2 f_{[5]}^{t} \right) \right) \right) \Big] ,
\end{aligned}
\tag{C.34}
$$

$$
\begin{aligned}
F_{[7]}^{(10,0)} = \frac{1}{2304 M^4} \Big[ & f_{[7]A}^{t} f_{[12]K}^{t} + 18 M f_{[9]A}^{r} f_{[7]A}^{t} - 7\sqrt{6} M f_{[7]A}^{t} f_{[10]E}^{t} \\
& - 7\sqrt{6} M f_{[7]A}^{t} f_{[12]A}^{t} - 162\sqrt{6} M^2 f_{[7]A}^{r} f_{[7]A}^{t} + 48 M^2 f_{[12]I}^{t} \\
& + 3888 M^3 f_{[5]}^{r} f_{[7]A}^{t} - 336\sqrt{6} M^3 f_{[10]A}^{t} \\
& + f_{[9]A}^{t} \left( f_{[12]A}^{t} + 9M \left( f_{[7]A}^{r} - 4\sqrt{6} M f_{[5]}^{r} \right) \right) \\
& + f_{[5]}^{t} \big( f_{[12]J}^{t} + M \big( 27 f_{[11]F}^{r} - 7\sqrt{6} f_{[10]D}^{t} - 7\sqrt{6} f_{[12]K}^{t} - 288\sqrt{6} M f_{[9]A}^{r} \\
& + 50 M f_{[8]A}^{t} + 294 M f_{[10]E}^{t} + 50 M f_{[12]A}^{t} + 8082 M^2 f_{[7]A}^{r} - 37944\sqrt{6} M^3 f_{[5]}^{r} \big) \big) \\
& + 432 M^3 \left( \left( F_{[2]A}^{\delta M} - 7\sqrt{6} M F_{[0]}^{\delta M} \right) \partial_{\delta M} f_{[5]}^{r} + F_{[0]}^{\delta M} \partial_{\delta M} f_{[7]A}^{r} \right. \\
& \left. + \left( F_{[2]A}^{\delta J} - 7\sqrt{6} M F_{[0]}^{\delta J} \right) \partial_{\delta J} f_{[5]}^{r} + F_{[0]}^{\delta J} \partial_{\delta J} f_{[7]A}^{r} \right) \Big] .
\end{aligned}
\tag{C.35}
$$

# D  Material for sections 5 and 6

## D.1  Self-force matching conditions

The self-force matching conditions involved in the asymptotic match between the quasi-circular inspiral (up to 2PA order) and the transition to plunge (up to 7PLT order) are given by

$$
f_{[5]}^{\mu} = f_{(1)}^{\mu} \Big|_{*} ,
\tag{D.1}
$$

$$
f_{[7]A}^{\mu} = \partial_{\Omega} f_{(1)}^{\mu} \Big|_{*} ,
\tag{D.2}
$$

$$
f_{[8]A}^{\mu} = f_{(2)B}^{\mu} \Big|_{*} ,
\tag{D.3}
$$

$$
f_{[9]A}^{\mu} = \frac{1}{2} \partial_{\Omega}^{2} f_{(1)}^{\mu} \Big|_{*} , \quad f_{[9]B}^{\mu} = f_{(3)E}^{\mu} \Big|_{*} ,
\tag{D.4}
$$

$$
f_{[10]A}^{\mu} = f_{(2)A}^{\mu} \Big|_{*} , \quad f_{[10]D}^{\mu} = \partial_{\Omega} f_{(2)B}^{\mu} \Big|_{*} , \quad f_{[10]E}^{\mu} = f_{(2)B}^{\mu} \Big|_{*} ,
\tag{D.5}
$$

$$
f_{[11]A}^{\mu} = f_{(3)C}^{\mu} \Big|_{*} , \quad f_{[11]B}^{\mu} = f_{(3)E}^{\mu} \Big|_{*} , \quad f_{[11]D}^{\mu} = f_{(3)D}^{\mu} \Big|_{*} ,
$$

$$
f_{[11]E}^{\mu} = \partial_{\Omega} f_{(3)E}^{\mu} \Big|_{*} , \quad f_{[11]F}^{\mu} = \frac{1}{6} \partial_{\Omega}^{3} f_{(1)}^{\mu} \Big|_{*} ,
\tag{D.6}
$$

$$
f_{[12]A}^{\mu} = f_{(2)B}^{\mu} \Big|_{*} , \quad f_{[12]I}^{\mu} = \partial_{\Omega} f_{(2)A}^{\mu} \Big|_{*} , \quad f_{[12]J}^{\mu} = \frac{1}{2} \partial_{\Omega}^{2} f_{(2)B}^{\mu} \Big|_{*} , \quad f_{[12]K}^{\mu} = \partial_{\Omega} f_{(2)B}^{\mu} \Big|_{*} .
\tag{D.7}
$$

We recall that $|_{*}$ denotes evaluation of functions (in this case of $\Omega$, $\delta M$ and $\delta J$) at $\Omega = \Omega_{*}$.

### D.2 Explicit values of the asymptotic coefficents

The breakdown and critical frequencies of the inspiral motion require the knowledge of the coefficients $F_{(0)/[0]}^{(3,-1)}$, $F_{(0)/[4]}^{(7,1)}$, $F_{(0)/[8]}^{(11,3)}$, $F_{(1)/[3]}^{(6,-2)}$ and $F_{(2)/[0]}^{(3,-6)}$ defined, respectively, in eqs. (B.11), (B.13), (B.15), (B.16) and (B.19). We take the value $F_{(1)/[3]}^{(6,-2)} = -3.537409407224891 \times 10^{-6}$ (setting $M = 1$) from [18], while evaluating the remaining coefficients only requires the first-order self-force data $f_{(1)}^t(\Omega_*)$ and $\partial_\Omega^n f_{(1)}^t(\Omega_*)$ with $n = 1, 2, 3, 4$.

We can obtain all the necessary self-force data from the first-order energy flux $\mathcal{F}_{0PA}$ of [53],

$$f_{(1)}^t(\Omega) = g^{tt}(r_{(0)})f_t^{(1)} = -\frac{U_{(0)}}{f(r_{(0)})}\mathcal{F}_{0PA} = -\frac{\mathcal{F}_{0PA}^\infty(\Omega) + \mathcal{F}_{0PA}^{\mathcal{H}}(\Omega)}{\left(1 - 3(M\Omega)^{2/3}\right)^{1/2}\left(1 - 2(M\Omega)^{2/3}\right)}. \tag{D.8}$$

The coefficients we are interested in then evaluate to (setting $M = 1$)

$$F_{(0)/[0]}^{(3,-1)} = -4.155752096668726 \times 10^{-5}, \tag{D.9}$$

$$F_{(0)/[4]}^{(7,1)} = -3.280188141361921 \times 10^{-2}, \tag{D.10}$$

$$F_{(0)/[8]}^{(11,3)} = -5.104634123185608 \times 10^{-2}, \tag{D.11}$$

$$F_{(2)/[0]}^{(3,-6)} = -4.746661778492238 \times 10^{-12}. \tag{D.12}$$

The value of the coefficient $F_{(2)/[2]}^{(5,-5)}$ (B.20) appearing in eq. (134) is

$$F_{(2)/[2]}^{(5,-5)} = -3.559185516345344 \times 10^{-10}. \tag{D.13}$$

Finally, setting again $M = 1$, we also obtain the numerical values for $f_{[5]}^t$ and $f_{[7]A}^t$ (which are related to the first-order self-force (D.8) through the matching conditions (D.1) and (D.2), respectively)

$$f_{[5]}^t = -1.994761006400989 \times 10^{-3}, \tag{D.14}$$

$$f_{[7]A}^t = -1.30787432411794 \times 10^{-1}. \tag{D.15}$$

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
