# Peer review of "Self-force framework for transition-to-plunge waveforms"

_SciPost Physics, doi:SciPost Phys. 17, 056 (2024)_

## Round 1 · Referee Report · Scott Hughes (Referee 1) · 2024-6-30

Strengths

A totally thorough and comprehensive analysis.
Well written, not difficult to follow.
Excellent comparisons with relevant past literature.

Weaknesses

None. (I provide a few minor pieces of feedback and suggestions, none of crucial importance.)

Report

The so-called "two-body" problem in general relativity is an extremely challenging problem in general, thanks to the non-linearity of the underlying gravitational field equations, the existence and contribution of radiation, and complicated boundary conditions associated with event horizons. Indeed, when speaking to general audiences, I like to emphasize that (at least for binary black hole systems) the "two-body" problem is really a "one-spacetime" problem, with this single spacetime having two-body properties in certain important limits. Although in principle just about any interesting two-body configuration can now be evolved numerically, in practice numerical relativity remains somewhat limited in scope. Especially with respect to the gravitational-wave applications which drive this field, the role of numerical relativity may be limited by accuracy and precision for a range of important astrophysical sources. Analytic or quasi-analytic methods which can provide highly precise input to this problem are thus of great value, even if they formally only apply to a limited domain of the two-body problem's parameter space.

This paper provides an extremely comprehensive study of an aspect of this problem, the transition from the inspiral (when the binary consists of two unambiguous separated bodies slowly spiraling together driven by gravitational backreaction) to plunge (when the smaller member of the binary falls into the larger and the system becomes a single black hole). Indeed, the manuscript is so comprehensive that it reads almost more like a monograph that a research paper! (Because of this, it took quite a while to find time to go through the draft thoroughly, and I regret the delay providing this report.)

The technique employed by these authors uses a separation of timescales, which follows in turn from a separation in mass scales between the binary's members. Although limited in the category of problems that it focuses upon here (it studies the transition from quasi-circular inspiral of a secondary body into a Schwarzschild black hole), its presentation is sufficiently general that many aspects of what the authors present can be expected to carry over to more generalized versions of this problem. Although I have a few suggestions that I would like the authors to consider, none are of crucial importance for the paper. Modulo their consideration of these points (which I list in the section "Requested changes", though I emphasize that these points can be considered optional), I am very happy to recommend this paper for publication. Their approach is very elegant and complete. This manuscript essentially reads like a textbook to any practitioner interested in understanding this problem.

Requested changes

The listing I provide here is in order of where I encountered the issue or text for which I have a suggestion; this is not a ranking of importance.

  1. On page 2 of the draft manuscript, the second complete paragraph begins "Accurately modeling the transition to plunge is expected to improve parameter estimation...". A related point that may be made here is that a proper handling of the transition and plunge significantly improves the utility of inspiral waveforms currently being investigated for the development of LISA data analysis. An inspiral-only waveform abruptly terminates when the secondary reaches the last stable orbit. This termination introduces spurious features into the time-frequency behavior, reducing the value of inspiral-only waveforms for data-analysis studies. One can taper such a waveform to reduce the influence of the late-inspiral behavior and avoid the abrupt termination, but a physically motivated termination is even more valuable. The framework provided here is probably more than is needed to "smooth" the waveforms' behavior for ongoing data-analysis studies, but it is a valuable motivator for the research program as a whole.

  2. In the text following Equation (2.18), the authors describe an expansion of $h_{\mu\nu}$ as identical to the expansion of $\bar h_{\mu\nu}$ but "with the $i = 3, 6$ terms flipped". It would be helpful to describe precisely what "flipped" means here -- are these terms of opposite sign? Does the $i = 3$ term in the $\bar h$ expansion somehow change places with the $i = 6$ term in the $h$ expansion? The term "flipped" on its own is a bit ambiguous, but the authors should be able to easily fix this minor bit of terminology.

  3. The paragraph which follows Equation (5.10) contains the text "..the composite solution should not be trusted for sufficiently large mass ratios." One should be somewhat cautious here, since in some contexts and in some papers "large" mass ratio means "one body much more massive than the other". Such a term is nothing more than a simple remapping of the "one body much less massive than the other" small mass ratio used in this paper. I am fairly confident that the authors' concern is for the case of mass ratios close to unity, and if correct would suggest rewording this sentence to something like

"In conclusion, the composite solution should not be trusted as the mass ratio approaches unity."

  1. It is very salubrious to see the authors explain the Apte-Hughes model in the context of their significantly more complete framework, and in particular to see how that simpler model could be extended based on the framework developed in this paper. One of the motivations of the Apte-Hughes model was to have a method for describing the transition and plunge (if only approximately) for all black hole spins. In this context, it would be useful to describe at least schematically the challenge of extending these results to Kerr. I imagine that many aspects of the two-timescale expansion remain unchanged, but that solving for the self-force corrections may be significantly more complicated. Though beyond the scope of what the authors consider here, this problem is a natural point for additional work, so a brief discussion of these challenges would be appropriate.

Recommendation

Publish (surpasses expectations and criteria for this Journal; among top 10%)

---

## Round 1 · Referee Report · Paul Ramond (Referee 2) · 2024-7-24

Strengths

  1. Presents an anticipated and essential result in the field
  2. Clear and well-structured exhibition of the results and methods
  3. Helpful contextualization and thorough comparison with existing literature

Weaknesses

None.

Report

The paper presents a complete analysis of the dynamics during the "transition-to-plunge" part of the coalescence of an asymmetric binary system. The multiscale framework adopted and detailed in the paper leads naturally to a waveform generation scheme that incorporates this new aspect of the two-body dynamics, from the late moments of inspiral all the way to the plunge. This work, to the extent provided in the article, was currently lacking in the literature on self-forced waveform generation schemes whose primary focused was, until now, on the "inspiral" part, before the last stable orbit.

The logic of the presentation is impeccable, and the whole manuscript reads very well. It makes precise references to previous works on which it builds, both regarding the context and the technical details. There is a nice balance of necessary technical details and more broad, synthetic considerations, obtained by relegating secondary material to appendices or external references. The numerical tools are freely available and make use of open-source software packages, making it all the more valuable.

Even though I am not an expert on these types of self-force calculations, thanks to the remarkably clear exposition, I was able to identify the challenges that the authors had to face and recognize the quality of the work. I strongly recommend this paper for publication. The comments I have made in the "requested changes" below can be considered as simple suggestions that may improve the paper's content to some readers, and the article may be very well be published as is.

Requested changes

  1. Discussion on extensions. As stated by the authors, this work fills a gap in the currently available waveform generation schemes. However, in order to really be exploited as such, one will require a more realistic model for the binary. In particular, a fully generic orbital configuration (inclined, eccentric), with spinning bodies (extending Schwarzschild background to Kerr and possibly accounting for the secondary's spin) will have to undergo the same treatment as the one given here (circular, Schwarzschild). There is no doubt that this will be the goal of follow-up papers, of which the present one contains (most of, if not all) the necessary technical material. It would be interesting to discuss this in the paper, in a dedicated section, where the challenges brought forth by considering more realistic orbits are described and how the tools and results are expected to be extended or generalized. In particular, in the tools/method devised in the paper, what works specifically for circular orbits in Schwarzschild, and what is independent of it? Are there any other challenges when going to Kerr and/or more intricate orbits?

  2. First principles. It is my understanding that one strength of the approach chosen by the authors, compared to previous attempts, is its "first-principles" nature. I would like to draw the attention to this particular feature. When the authors say this, do they mean that the waveform generation scheme including the transition to plunge they provided only requires the usual EMRI parameters typically considered for inspirals (masses, spins, orbital configuration and celestial orientation) ? If yes, I believe that this could be more clearly emphasized. If not, what is enforced "by hand" in the calculation or requires a "choice" (perhaps in the matching conditions) ? Could these be lifted to make the scheme completely dependent on input parameters values? I believe that such discussion could help the reader understand what is meant by "first principles" in the paper.

Recommendation

Publish (surpasses expectations and criteria for this Journal; among top 10%)

---

## Editorial Decision

published